# On Sample-Efficient Offline Reinforcement Learning: Data Diversity, Posterior Sampling, and Beyond

**Thanh Nguyen-Tang**
Johns Hopkins University
Baltimore, MD 21218
nguyent@cs.jhu.edu

**Raman Arora**
Johns Hopkins University
Baltimore, MD 21218
arora@cs.jhu.edu

## Abstract

We seek to understand what facilitates sample-efficient learning from historical datasets for sequential decision-making, a problem that is popularly known as offline reinforcement learning (RL). Further, we are interested in algorithms that enjoy sample efficiency while leveraging (value) function approximation. In this paper, we address these fundamental questions by (i) proposing a notion of data diversity that subsumes the previous notions of coverage measures in offline RL and (ii) using this notion to *unify* three distinct classes of offline RL algorithms based on version spaces (VS), regularized optimization (RO), and posterior sampling (PS). We establish that VS-based, RO-based, and PS-based algorithms, under standard assumptions, achieve *comparable* sample efficiency, which recovers the state-of-the-art sub-optimality bounds for finite and linear model classes with the standard assumptions. This result is surprising, given that the prior work suggested an unfavorable sample complexity of the RO-based algorithm compared to the VS-based algorithm, whereas posterior sampling is rarely considered in offline RL due to its explorative nature. Notably, our proposed model-free PS-based algorithm for offline RL is *novel*, with sub-optimality bounds that are *frequentist* (i.e., worst-case) in nature.

## 1 Introduction

Learning from previously collected experiences is a vital capability for reinforcement learning (RL) agents, offering a broader scope of applications compared to online RL. This is particularly significant in domains where interacting with the environment poses risks or high costs. However, effectively extracting valuable policies from historical datasets remains a considerable challenge, especially in high-dimensional spaces where the ability to generalize across various scenarios is crucial. In this paper, our objective is to comprehensively examine the efficiency of offline RL in the context of (value) function approximation. We aim to analyze this within the broader framework of general data collection settings.

The problem of learning from historical datasets for sequential decision-making, commonly known as *offline RL* or *batch RL*, originated in the early 2000s [Ernst et al., 2005, Antos et al., 2006, Lange et al., 2012] and has recently regained significant attention [Levine et al., 2020, Uehara et al., 2022a]. In offline RL, where direct interaction with environments is not possible, our goal is to learn an effective policy by leveraging pre-collected datasets, typically obtained from different policies known as *behavior policies*. The sample efficiency of an offline RL algorithm is measured by the sub-optimality of the policies it executes compared to a "good" comparator policy, which may or may not be an optimal policy. Due to the lack of exploration inherent in offline RL, designing an algorithm with low sub-optimality requires employing the fundamental principle of *pessimistic extrapolation*. This means that the agent extrapolates from the offline data while considering the worst-case scenarios

37th Conference on Neural Information Processing Systems (NeurIPS 2023).

that are consistent with that data. Essentially, the *diversity* present in the offline data determines the agent's ability to construct meaningful extrapolations. Hence, a suitable notion of data diversity plays a crucial role in offline RL.

To address the issue of data diversity, several prior methods have made the assumption that the offline data is *uniformly diverse* – this implies that the data should cover the entire trajectory space with some probability that is bounded from below [Munos and Szepesvári, 2008, Chen and Jiang, 2019, Nguyen-Tang et al., 2022b]. This assumption is often too strong and not feasible in many practical scenarios. In more recent approaches [Jin et al., 2021b, Xie et al., 2021, Uehara and Sun, 2022, Chen and Jiang, 2022, Rashidinejad et al., 2023], the stringent assumption of uniform diversity has been relaxed to only require *partial* diversity in the offline data. Various measures have been proposed to capture this partial diversity, such as single-policy concentrability coefficients [Liu et al., 2019, Rashidinejad et al., 2021, Yin and Wang, 2021], relative condition numbers [Agarwal et al., 2021, Uehara and Sun, 2022], and Bellman residual ratios [Xie et al., 2021]. These measures aim to quantify the extent to which the data captures diverse states and behaviors. However, it should be noted that in some practical scenarios, these measures may become excessive or may not hold at all.

In terms of algorithmic approaches, existing sample-efficient offline RL algorithms explicitly construct pessimistic estimates of models or value functions to effectively learn from datasets with partial diversity. This is typically achieved through the construction of lower confidence bounds (LCBs) [Jin et al., 2021b, Rashidinejad et al., 2021] or version spaces (VS) [Xie et al., 2021, Zanette et al., 2021]. LCB-based algorithms incorporate a bonus term subtracted from the value estimates to enforce pessimism across all state-action pairs and stages. However, it has been observed that LCB-based algorithms tend to impose unnecessarily aggressive pessimism, leading to sub-optimal bounds [Zanette et al., 2021]. On the other hand, VS-based algorithms search through the space of consistent hypotheses to identify the one with the smallest value in the initial states. These algorithms have demonstrated state-of-the-art bounds [Zanette et al., 2021, Xie et al., 2021].

In contrast to LCB-based and VS-based algorithms, regularized (minimax) optimization (RO) and posterior sampling (PS) are more amenable to tractable implementations but are relatively new in the offline RL literature. The RO-based algorithm initially introduced by [Xie et al., 2021, Algorithm 1] incorporates pessimism implicitly through a regularization term that promotes pessimism in the initial state. This approach eliminates the need for an intractable search over the version space. However, Xie et al. [2021] demonstrate that the RO-based algorithm exhibits a significantly slower sub-optimality rate than standard VS-based algorithms. Specifically, the RO-based algorithm achieves a sub-optimality rate of $K^{-1/3}$, whereas VS-based algorithms achieve a faster rate of $K^{-1/2}$, where $K$ represents the number of episodes in the offline data.

On the other hand, posterior sampling (PS) [Thompson, 1933, Russo and Van Roy, 2014], a popular and successful method in online RL, is rarely explored in the context of offline RL. PS involves sampling from a constructed posterior distribution over the model or value function and acting accordingly. However, PS is less commonly considered in offline RL due to its explorative nature, which stems from the randomness of the posterior distribution. This randomness is well-suited for addressing the exploration challenge in online RL tasks [Zhang, 2022, Dann et al., 2021, Zhong et al., 2022, Agarwal and Zhang, 2022]. The only work that considers PS for offline RL is Uehara and Sun [2022], where they maintain a posterior distribution over Markov decision process (MDP) models. However, this model-based PS approach is limited to small-scale problems where computing the optimal policy from an MDP model is computationally feasible. In addition, this work only provides a weak form of guarantees via Bayesian bounds.

In the context of (value) function approximation, achieving sample-efficient offline RL relies on certain conditions that facilitate effective learning. The identification of the minimum condition required for sample efficiency, as well as the algorithms that can exploit such conditions, is an important research question that we aim to address here. We advance our understanding by making the following contributions: (I) *We introduce a new notion of data diversity that subsumes and expands all the prior distribution shift measures in offline RL*, and (II) *We show that all VS-based, RO-based and PS-based algorithms are in fact (surprisingly) competitive to each other, i.e., under standard assumptions, they achieve the same sub-optimality bounds (up to constant and log factors).* We summarize our key results in comparison with related work in Table 1. Our results further expand the class of sample-efficient offline RL problems (Figure 1) and provide more choices of offline RL algorithms with competitive guarantees and tractable approximations for practitioners to choose from.

| Algorithms | Sub-optimality Bound | Data |
|---|---|---|
| VS in [Xie et al., 2021] | $Hb\sqrt{C_2(\pi) \cdot \ln(\|\mathcal{F}\| \cdot \|\Pi^{all}\|)} \cdot K^{-1/2}$ | I |
| RO in [Xie et al., 2021] | $Hb\sqrt{C_2(\pi)} \cdot \sqrt[3]{\ln(\|\mathcal{F}\| \cdot \|\Pi^{soft}(T)\|)} \cdot K^{-1/3} + Hb/\sqrt{T}$ | I |
| MBPS in [Uehara and Sun, 2022] | $Hb\sqrt{C^{Bayes} \cdot \ln\|\mathcal{M}\|} \cdot K^{-1/2}$ (Bayesian) | I |
| VS in Algorithm 2 | $Hb\sqrt{\mathcal{C}(\pi; 1/\sqrt{K}) \cdot \ln(\|\mathcal{F}\| \cdot \|\Pi^{soft}(T)\|)} \cdot K^{-1/2} + Hb/\sqrt{T}$ | A |
| RO in Algorithm 3 | $Hb\sqrt{\mathcal{C}(\pi; 1/\sqrt{K}) \cdot \ln(\|\mathcal{F}\| \cdot \|\Pi^{soft}(T)\|)} \cdot K^{-1/2} + Hb/\sqrt{T}$ | A |
| MFPS in Algorithm 4 | $Hb\sqrt{\mathcal{C}(\pi; 1/\sqrt{K}) \cdot \ln(\|\mathcal{F}\| \cdot \|\Pi^{soft}(T)\|)} \cdot K^{-1/2} + Hb/\sqrt{T}$ (frequentist) | A |

Table 1: Comparison of our bounds with SOTA bounds for offline RL under partial coverage and function approximation, where gray cells mark our contributions. **Algorithms**: VS = version space, RO = regularized optimization, MBPS = model-based posterior sampling, and MFPS = model-free posterior sampling. **Sub-optimality bound**: $K$ = #number of episodes, $\pi$ = an *arbitrary* comparator policy, $H$ = horizon, $b$ = boundedness, $T$ = the number of algorithmic updates, $\ln\|\mathcal{F}\|, \ln\|\Pi^{soft}(T)\|, \ln\|\Pi^{all}\|, \ln\|\mathcal{M}\|$: complexity measures of some value function class $\mathcal{F}$, "induced" policy class $\Pi^{soft}(T)$, the class of all comparator policies $\Pi^{all}$, and model class $\mathcal{M}$, where typically $\Pi^{soft}(T) \subset \Pi^{all}, \forall T$. **Data**: I = independent episodes, A = adaptively collected data. Here $\mathcal{C}(\pi; 1/\sqrt{K})$ and $C_2(\pi)$ are some measures of extrapolation from the offline data to target policy $\pi$.

For establishing (II), we need to construct concrete VS-based, RO-based and PS-based algorithms. While the key components of the VS-based and RO-based algorithms appear in the literature [Xie et al., 2021], we propose a novel, a first-of-its-kind, model-free posterior sampling algorithm for offline RL. The algorithm contains two new ingredients: a pessimistic prior that encourages pessimistic value functions when being sampled from the posterior distribution and integration of posterior sampling with the actor-critic framework that incrementally updates the learned policy.

**Overview of Techniques.** Our analysis method presents a "decoupling" argument tailored for the batch setting, drawing inspiration from recent decoupling arguments in the online RL setting [Foster et al., 2021, Jin et al., 2021a, Zhang, 2022, Dann et al., 2021, Zhong et al., 2022, Agarwal and Zhang, 2022]. The core idea behind our decoupling argument is to establish a relationship between the Bellman error under any comparator policy $\pi$ and the squared Bellman error under the behavior policy. This relationship is mediated through our novel concept of data diversity, denoted as $\mathcal{C}(\pi; \epsilon_c)$, which is defined in detail in Definition 3. This allows to separate the sub-optimality of a learned policy into two main sources of errors: the extrapolation error, which captures the out-of-distribution (OOD) generalization from the behavior policy to a target policy, and the in-distribution error, which focuses on generalization within the same behavior distribution. The OOD error is effectively managed by controlling the data diversity $\mathcal{C}(\pi; \epsilon_c)$, while the in-distribution error is carefully addressed by utilizing the algorithmic structures and the martingale counterpart to Bernstein's inequality (i.e., Freedman's inequality).

In the process of bounding the in-distribution error of our proposed PS algorithm that we built upon the technique of Dann et al. [2021], we correct a non-rigorous argument of Dann et al. [2021] (which we discuss in detail in Section E.3.1) and develop a new technical argument to handle the statistical dependence induced by the data-dependent target policy in the actor-critic framework. Our new argument carefully incorporates the uniform convergence argument into the in-expectation bounds of PS. We give a detailed description of this argument in Section E.3. As an immediate application, our technique fixes a technical mistake involving how to handle the statistical dependence induced by the min player in the self-play posterior sampling algorithm of Xiong et al. [2022].

## 2 Background and Problem Formulation

### 2.1 Episodic Time-inhomogenous Markov Decision Process

Let $\mathcal{S}$ and $\mathcal{A}$ denote Lebesgue-measurable state and action spaces (possibly infinite), respectively. Let $\mathcal{P}(\mathcal{S})$ denote the space of all probability distributions over $\mathcal{S}$. We consider an episodic time-inhomogeneous Markov decision process $M = (\mathcal{S}, \mathcal{A}, P, r, H)$, where $P = \{P_h\}_{h \in [H]} \in$

$\{\mathcal{S} \times \mathcal{A} \to \mathcal{P}(\mathcal{S})\}^H$ are the transition probabilities (where $[H] := \{1, \ldots, H\}$), $r = \{r_h\}_{h \in [H]} \in \{\mathcal{S} \times \mathcal{A} \to \mathbb{R}\}^H$ is the mean reward functions, and $H \in \mathbb{N}$ is the length of the horizon for each episode. For any policy $\pi = \{\pi_h\}_{h \in [H]} \in \{\mathcal{S} \to \mathcal{P}(\mathcal{A})\}^H$, the action-value functions and the value functions under policy $\pi$ are defined, respectively, as $Q_{h,M}^\pi(s,a) = \mathbb{E}_\pi[\sum_{i=h}^H r_i(s_i, a_i) \mid (s_h, a_h) = (s, a)]$, and $V_{h,M}^\pi(s) = \mathbb{E}_\pi[\sum_{i=h}^H r_i(s_i, a_i) | s_h = s]$. Here $\mathbb{E}_\pi[\cdot]$ denotes the expectation with respect to the randomness of the trajectory $(s_h, a_h, \ldots, s_H, a_H)$, with $a_i \sim \pi_i(\cdot|s_i)$ and $s_{i+1} \sim P_i(\cdot|s_i, a_i)$ for all $i$. For any policy $\pi$, we define the visitation density probability functions $d_M^\pi = \{d_{h,M}^\pi\}_{h \in [H]} \in \{\mathcal{S} \times \mathcal{A} \to \mathbb{R}_+\}^H$ as $d_h^\pi(s, a) := \frac{d\Pr((s_h, a_h)=(s,a)|\pi, M)}{d\rho(s,a)}$ where $\rho$ is the Lebesgue measure on $\mathcal{S} \times \mathcal{A}$ and $\Pr((s_h, a_h) = (s, a)|\pi, M)$ is the probability of policy $\pi$ reaching state-action pair $(s, a)$ at timestep $h$. The Bellman operator $\mathbb{T}_h^\pi$ is defined as $[\mathbb{T}_h^\pi Q](s, a) := r_h(s, a) + \mathbb{E}_{s' \sim P_h(\cdot|s,a), a' \sim \pi_{h+1}(\cdot|s')}[Q(s', a')]$, for any $Q : \mathcal{S} \times \mathcal{A} \to \mathbb{R}$. Let $\pi^*$ be an optimal policy, i.e., $Q_h^{\pi^*}(s, a) \geq Q_h^\pi(s, a), \forall (s, a, h, \pi) \in \mathcal{S} \times \mathcal{A} \times [H] \times \Pi^{all}$, where $\Pi^{all} := \{\mathcal{S} \to \mathcal{P}(\mathcal{A})\}^H$ is the set of all possible policies. For simplicity, we assume that the initial state $s_1$ is deterministic across all episodes.[1] We also assume that there is some $b > 0$ such that for any trajectory $(s_1, a_1, r_1, \ldots, s_H, a_H, r_H)$ generated under any policy, $|r_h| \leq b, \forall h$ and $|\sum_{h=1}^H r_h| \leq b$ almost surely.[2] This boundedness assumption is standard and subsumes the boundedness conditions in the previous works, e.g., Zanette et al. [2021] set $b = 1$ and Jin et al. [2021b] use $b = H$ (and further assume that $r_h \in [0, 1], \forall h$).[3] Without loss of generality, we assume that $b \geq 1$.

**Additional Notation.** For any $u : \mathcal{S} \times \mathcal{A} \to \mathbb{R}$ and any $\pi : \mathcal{S} \to \mathcal{P}(\mathcal{A})$, we overload the notation $u(s, \pi) := \mathbb{E}_{a \sim \pi(\cdot|s)}[u(s, a)]$. For any $f : \mathcal{S} \times \mathcal{A} \to \mathbb{R}$, denote the supremum norm $\|f\|_\infty = \max_{(s,a) \in \mathcal{S} \times \mathcal{A}} |f(s, a)|$. We write $\mathbb{E}[g]^2 := (\mathbb{E}[g])^2$. For a probability measure $\nu$ on some measurable space $(\Omega, \mathcal{B})$, we denote by $\mathrm{supp}(\nu)$ the support of $\nu$, $\mathrm{supp}(\nu) := \{B \in \mathcal{B} : \nu(B) > 0\}$. We denote $x \lesssim y$ to mean that $x = \mathcal{O}(y)$.

## 2.2 Offline Data Generation

Denote the pre-collected dataset by $\mathcal{D} := \{(s_h^t, a_h^t, r_h^t)\}_{h \in [H]}^{t \in [K]}$, where $s_{h+1}^t \sim P_h(\cdot|s_h^t, a_h^t)$ and $\mathbb{E}[r_h^t|s_h^t, a_h^t] = r_h(s_h^t, a_h^t)$. We consider the adaptively collected data setting where the offline data is collected by *time-varying* behavior policies $\{\mu^k\}_{k \in [K]}$, concretely, defined as follows.

**Definition 1** (Adaptively collected data[4]). *$\mu^k$ is a function of $\{(s_h^i, a_h^i, r_h^i)\}_{h \in [H]}^{i \in [k-1]}$, $\forall k \in [K]$.*

For simplicity, we denote $\mu = \frac{1}{K}\sum_{k=1}^K \mu^k$, $d^\mu = \frac{1}{K}\sum_{k=1}^K d^{\mu^k}$, and $\mathbb{E}_\mu[\cdot] = \frac{1}{K}\sum_{k=1}^K \mathbb{E}_{\mu^k}[\cdot]$. The setting of adaptively collected data covers a common practice where the offline data is collected by using some adaptive experimentation [Zhan et al., 2023]. When $\mu^1 = \cdots = \mu^K$, it recovers the setting of independent episodes in Duan et al. [2020].

**Value sub-optimality.** The goodness of a learned policy $\hat{\pi} = \hat{\pi}(\mathcal{D})$ against a comparator policy $\pi$ for the underlying MDP $M$ is measured by the (value) sub-optimality defined as

$$\mathrm{SubOpt}_\pi^M(\hat{\pi}) := V_1^\pi(s_1) - V_1^{\hat{\pi}}(s_1). \tag{1}$$

Whenever the context is clear, we drop $M$ in $Q_M^\pi$, $V_M^\pi$, $d_M^\pi$, and $\mathrm{SubOpt}_\pi^M(\hat{\pi})$.

## 2.3 Policy and function classes

Next, we define the policy space and the action-value function space over which we optimize the value sub-optimality. We consider a (Cartesian product) function class $\mathcal{F} = \mathcal{F}_1 \times \cdots \times \mathcal{F}_H \in \{\mathcal{S} \times \mathcal{A} \to [-b, b]\}^H$. The function class $\mathcal{F}$ induces the following (Cartesian product) policy class

---

[1]This assumption is merely for the sake of clean presentation which does not affect any results.

[2]Note that we allow the reward samples to be negative.

[3]We can replace the condition $|r_h| \leq b, \forall h$ with 1-sub-Gaussian condition: $r_h \sim R_h(s_h, a_h)$ wherein $R_h(s_h, a_h)$ is sub-Gaussian with mean $r_h(s_h, a_h)$ – which replaces $b$ in our main theorems by $b + \ln(KH/\delta)$.

[4]It is essentially the "measurability" condition in Zanette et al. [2021] and "compliance" condition in Jin et al. [2021b].

$\Pi^{soft}(T) = \Pi_1^{soft}(T) \times \cdots \times \Pi_H^{soft}(T)$, where $\Pi_h^{soft}(T) := \{\pi_h(a|s) \propto \exp(\eta \sum_{i=1}^t g_i(s,a)) : t \in [T], g_i \in \mathcal{F}_h, \forall i \in [t], \eta \in [0,1]\}$ for any $T \in \mathbb{N}$. The motivation for the induced policy class $\Pi^{soft}(T)$ is from the soft policy iteration (SPI) update where we incrementally update the policy.

We now discuss a set of assumptions that we impose on the policy and function classes.

**Assumption 2.1** (Approximate realizability). *There exist $\{\xi_h\}_{h\in[H]}$ where $\xi_h \geq 0$ such that,*

$$\sup_{T\in\mathbb{N}, \pi\in\Pi^{soft}(T), (s_h,a_h)\in\text{supp}(d_h^\mu)} \inf_{f\in\mathcal{F}} |f_h(s_h,a_h) - Q_h^\pi(s_h,a_h)| \leq \xi_h, \ \ \forall h \in [H].$$

Assumption 2.1 establishes that $\mathcal{F}$ can realize $Q^\pi$ for any $\pi \in \Pi^{soft}(T)$ up to some error $\xi \in \mathbb{R}^H$ in the supremum norm over the $\mu$-feasible state-action pairs. It strictly generalizes the assumption in Zanette et al. [2021] which restricts $\xi_h = 0$, $\forall h$ (i.e., assume realizability) and the assumption in Xie et al. [2021] which constrains the approximation error under any feasible state-action distribution.

The realizability in *value* functions alone is known to be insufficient for sample-efficient offline RL [Wang et al., 2021]; thus, one needs to impose a stronger assumption for polynomial sample complexity of model-free methods.[5] In this paper, we impose an assumption on the closedness of the Bellman operator.

**Assumption 2.2** (General Restricted Bellman Closedness). *There exists $\nu \in \mathbb{R}^H$ such that*

$$\sup_{T\in\mathbb{N}, f_{h+1}\in\mathcal{F}_{h+1}, \tilde{\pi}\in\Pi^{soft}(T)} \inf_{f_h'\in\mathcal{F}_h} \|f_h' - \mathbb{T}_h^{\tilde{\pi}} f_{h+1}\|_\infty \leq \nu_h, \ \ \forall h \in [H].$$

Assumption 2.2 ensures that the value function space $\mathcal{F}$ and the induced policy class $\Pi^{soft}(T)$ for any $T \in \mathbb{N}$ are closed under the Bellman operator up to some error $\nu \in \mathbb{R}^H$ in the supremum norm. This assumption is a direct generalization of the Linear Restricted Bellman Closedness in Zanette et al. [2020] from a linear function class to a general function class. As remarked by Zanette et al. [2021], the Linear Restricted Bellman Closedness is already strictly more general than the low-rank MDPs [Yang and Wang, 2019, Jin et al., 2020].

## 2.4 Effective sizes of policy and function classes

When the function class and the policy class have finite elements, we use their cardinality $|\mathcal{F}_h|$ and $|\Pi_h^{soft}(T)|$ to measure their sizes [Jiang et al., 2017, Xie et al., 2021]. When they have infinite elements, we use log-covering numbers, defined as

$$d_\mathcal{F}(\epsilon) := \max_{h\in[H]} \ln N(\epsilon; \mathcal{F}_h, \|\cdot\|_\infty), \text{ and } d_\Pi(\epsilon, T) := \max_{h\in[H]} \ln N(\epsilon; \Pi_h^{soft}(T), \|\cdot\|_{1,\infty}),$$

where $\|\pi - \pi'\|_{1,\infty} = \sup_{s\in\mathcal{S}} \int_\mathcal{A} |\pi(a|s) - \pi'(a|s)| d\rho(a)$ for any $\pi, \pi' \in \{\mathcal{S} \to \mathcal{P}(\mathcal{A})\}$ and $N(\epsilon; \mathcal{X}, \|\cdot\|)$ denotes the covering number of a pseudometric space $(\mathcal{X}, \|\cdot\|)$ with metric $\|\cdot\|$ [Zhang, 2023, e.g. Definition 4.1].

We also define a complexity measure that depends on a prior distribution $p_0$ over $\mathcal{F}$ that we employ to favor certain regions of the function space. Our notion, presented in Definition 2, is simply a direct adaptation of a similar notation of Dann et al. [2021] to the actor-critic setting.

**Definition 2.** *For any function $f' \in \mathcal{F}_{h+1}$ and any policy $\tilde{\pi} \in \Pi^{all}$, we define $\mathcal{F}_h^{\tilde{\pi}}(\epsilon; f') := \{f \in \mathcal{F}_h : \|f - \mathbb{T}_h^{\tilde{\pi}} f'\|_\infty \leq \epsilon\}$, for any $\epsilon \geq 0$, and subsequently define*

$$d_0(\epsilon) := \sup_{T\in\mathbb{N}, f\in\mathcal{F}, \tilde{\pi}\in\Pi^{soft}(T)} \sum_{h=1}^H \ln \frac{1}{p_{0,h}(\mathcal{F}_h^{\tilde{\pi}}(\epsilon; f_{h+1}))}, d_0'(\epsilon) := \sup_{T\in\mathbb{N}, \tilde{\pi}\in\Pi^{soft}(T)} \sum_{h=1}^H \ln \frac{1}{p_{0,h}(\mathcal{F}_h^{\tilde{\pi}}(\epsilon; Q_{h+1}^{\tilde{\pi}}))}.$$

The quantity $d_0(\epsilon)$ and $d_0'(\epsilon)$ measures the concentration of the prior $p_0$ over all functions $f \in \mathcal{F}$ that are $\epsilon$-close (element-wise) under $\mathbb{T}^{\tilde{\pi}}$ and $\epsilon$-close (element-wise) to $Q_h^{\tilde{\pi}}$, respectively. If a stronger version of Assumption 2.1 is met, i.e., $Q_h^{\tilde{\pi}} \in \mathcal{F}_h, \forall \tilde{\pi} \in \Pi_h^{all}, h \in [H]$, we have $d_0'(\epsilon) \leq d_0(\epsilon), \forall \epsilon$. For

---

[5] A stronger form of realizability is sufficient for polynomial sample complexity, e.g., realizability for a density ratio w.r.t. the behavior state-action distribution in dual-primal methods [Zhan et al., 2022, Chen and Jiang, 2022, Rashidinejad et al., 2023] or realizability for the underlying MDP in model-based methods [Uehara and Sun, 2022]. Instead, we pursue model-free value-based methods.

the finite function class $\mathcal{F}$ and an uninformative prior $p_{0,h}(f_h) = 1/|\mathcal{F}_h|$, under a stronger version of Assumption 2.2, i.e., $\nu_h = 0, \forall h$, we have $d_0(\epsilon) \leq \sum_{h=1}^{H} \ln |\mathcal{F}_h| = \ln |\mathcal{F}|$. For a parametric model, where each $f_h = f_h^{\theta}$ is represented by a $d$-dimensional parameter $\theta \in \Omega_h^{\theta} \subset \mathbb{R}^d$, a prior over $\Omega^{\theta}$ induces a prior over $\mathcal{F}$. If each $\Omega_h^{\theta}$ is compact, we can generally assume the prior that satisfies $\sup_{\theta} \ln \frac{1}{p_{0,h}(\theta': \|\theta - \theta'\| \leq \epsilon)} \leq d \ln(c_0/\epsilon)$ for some constant $c_0$. If $f_h = f_h^{\theta}$ is Lipschitz in $\theta$, we can assume that $\sup_{\theta} \ln \frac{1}{p_{0,h}(\theta': \|\theta - \theta'\| \leq \epsilon)} \leq c_1 d \ln(c_2/\epsilon)$ for some constants $c_1, c_2$. Overall, we can assume that $d_0(\epsilon) \leq c_1 H d \ln(c_2/\epsilon)$. A similar discussion can be found in Dann et al. [2021].

# 3 Algorithms

Next, we present concrete instances of PS-based, RO-based, and VS-based algorithms. The RO-based and VS-based algorithms presented here are slight refinements of their original versions in Xie et al. [2021]. The PS-based algorithm is novel. All three algorithms resemble the actor-critic style update, inspired by Zanette et al. [2021]. We refer to this generic framework as GOPO (Generic Offline Policy Optimization) presented in Algorithm 1. At each round $t$, a critic estimates the value $\underline{Q}_h^t$ of the actor (i.e., policy $\pi^t$) using the procedure

---

**Algorithm 1** GOPO($\mathcal{D}, \mathcal{F}, \eta, T,$ CriticCompute ):
Generic Offline Policy Optimization Framework

**Input:** Offline data $\mathcal{D}$, function class $\mathcal{F}$, learning rate $\eta > 0$, and iteration number $T$
1: Uniform policy $\pi^1 = \{\pi_h^1\}_{h \in [H]}$
2: **for** $t = 1, \ldots, T$ **do**
3: $\quad \underline{Q}^t =$ CriticCompute $(\pi^t, \mathcal{D}, \mathcal{F}, \ldots)$
4: $\quad \pi_h^{t+1}(a|s) \propto \pi_h^t(a|s) \exp(\eta \underline{Q}_h^t(s, a)), \forall(s, a, h)$
5: **end for**
**Output:** $\hat{\pi} \sim \text{Uniform}(\{\pi^t\}_{t \in [T]})$

---

CriticCompute on Line 3, and the actor improves the policy using a multiplicative weights update [Arora et al., 2012] (Line 4). After $T$ iterations, GOPO returns a policy $\hat{\pi}$ that is sampled uniformly from the set of the obtained policies $\{\pi^t\}_{t \in [T]}$.

To incorporate the pessimism principle, a critic should generate pessimistic estimates of the value of the actor $\pi^t$ in Line 3. This is where the three approaches differ – each invokes a different method to compute the critic. Here, we provide a detailed description of the critic module for each approach. To aid the presentation, we introduce the total temporal difference (TD) loss $\hat{L}_{\tilde{\pi}}$, defined as $\hat{L}_{\tilde{\pi}}(f_h, f_{h+1}) := \sum_{k=1}^{K} l_{\tilde{\pi}}(f_h, f_{h+1}; z_h^k)$, where $z_h := (s_h, a_h, r_h, s_{h+1})$, $z_h^k := (s_h^k, a_h^k, r_h^k, s_{h+1}^k)$, and $l_{\tilde{\pi}}(f_h, f_{h+1}; z_h) := (f_h(s_h, a_h) - r_h - f_{h+1}(s_{h+1}, \tilde{\pi}))^2$.

**Version Space-based Critic (VSC) (Algorithm 2).** Given the actor $\pi^t$, at each step $h \in [H]$, VSC directly maintains a local regression constraint using the offline data: $\hat{L}_{\pi^t}(f_h, f_{h+1}) \leq \inf_{g \in \mathcal{F}} \hat{L}_{\pi^t}(g_h, f_{h+1}) + \beta$, where $\beta$ is a confidence parameter and $\hat{L}_{\pi^t}(\cdot, \cdot)$ is serving as a proxy to the squared Bellman residual at step $h$. By taking the function that minimizes the initial value, VSC then finds the most pessimistic value function $\underline{Q}^t$ from the version space $\mathcal{F}(\beta; \pi^t) \subseteq \mathcal{F}$. In general, the constrained optimization in Line 2 is computationally intractable. Note that a minimax variant of GOPO+VSC first appeared in Xie et al. [2021], where they directly perform an (intractable) search over the policy space, instead of using the multiplicative weights algorithm (Line 4) of Algorithm 1.

**Regularized Optimization-based Critic (ROC) (Algorithm 3).** Instead of solving the global constrained optimization in VSC, ROC solves $\arg\inf_{f \in \mathcal{F}} \{\lambda f_1(s_1, \pi_1^t) + \mathcal{L}_{\pi^t}(f)\}$, where $\lambda$ is a regularization parameter and $\mathcal{L}_{\pi^t}(f)$, defined in Line 1 of Algorithm 3. Note that in ROC, pessimism is implicitly encouraged through the regularization term $\lambda f_1(s_1, \pi^t)$. We remark that, unlike VSC, ROC admits tractable approximations that use adversarial training and work competitively in practice [Cheng et al., 2022]. Note that a discounted variant of GOPO-ROC first appears in [Xie et al., 2021] in discounted MDPs.

---

**Algorithm 2** VSC($\mathcal{D}, \mathcal{F}, \pi^t, \beta$): Version Space-based Critic

1: $\mathcal{F}(\beta; \pi^t) := \{f \in \mathcal{F} : \hat{L}_{\pi^t}(f_h, f_{h+1}) \leq \inf_{g \in \mathcal{F}} \hat{L}_{\pi^t}(g_h, f_{h+1}) + \beta, \forall h \in [H]\}$
2: $\underline{Q}^t \in \arg\min_{f \in \mathcal{F}(\beta; \pi^t)} f_1(s_1, \pi^t)$
**Output:** $\underline{Q}^t$

---

**Algorithm 3** ROC($\mathcal{D}, \mathcal{F}, \pi^t, \lambda$): Regularized Optimization-based Critic

1: $\mathcal{L}_{\pi^t}(f) := \sum_{h=1}^{H} \hat{L}_{\pi^t}(f_h, f_{h+1}) - \inf_{g \in \mathcal{F}} \sum_{h=1}^{H} \hat{L}_{\pi^t}(g_h, f_{h+1})$
2: $\underline{Q}^t \leftarrow \arg\inf_{f \in \mathcal{F}} \{\lambda f_1(s_1, \pi^t) + \mathcal{L}_{\pi^t}(f)\}$
**Output:** $\underline{Q}^t$

---

**Posterior sampling-based critic (PSC) in Algorithm 4.** Instead of solving a regularized minimax optimization, PSC samples the value function $\underline{Q}_h^t$ from the data posterior $\hat{\pi}(f|\mathcal{D}, \pi^t) \propto \tilde{p}_0(f) \cdot p(\mathcal{D}|f, \pi^t)$, where $\tilde{p}_0(f)$ is the prior over $\mathcal{F}$ and $p(\mathcal{D}|f, \pi^t)$ is the likelihood function of the offline data $\mathcal{D}$. To formulate the likelihood function $p(\mathcal{D}|f, \pi^t)$, we make use of the squared TD error $\hat{L}_{\pi^t}(\cdot, \cdot)$ and normalization method in [Dann et al., 2021] to construct an unbiased proxy of the squared Bellman errors. In particular, $p(\mathcal{D}|f, \pi^t) = \prod_{h \in [H]} \frac{\exp(-\gamma \hat{L}_{\pi^t}(f_h, f_{h+1}))}{\mathbb{E}_{f_h' \sim p_{0,h}} \exp(-\gamma \hat{L}_{\pi^t}(f_h', f_{h+1}))}$, where $\gamma$ is a learning rate and $p_0$ is an (unregularized) prior over $\mathcal{F}$. A value function sampled from the posterior with this likelihood function is encouraged to have small squared TD errors. The key ingredient in our algorithmic design is the "pessimistic" prior $\tilde{p}_0(f) = \exp(-\lambda f_1(s_1, \pi_1))p_0(f)$ where we add a new regularization term $\exp(-\lambda f_1(s_1, \pi_1))$, with $\lambda$ being a regularization parameter – which is inspired by the optimistic prior in the online setting [Zhang, 2022, Dann et al., 2021]. This pessimistic prior encourages the value function sampled from the posterior to have a small value in the initial state, implicitly enforcing pessimism. We remark that PSC requires a sampling oracle and expectation oracle (to compute the normalization term in the posterior distribution), which could be amenable to tractable approximations, including replacing expectation oracle with a sampling oracle [Agarwal and Zhang, 2022] while the sampling oracle can be implemented via first-order sampling methods [Welling and Teh, 2011] or ensemble methods [Osband et al., 2016].

---

**Algorithm 4** $\mathrm{PSC}(\mathcal{D}, \mathcal{F}, \pi^t, \lambda, \gamma, p_0)$: Posterior Sampling-based Critic

---

1: $\underline{Q}^t \sim \hat{p}(f|\mathcal{D}, \pi^t) \propto \exp\left(-\lambda f_1(s_1, \pi^t)\right) p_0(f) \prod_{h \in [H]} \frac{\exp\left(-\gamma \hat{L}_{\pi^t}(f_h, f_{h+1})\right)}{\mathbb{E}_{f_h' \sim p_{0,h}} \exp\left(-\gamma \hat{L}_{\pi^t}(f_h', f_{h+1})\right)}$

**Output:** $\underline{Q}^t$

---

## 4 Main Results

In this section, we shall present the upper bounds of the sub-optimality of the policies executed by GOPO-VSC, GOPO-ROC, and GOPO-PSC. Our upper bounds are expressed in terms of a new notion of data diversity.

### 4.1 Data diversity

We now introduce the key notion of data diversity for offline RL. Since the offline learner does not have direct access to the trajectory of a comparator policy $\pi \in \Pi^{all}$, they can only observe partial information about the goodness of $\pi$ channeled through the "transferability" with the behavior policy $\mu$. The transferability from $\mu$ to $\pi$ depends on how *diverse* the offline data induced by $\mu$ can be in supporting the extrapolation to $\pi$. Many prior works require uniform diversity where $\mu$ covers all feasible scenarios of all comparator policies $\pi$. The data diversity can be essentially captured by how well the Bellman error under the state-action distribution induced by $\mu$ can predict the counterpart quantity under the state-action distribution induced by $\pi$. Our notion of data diversity, which is inspired by the notion of task diversity in transfer learning literature [Tripuraneni et al., 2020, Watkins et al., 2023], essentially encodes the ratio of some proxies of expected Bellman errors induced by $\mu$ and $\pi$, and is defined as follows.

**Definition 3.** *For any comparator policy $\pi \in \Pi^{all}$, we measure the data diversity of the behavior policy $\mu$ with respect to a target policy $\pi$ by*

$$\mathcal{C}(\pi; \epsilon) := \max_{h \in [H]} \chi_{(\mathcal{F}_h - \mathcal{F}_h)}(\epsilon; d_h^\pi, d_h^\mu), \forall \epsilon \geq 0, \tag{2}$$

*where $\mathcal{F}_h - \mathcal{F}_h$ is the Minkowski difference between the function class $\mathcal{F}_h$ and itself, i.e., $\mathcal{F}_h - \mathcal{F}_h := \{f_h - f_h' : f_h, f_h' \in \mathcal{F}\}$, and $\chi_{\mathcal{Q}}(\epsilon; q, p)$ is the discrepancy between distributions $q$ and $p$ under the witness of function class $\mathcal{Q}$ defined as*

$$\chi_{\mathcal{Q}}(\epsilon; q, p) = \inf\left\{C \geq 0 : (\mathbb{E}_q[g])^2 \leq C \cdot \mathbb{E}_p[g^2] + \epsilon, \forall g \in \mathcal{Q}\right\}$$

*with $\mathcal{Q}$ being a function class and $p$ and $q$ being two distributions over the same domain.*

Up to a small additive error $\epsilon$, a finite $\mathcal{C}(\pi; \epsilon)$ ensures that a proxy of the Bellman error under the $\pi$-induced state-action distribution is controlled by that under the $\mu$-induced state-action distribution.

Despite the abstraction in the definition of this data diversity, it is *always* upper bounded by the single-policy concentrability coefficient [Liu et al., 2019, Rashidinejad et al., 2021] and the relative condition number [Agarwal et al., 2021, Uehara et al., 2022b, Uehara and Sun, 2022] that are both commonly used in many prior offline RL works. We further discuss our data diversity measure in more detail in Section 4.2.

## 4.2 Offline learning guarantees

We now utilize data diversity to give learning guarantees of the considered algorithms for extrapolation to an arbitrary comparator policy $\pi \in \Pi^{all}$. To aid the representation, in all of the following theorems we are about to present, we shall set $\eta = \sqrt{\frac{\ln \mathrm{Vol}(\mathcal{A})}{4(e-2)b^2 T}}$ in Algorithm 1, where $\mathrm{Vol}(\mathcal{A})$ is the volume of the action set $\mathcal{A}$ (e.g., $\mathrm{Vol}(\mathcal{A}) = |\mathcal{A}|$ for finite $\mathcal{A}$), and define, for simplicity, the misspecification errors $\zeta_{msp} := K \sum_{h=1}^{H} \left( \nu_h^2 + b\nu_h \right)$, $\tilde{\zeta}_{msp} := \zeta_{msp} + bK \sum_{h=1}^{H} \xi_h$, $\bar{\nu} := \sum_{h=1}^{H} \nu_h$, the optimization error $\zeta_{opt} := Hb\sqrt{T^{-1} \ln \mathrm{Vol}(\mathcal{A})}$, and the complexity measures $\tilde{d}_{opt}(\epsilon, T) := \max\{d_{\mathcal{F}}(\epsilon), d_{\Pi}(\epsilon, T)\}$, and $\tilde{d}_{ps}(\epsilon, T) := \max\{d_{\mathcal{F}}(\epsilon), d_{\Pi}(\epsilon, T), \frac{d_0(\epsilon)}{\gamma Hb^2}, \frac{d_0'(\epsilon)}{\gamma Hb^2}\}$.

---

**Theorem 1** (Guarantees for GOPO-VSC). *Let $\hat{\pi}^{vs}$ be the output of Algorithm 1 invoked with $\boxed{CriticCompute}$ being VSC($\mathcal{D}, \mathcal{F}, \pi^t, \beta$) (Algorithm 2) with $\beta = \mathcal{O}(Hb^2 \max\{\tilde{d}_{opt}(\epsilon, T), \ln(H/\delta)\} + b^2 K\epsilon + bK \max_{h\in[H]} \xi_h)$. Fix any $\delta \in (0, 1]$. Under Assumption 2.1-2.2, with probability at least $1 - 2\delta$ (over the randomness of the offline data), for any $\epsilon, \epsilon_c, \lambda > 0$, and any $\pi \in \Pi^{all}$, we have*

$$\mathbb{E}\left[\mathrm{SubOpt}_\pi(\hat{\pi}^{vs}) | \mathcal{D}\right] \lesssim \frac{Hb^2 \cdot \max\{\tilde{d}_{opt}(\epsilon, T), \ln(H/\delta)\} + b^2 KH\epsilon + \tilde{\zeta}_{msp}}{\lambda} + \frac{\lambda H \cdot \mathcal{C}(\pi; \epsilon_c)}{2K}$$
$$+ H\epsilon_c + \xi_1 + \bar{\nu} + \zeta_{opt}.$$

---

**Theorem 2** (Guarantees for GOPO-ROC). *Let $\hat{\pi}^{ro}$ be the output of Algorithm 1 invoked with $\boxed{CriticCompute}$ being ROC($\mathcal{D}, \mathcal{F}, \pi^t, \lambda$) (Algorithm 3). Fix any $\delta \in (0, 1]$. Under Assumption 2.1-2.2, with probability at least $1 - 2\delta$ (over the randomness of the offline data), for any $\epsilon, \epsilon_c, \lambda > 0$, and any $\pi \in \Pi^{all}$, we have*

$$\mathbb{E}\left[\mathrm{SubOpt}_\pi(\hat{\pi}^{ro}) | \mathcal{D}\right] \lesssim \frac{Hb^2 \cdot \max\{\tilde{d}_{opt}(\epsilon, T), \ln \frac{H}{\delta}\} + b^2 KH\epsilon + \tilde{\zeta}_{msp}}{\lambda} + \frac{\lambda H \cdot \mathcal{C}(\pi; \epsilon_c)}{2K}$$
$$+ H\epsilon_c + \xi_1 + \bar{\nu} + \zeta_{opt}.$$

---

**Theorem 3** (Guarantees for GOPO-PSC). *Let $\hat{\pi}^{ps}$ be the output of Algorithm 1 invoked with $\boxed{CriticCompute}$ being PSC($\mathcal{D}, \mathcal{F}, \pi^t, \lambda, \gamma, p_0$) (Algorithm 4). Under Assumption 2.2, for any $\gamma \in [0, \frac{1}{144(e-2)b^2}]$, and $\epsilon, \epsilon_c, \delta, \lambda > 0$, and any $\pi \in \Pi^{all}$, we have*

$$\mathbb{E}\left[\mathrm{SubOpt}_\pi(\hat{\pi}^{ps})\right] \lesssim \frac{\gamma Hb^2 \cdot \max\{\tilde{d}_{ps}(\epsilon, T), \ln \frac{\ln Kb^2}{\delta}\} + \gamma b^2 KH \cdot \max\{\epsilon, \delta\} + \gamma\zeta_{msp}}{\lambda}$$
$$+ \frac{\lambda H \cdot \mathcal{C}(\pi; \epsilon_c)}{K\gamma} + H\epsilon_c + \epsilon + \bar{\nu} + \zeta_{opt}.$$

---

Our results provide a family of upper bounds on the sub-optimality of each of $\{\hat{\pi}^{vs}, \hat{\pi}^{ro}, \hat{\pi}^{ps}\}$, indexed by our choices of the comparator $\pi$ with the data diversity $\mathcal{C}(\pi; \epsilon_c)$, additive (extrapolation) error $\epsilon_c$, the discretization level $\epsilon$ in log-covering numbers, the "failure" probability $\delta$, and other algorithm-dependent parameters ($\lambda$ for $\hat{\pi}^{ro}$ and $(\lambda, \gamma)$ for $\hat{\pi}^{ps}$). Note that the optimization error $\zeta_{opt}$ captures the error rate of the actor and can be made arbitrarily small with large iteration number $T$ whereas $\zeta_{msp}, \tilde{\zeta}_{msp}, \bar{\nu}$, and $\xi_1$ are simply misspecification errors aggregated over all stages. Also note that our bound does not scale with the complexity of the comparator policy class $\Pi^{all}$. We next highlight the key characteristics of our main results in comparison with existing work.

**(I) Tight characterization of data diversity.** Our bounds in all the above theorems are expressed in terms of $\mathcal{C}(\pi; \epsilon_c)$. Several remarks are in order. First, $\mathcal{C}(\pi; \epsilon_c)$ is a *non-increasing* function of $\epsilon_c$; thus $\mathcal{C}(\pi; \epsilon_c)$ is always smaller or at least equal to $\mathcal{C}(\pi; 0)$. In fact, it is possible that $\mathcal{C}(\pi; 0) = \infty$ yet $\mathcal{C}(\pi; \epsilon_c) < \infty$ for some $\epsilon > 0$. For instance, if there exists $g \in \mathcal{Q}$ such that $g(x) = 0, \forall x \in \text{supp}(p)$ and $\{x : g(x) \neq 0\}$ has a positive measure under $q$, then $\chi_{\mathcal{Q}}(0; q, p) = \infty$ while $\chi_{\mathcal{Q}}(\sup_{g \in \mathcal{Q}} \mathbb{E}_q[g]^2; q, p) = 0$. Second, $\mathcal{C}(\pi; 0)$ is always bounded from above by (often substantially smaller than) the *single-policy concentrability coefficient* between the $\pi$-induced and $\mu$-induced state-action distribution [Liu et al., 2019, Rashidinejad et al., 2021], which been used extensively in recent offline RL works [Yin and Wang, 2021, Nguyen-Tang et al., 2022a, 2023, Jin et al., 2022, Zhan et al., 2022, Nguyen-Tang and Arora, 2023, Zhao et al., 2023]. This is essentially because $d^\pi$ can cover the region that is not covered by $d^\mu$ but still the integration of functions in $\mathcal{F}_h - \mathcal{F}_h$ over two distributions are close to each other. Third, $\mathcal{C}(\pi; 0)$ is always upper bounded by the *relative condition numbers* used in [Agarwal et al., 2021,

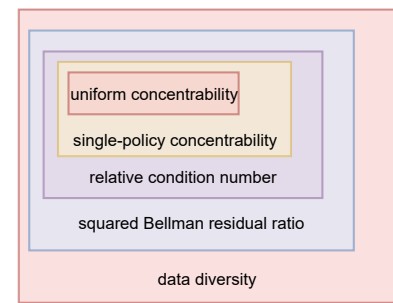

Figure 1: The relations of sample-efficient offline RL classes under different data coverage measures. Given the same MDP and a target policy (e.g., an optimal policy of the MDP), each data coverage measure induces a corresponding set of behavior policies (represented by the rectangle labelled by the data coverage measure) from which the target policy is offline-learnable.

Uehara et al., 2022b, Uehara and Sun, 2022]. Our data diversity at $\epsilon = 0$ is similar to the notion of distribution mismatch in Duan et al. [2020], Ji et al. [2022], though our notion is motivated by transfer learning and discovered naturally from our decoupling argument. Our data diversity measure at $\epsilon = 0$ is smaller than the Bellman residual ratio measure used in Xie et al. [2021] (follows using Jensen's inequality). Finally, the concurrent work of Di et al. [2023] proposed a notion of $D^2$-divergence to capture the data disparity of a data point to the offline data. Our data diversity is in general less restricted as we only need to ensure the diversity between two data distributions (of the target policy and the behavior policy), not necessarily between each of their individual data points.

In summary, $\mathcal{C}(\pi; \epsilon_c)$, to the best of our knowledge, provides the tightest characterization of distribution mismatch compared to the prior data coverage notions. We sketch the relationships of the discussed notions in Figure 1, where with our data diversity notion, we show that the scenarios for the offline data in which offline RL is learnable are enlarged compared to the picture depicted by the prior data coverage notions.

**(II) Competing with all comparator policies simultaneously.** Similar to some recent results in offline RL, our offline RL algorithms compete with all comparator policies that are supported by offline data in some sense. In particular, the choice of the comparator $\pi$ provides the flexibility to *automatically* compete with the best policy within a certain diversity level of our choice. For instance, if we want to limit the level $\mathcal{C}(\pi; \epsilon_c) \leq C$ for some arbitrary $C > 0$, our bound automatically competes with $\pi = \arg\max_{\pi \in \Pi^{all}} \{V_1^\pi(s) : \mathcal{C}(\pi; \epsilon_c) \leq C\}$. This is immensely meaningful since the offline data might not support extrapolation to an optimal policy in practice.

**(III) State-of-the-art bounds for standard assumptions.** We compare our bounds with other recent guarantees of similar assumptions.[6] To ease comparison, we assume for simplicity, that there is no misspecification, i.e., $\nu_h = \xi_h = 0, \forall h \in [H]$, and $T \geq K \ln \text{Vol}(\mathcal{A})$, and we minimize the bounds in Theorem 2 and Theorem 3 with respect to $\lambda$. The three theorems can then be simplified into a unified result presented in Proposition 1.

**Proposition 1** (A unified guarantee for VS, RO and PS). *Under Assumption 2.1-2.2 with no misspecification, i.e., $\nu_h = \xi_h = 0, \forall h \in [H]$, $\forall \hat{\pi} \in \{\hat{\pi}^{vs}, \hat{\pi}^{ro}, \hat{\pi}^{ps}\}$, $\mathbb{E}[\text{SubOpt}_\pi(\hat{\pi})] = \tilde{\mathcal{O}}(Hb\sqrt{\tilde{d}(1/K, T) \cdot \mathcal{C}(\pi; 1/\sqrt{K})/K} + \xi_{opt})$, where $\tilde{d}(1/K, T) = \tilde{d}_{opt}(1/K, T)$ if $\hat{\pi} \in \{\hat{\pi}^{vs}, \hat{\pi}^{ro}\}$ and $\tilde{d}(1/K, T) = \tilde{d}_{ps}(1/K, T)$ if $\hat{\pi} = \hat{\pi}^{ps}$. In addition,*

---

[6]Recent primal-dual methods achieve favorable guarantees for offline RL. However, these guarantees are not directly comparable to the guarantees of our value-based methods due to a different set of assumptions. Nonetheless, we make a detailed discussion in Section A.2.

- If $\mathcal{F}_h$ and $\Pi_h^{soft}(T)$ *have finite elements for all* $h \in [H]$, $\tilde{d}(1/K, T) = \mathcal{O}(\max_{h \in [H]} \max\{\ln|\mathcal{F}_h|, \ln|\Pi_h^{soft}(T)|\})$;

- *If* $\mathcal{F}_h = \{(s,a) \mapsto \langle \phi_h(s,a), w \rangle : \|w\|_2 \le b\}$ *is a linear model, where* $\phi_h : \mathcal{S} \times \mathcal{A} \to \mathbb{R}^d$ *is a known feature map and w.l.o.g.* $\max_h \|\phi_h\|_\infty \le 1$, $\tilde{d}(1/K, T) = \mathcal{O}(d \log(1 + KTb)), \forall T$.

Proposition 1 essentially asserts that VS-based, RO-based, and PS-based algorithms obtain comparable guarantees for offline RL in the realizable case. We now compare our results to related work in various instantiation of function classes.

**Compared with Xie et al. [2021] when the function class is finite.** In this case, the analysis of the VS-based algorithms and RO-based algorithms of Xie et al. [2021] give the bounds that in our setting can be translated[7] into: $Hb\sqrt{\max_h \ln(|\mathcal{F}_h||\Pi_h^{all}|) \cdot C_2(\pi)/K}$ and $Hb\sqrt{C_2(\pi)} \sqrt[3]{\max_h \ln(|\mathcal{F}_h||\Pi_h^{soft}(T)|)/K} + Hb/\sqrt{T}$, respectively, where $C_2(\pi) := \max_{h \in [H], \tilde{\pi} \in \Pi^{all}, f \in \mathcal{F}} \frac{\|f_h - \mathbb{T}_h^{\tilde{\pi}} f_{h+1}\|_{2,\pi}^2}{\|f_h - \mathbb{T}_h^{\tilde{\pi}} f_{h+1}\|_{2,\mu}^2}$. Instead, our bounds for both the VS-based and RO-based algorithms are $Hb\sqrt{\max_h \ln(|\mathcal{F}_h||\Pi_h^{soft}(T)|) \cdot \mathcal{C}(\pi; 1/\sqrt{K})/K} + Hb/\sqrt{T}$. We improve upon the results of Xie et al. [2021] on several fronts. First, our diversity measures $\mathcal{C}(\pi; 1/K)$ is always smaller than their measure $C_2(\pi)$, since $\mathcal{C}(\pi; 1/\sqrt{K}) \le \mathcal{C}(\pi; 0) \le C_2(\pi)$. Second, for the VS-based algorithm, $\Pi^{soft}(T) \subset \Pi^{all}, \forall T$, our bound is always tighter. In fact, $|\Pi^{all}|$ is arbitrarily large that bounds depending on this quantity is vacuous. Third, for the RO-based algorithm, the rates in terms of $K$ in the bound of Xie et al. [2021] are slower than that in our bound. Specifically, if $\Pi_h^{soft}(T) = \tilde{\mathcal{O}}_T(1)$, then these rates are $K^{-1/3}$ vs $K^{-1/2}$ (with an optimal choice of $T = K$ for both bounds). If we consider the worst case that $\Pi_h^{soft}(T) = \mathcal{O}(T \log|\mathcal{F}_h|)$, then these rates are $K^{-1/5}$ vs $K^{-1/4}$ (with an optimal choice of $T = K^{2/5}$ and $T = \sqrt{K}$ in the respective bounds). Finally, our results hold under the general adaptively collected data rather than their independent episode setting. We summarize the bounds in the finite function class cases in Table 1, and give comparisons for the linear model cases in Table 2.

**Compared with LCB-based algorithms.** When $\mathcal{F}_h$ is a $d$-dimensional linear model with feature maps $\{\phi_h\}_{h \in [H]}$, our bounds reduce into $Hb\sqrt{d \cdot K^{-1} \cdot \mathcal{C}(\pi; 1/\sqrt{K})}$ (Proposition 1), which matches the order of (and potentially tighter than) the bound in Zanette et al. [2021], since $\mathcal{C}(\pi; 1/\sqrt{K})$ is always smaller (or at least equal to) than the relative condition number $\max_h \sup_{x \in \mathbb{R}^d} \frac{x^T \mathbb{E}_\pi[\phi_h(s_h, a_h)\phi_h(s_h, a_h)^T]x}{x^T \mathbb{E}_\mu[\phi_h(s_h, a_h)\phi_h(s_h, a_h)^T]x}$. Compared with the bound of LCB-based algorithms in Jin et al. [2021b], we improve a factor $\sqrt{d}$ and holds under the more general Assumption 2.2 which includes low-rank MDPs. In a more refined analysis [Xiong et al., 2023], the LCB-based algorithm obtains the same dependence on $d$ for low-rank MDPs as our guarantees. However, this improvement relies on a uniform coverage assumption, i.e., $\min_{h \in [H]} \lambda_{\min}\left(\mathbb{E}_{(s_h,a_h) \sim d_h^\mu}[\phi_h(s_h, a_h)\phi_h(s_h, a_h)^T]\right) > 0$, which we do not require. Di et al. [2023] generalize the results of Xiong et al. [2023] from linear MDPs to MDPs with general function approximation. However, they still rely on a uniform coverage assumption. Finally note that, for VS-based and RO-based algorithms, we provide high-probability bounds for a smoothing version of $\hat{\pi}$ over the randomization of the algorithms, not for $\hat{\pi}$ itself.

**Compared with model-based PS.** Uehara and Sun [2022] consider model-based PS for offline RL, where they obtain the *Bayesian* sub-optimality bound of $H^2\sqrt{C^{\text{Bayes}} \cdot \ln|\mathcal{M}|/K}$ where $C^{\text{Bayes}}$ is the Bayesian version of a relative condition number and $\mathcal{M}$ is a finite model class. Two key distinctions are that our method in Algorithm 4 is model-free, and our achieved bound is in the frequentist (i.e., worst-case) nature, which is a stronger result than the Bayesian bound of the same order.

## 5 Conclusion

We contributed to the understanding of sample-efficient offline RL in the context of (value) function approximation. We proposed a notion of data diversity that generalizes the previous data coverage measures and importantly expands the class of sample-efficient offline RL. We studied three different algorithms: VS, RO, and PS, where the PS-based algorithm is our novel proposal. We showed that VS, RO, and PS all have same-order guarantees under standard assumptions.

---

[7]Xie et al. [2021] consider discounted MDP and a *restricted* policy class for the comparator class.

## Acknowledgements

This research was supported, in part, by DARPA GARD award HR00112020004, NSF CAREER award IIS-1943251, funding from the Institute for Assured Autonomy (IAA) at JHU, and the Spring'22 workshop on "Learning and Games" at the Simons Institute for the Theory of Computing.

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

# Contents

# Appendices

## Appendix A   Extended Discussion

We also extend the discussion of our data diversity in comparison to the existing distribution mismatch measures in the special case where each $\mathcal{F}_h$ is a linear function in a known feature map.

### A.1   Linear function classes

In this section, we consider the linear model cases, where there are known feature maps $\phi_h : \mathcal{X} \times \mathcal{A} \to \mathbb{R}^d$ and w.l.o.g. $\max_{h \in [H]} \|\phi(\cdot, \cdot)\|_\infty \leq 1$, such that $\mathcal{F}_h = \{\phi_h(\cdot, \cdot)^T w : w \in \mathbb{R}^d, \|w\|_2^2 \leq b\}$. Recall that, in this case, e.g., it follows from [Zanette et al., 2021, Lemma 6], that we have

$$\log N(\epsilon; \mathcal{F}_h, \|\cdot\|_\infty) \leq d \log(1 + \frac{2}{\epsilon}),$$

$$\log N(\epsilon; \Pi_h^{soft}(T), \|\cdot\|_\infty) \leq d \log(1 + \frac{16bT}{\epsilon}).$$

Thus, our bounds from Proposition 1 can simplified as

$$\tilde{\mathcal{O}}(Hb\sqrt{d\log(1 + 16bKT) \cdot \mathcal{C}(\pi; 1/\sqrt{K})/K} + Hb\sqrt{T^{-1}\ln \operatorname{Vol}(\mathcal{A})})$$

$$= \tilde{\mathcal{O}}\left(Hb\sqrt{\frac{\max\{d\mathcal{C}(\pi; 1/\sqrt{K}), \ln \operatorname{Vol}(\mathcal{A})\}}{K}}\right)$$

where we choose $T = K$. To simplify the comparison, we assume that $d\mathcal{C}(\pi; 1/\sqrt{K}) \geq \ln \operatorname{Vol}(\mathcal{A})$.

Let us now compute various notions of data coverage in this linear model case. s We first need to define the following quantities (various forms of covariance matrices).

$$\Sigma_h := \lambda I + \sum_{k=1}^{K} \phi_h(s_h^k, a_h^k)\phi_h(s_h^k, a_h^k)^T,$$

$$\Lambda_h := \lambda I + \sum_{k=1}^{K} \phi_h(s_h^k, a_h^k)\phi_h(s_h^k, a_h^k)^T / [\mathbb{V}_h V_{h+1}^\pi](s_h^k, a_h^k),$$

$$\bar{\phi}_h^\pi := \mathbb{E}_\pi[\phi_h(s_h, a_h)],$$

$$\bar{\Sigma}_h := \mathbb{E}_\mu\left[\phi(s_h, a_h)\phi(s_h, a_h)^T\right].$$

We define the following distribution mismatch quantities, which were used in the literature.

$$C_{pevi}(\pi) := \max_{h \in [H]} \left(\mathbb{E}_\pi\left[\|\phi_h(s_h, a_h)\|_{\Sigma_h^{-1}}\right]\right)^2,$$

$$C_{pevi-adv}(\pi) := \max_{h \in [H]} \left(\mathbb{E}_\pi\left[\|\phi_h(s_h, a_h)\|_{\Lambda_h^{-1}}\right]\right)^2,$$

$$C_{pacle}(\pi) := \max_{h \in [H]} \|\bar{\phi}_h^\pi\|_{\Sigma_h^{-1}}^2,$$

$$C_{bcp}(\pi) := \max_{h \in [H]} \left(\mathbb{E}_\pi\left[\|\phi_h(s_h, a_h)\|_{\bar{\Sigma}_h}\right]\right)^2.$$

The sub-optimality bounds of various methods are summarized in Table 2. For comparing our data diversity measure with different notions of distribution mismatch, we have

$$C_{pevi}(\pi) \geq C_{pacle}(\pi) \approx \mathcal{C}(\pi; 0)/K \leq C_{bcp}(\pi)/K.$$

where the "$\approx$" denotes that the involved terms scale in the same order and can be implied by Fredman's matrix inequality (see [Duan et al., 2020, Lemma B.5]) (under additional conditions). Note that $\mathcal{C}(\pi; 1/\sqrt{K}) \leq \mathcal{C}(\pi; 0)$, thus our data diversity is the tightest quantity among all that are considered.

| Algorithm | Sub-optimality bound |
|:---:|:---:|
| PEVI [Jin et al., 2021b] | $Hb\sqrt{C_{pevi}(\pi) \cdot d}$ |
| PEVI-ADV+ [Xiong et al., 2023] | $H\sqrt{C_{pevi-avi}(\pi) \cdot d}$ |
| PACLE [Zanette et al., 2021] | $Hb\sqrt{C_{pacle}(\pi) \cdot d}$ |
| VC [Xie et al., 2021, Section 3] | $Hb\sqrt{C_{bcp}(\pi) \cdot d/K}$ |
| RO [Xie et al., 2021, Section 4] | $Hb\sqrt{C_{bcp}(\pi)}\sqrt[3]{d/K}$ |
| Ours (VS, RO, PS) | $Hb\sqrt{\mathcal{C}(\pi; 1/\sqrt{K}) \cdot d/K}$ |

Table 2: Sub-optimality bounds when the function class $\mathcal{F}_h$ is linear in $\phi_h : \mathcal{S} \times \mathcal{A} \to \mathbb{R}^d$.

Note that the data coverage measure in Xiong et al. [2023], roughly speaking, can be bounded as follows:

$$C_{pevi-adv}(\pi) \le b^2 C_{pevi}(\pi),$$

where we use the inequality $[\mathbb{V}_h V_{h+1}^\pi](s_h^k, a_h^k) \le b^2$. Thus the bound of Xiong et al. [2023] in general has a tighter dependence on $b$ (which implicitly depends on $H$) than all the bounds of all other works considered in Table 2, due to that Xiong et al. [2023] incorporated the variance information into the estimation via the variance-weighted value iteration algorithm. However, obtaining this improved bound in Xiong et al. [2023] relies on a uniform coverage assumption which we do not require.

## A.2   Comparison with primal-dual methods for offline RL

As opposed to the value-based methods we considered in our paper, an important alternative approach to offline RL is the primal-dual methods [Zhan et al., 2022, Chen and Jiang, 2022, Rashidinejad et al., 2023, Gabbianelli et al., 2023, Ozdaglar et al., 2023]. However, the guarantees of primal-dual methods use a different set of assumptions than the value-based methods we consider (the former assumes realizability for the ratio between the state-action occupancy density of the target policy and the state-action occupancy density of the behavior policy, except for Gabbianelli et al. [2023] where this realizability assumption is implicitly encoded under a stronger assumption of linear MDP). This makes the results presented in our paper and the results in the primal-dual methods not directly comparable.

Since the work of Gabbianelli et al. [2023] considers linear MDPs, it is more comparable (than the other primal-dual methods we mentioned) to the instantiating of our results to the linear function class. Gabbianelli et al. [2023] consider primal-dual methods for offline RL in both infinite-horizon discounted MDP and average-reward MDP. Our analysis framework for the regularized optimization method in the episodic MDP should work for the infinite-horizon discounted MDP as well, where the regularized optimization achieves the optimal sample complexity of $\mathcal{O}(\epsilon^{-2})$ while the sample complexity in Gabbianelli et al. [2023] in this setting is $\mathcal{O}(\epsilon^{-4})$. However, Gabbianelli et al. [2023] offers a better computational complexity ($\mathcal{O}(K)$ vs $\mathcal{O}(K^{7/5})$) and also works in the average-reward MDP setting which is beyond the episodic MDP setting considered in our work; though our bounds hold for general function approximation that is beyond the strong assumption of linear MDPs.

The concurrent work of Zhu et al. [2023] combined the actor-critic framework with marginalized importance sampling (MIS) for an RO-based algorithm, which also improves the sub-optimal rate of order $1/K^{1/3}$ by Xie et al. [2021], Cheng et al. [2022] to the optimal rate of order $1/\sqrt{K}$. Instead, we obtain the optimal rate of order $1/\sqrt{K}$ with a refined analysis for a standard RO-based algorithm. That is, unlike Zhu et al. [2023], we do not use MIS; consequently, we do not require the realizability assumption for the ratio between the state-action occupancy density of the target policy and that of the behavior policy.

# Appendix B  Preparation

We now get into more involved parts where we present the proof process and the technical results for obtaining Theorem 1, Theorem 2, and Theorem 3. In order to prove our main results in Section 4, we shall need some old tools and develop some new useful tools. For convenience, we start out with both old and new notations of quantities summarized in Table 3 that we are going to use frequently in our proofs.

| Name | Notation | Expression |
|------|----------|------------|
| transition sample | $z_h^k$ | $(s_h^k, a_h^k, r_h^k, a_{h+1}^k)$ |
| transition sample | $z_h$ | $(s_h, a_h, r_h, s_{h+1})$ |
| Bellman error | $\mathcal{E}_h^{\tilde{\pi}}(f_h, f_{h+1})(s_h, a_h)$ | $(\mathbb{T}_h^{\tilde{\pi}} f_{h+1} - f_h)(s_h, a_h)$ |
| TD loss function | $l_{\tilde{\pi}}(f_h, f_{h+1}; z_h)$ | $(f_h(s_h, a_h) - r_h - f_{h+1}(s_{h+1}, \tilde{\pi}))^2$ |
| empirical squared Bellman error (SBE) | $\hat{L}_{\tilde{\pi}}(f_h, f_{h+1})$ | $\sum_{k=1}^K l_{\tilde{\pi}}(f_h, f_{h+1}; z_h^k)$ |
| empirical bias-adjusted SBE | $\mathcal{L}_{\tilde{\pi}}(f)$ | $\sum_{h=1}^H \hat{L}_{\tilde{\pi}}(f_h, f_{h+1}) - \inf_{g \in \mathcal{F}} \sum_{h=1}^H \hat{L}_{\tilde{\pi}}(g_h, f_{h+1})$ |
| excess TD loss | $\Delta L_{\tilde{\pi}}(f_h, f_{h+1}; z_h)$ | $l_{\tilde{\pi}}(f_h, f_{h+1}; z_h) - l_{\tilde{\pi}}(\mathbb{T}_h^{\tilde{\pi}} f_{h+1}, f_{h+1}; z_h)$ |
| – | $\mathbb{E}_\mu[\cdot]$ | $\frac{1}{K} \sum_{k=1}^K \mathbb{E}_{\mu^k}[\cdot]$ |
| – | $\mathbb{E}_k[\cdot](:= \mathbb{E}_{\mu^k}[\cdot])$ | $\mathbb{E}\left[\cdot \middle| \{z_h^i\}_{h \in [H]}^{i \in [k-1]}\right]$ |

Table 3: A summary of notations and quantities of interest.

The quantity $l_{\tilde{\pi}}(f_h, f_{h+1}; z_h)$ can be viewed as *a temporal difference (TD) loss function* defined on data point $z_h$ conditioned on each $f_{h+1}$ and $\tilde{\pi}$. The quantity $\mathbb{T}_h^{\tilde{\pi}} f_{h+1}$ can be viewed as *the Bellman regression function*, where, conditioned on each $f_{h+1}$ and $\tilde{\pi}$, for any $(s_h, a_h)$, we have

$$\mathbb{T}_h^{\tilde{\pi}} f_{h+1}(s_h, a_h) = \mathbb{E}_{r_h, s_{h+1}|s_h, a_h}[r_h + f_{h+1}(s_{h+1}, \tilde{\pi})] = \arg\inf_g \mathbb{E}_{r_h, s_{h+1}|s_h, a_h} l_{\tilde{\pi}}(g, f_{h+1}; z_h).$$

Thus, the quantity $\Delta L_{\tilde{\pi}}(f_h, f_{h+1}; z_h)$ can be referred to as the *excess TD loss*, incurred by the predictor $f_h$, relative to the TD regression function $\mathbb{T}_h^{\tilde{\pi}} f_{h+1}$, on data $z_h$ and conditioned on $f_{h+1}$ and $\tilde{\pi}$.

## B.1  Variance condition and Bernstein's inequality

We also define the $\sigma$-algebra $\mathcal{A}_h^k := \sigma(\mathcal{D}_{k-1} \cup \{(s_{h'}^k, a_{h'}^k, r_{h'}^k)\}_{h' \in [h-1]} \cup (s_h^k, a_h^k))$ and denote $\mathbb{E}_{k,h}[\cdot] := \mathbb{E}[\cdot|\mathcal{A}_h^k]$. The following lemma establishes the variance condition on the excess TD loss, a TD analogous to the variance condition that is widely used in the empirical process theory [Massart, 2000].

**Lemma B.1.** *For any $\mathcal{A}_h^k$-measurable policy $\pi$, we have*

$$\mathbb{E}_{k,h}[\Delta L_\pi(f_h, f_{h+1}; z_h^k)] = \mathcal{E}_h^\pi(f_h, f_{h+1})(s_h^k, a_h^k)^2,$$
$$\mathbb{E}_{k,h}[\Delta L_\pi(f_h, f_{h+1}; z_h^k)^2] \leq 36b^2 \mathcal{E}_h^\pi(f_h, f_{h+1})(s_h^k, a_h^k)^2.$$

*Proof of Lemma B.1.* The result directly exploits the boundedness of the TD loss function and the squared loss is Lipschitz. In concrete, it is a direct application of Lemma G.1. □

The following lemma establishes the martingale extension of Bernstein's inequality, typically called Freedman's inequality [Freedman, 1975]. In this lemma, we prove a slightly modified version of the original Freedman's inequality for our own convenience. The proof for this lemma is elementary which we also show here.

**Lemma B.2** (Freedman's inequality). *Let $X_1, \ldots, X_T$ be* any *sequence of real-valued random variables. Denote $\mathbb{E}_t[\cdot] = \mathbb{E}[\cdot | X_1, \ldots, X_{t-1}]$. Assume that $X_t \leq R$ for some $R > 0$ and $\mathbb{E}_t[X_t] = 0$ for all $t$. Define the random variables*

$$S := \sum_{t=1}^{T} X_t, \quad V := \sum_{i=1}^{T} \mathbb{E}_t[X_t^2].$$

*Then for any $\delta > 0$, with probability at least $1 - \delta$, for any $\lambda \in [0, 1/R]$,*

$$S \leq (e - 2)\lambda V + \frac{\ln(1/\delta)}{\lambda}.$$

*Proof of Lemma B.2.* Let us define the following sequence of random variables: $Z_0 = 1, Z_t = Z_{t-1} \frac{e^{\lambda X_t}}{\mathbb{E}_t[e^{\lambda X_t}]}$. We have

$$\mathbb{E}_t[Z_t] = \mathbb{E}_t \left[ Z_{t-1} \frac{e^{\lambda X_t}}{\mathbb{E}_t[e^{\lambda X_t}]} \right] = \frac{Z_{t-1}}{\mathbb{E}_t[e^{\lambda X_t}]} \mathbb{E}_t[e^{\lambda X_t}] = Z_{t-1}.$$

Thus, we have

$$\mathbb{E}[Z_T] = \mathbb{E}\mathbb{E}_T[Z_T] = \mathbb{E}[Z_{T-1}] = \ldots = \mathbb{E}[Z_0] = 1.$$

Note that

$$Z_T = \frac{e^{\lambda S}}{\prod_{t=1}^{T} \mathbb{E}_t[e^{\lambda X_t}]} = \frac{e^{\lambda S}}{\sum_{t=1}^{T} e^{\ln \mathbb{E}_t[e^{\lambda X_t}]}} = \exp\left( \lambda S - \sum_{t=1}^{T} \ln \mathbb{E}_t[e^{\lambda X_t}] \right). \tag{3}$$

Since $Z_T \geq 0$, it follows from Markov's inequality that, for any $\delta > 0$, we have

$$\Pr(Z_T \geq 1/\delta) \leq \delta \mathbb{E}[Z_T] = \delta. \tag{4}$$

We now bound the logarithmic moment generating function $\ln \mathbb{E}_t[e^{\lambda X_t}]$ using elementary inequalities: For any $\lambda \in [0, 1/R]$, we have

$$\ln \mathbb{E}_t[e^{\lambda X_t}] \leq \mathbb{E}_t[e^{\lambda X_t}] - 1 \leq \lambda \mathbb{E}_t[X_t] + (e - 2)\lambda^2 \mathbb{E}_t[X_t^2], \tag{5}$$

where the first inequality uses $\ln z \leq z - 1, \forall z \geq 0$ and the second inequality uses that $e^z \leq 1 + z + (e - 2)z^2, \forall z \leq 1$ and that $\lambda X_t \leq 1$.

Plugging Equation (5) into Equation (3), then all together into Equation (4) complete the proof.

$\square$

## B.2 Functional projections for misspecification

Since Assumption 2.1 and Assumption 2.2 allow misspecification up to some errors $\xi$ and $\nu$, while we are working on the function class $\mathcal{F}$, we rely on the following projection operators, Definition 4 and Definition 5, to handle misspecification.

**Definition 4** (Projection of action-value functions). *For any $\tilde{\pi} \in \Pi^{soft}(T)$ for some $T \in \mathbb{N}$, we define the projection of the state-action value function $\tilde{\pi}$ onto $\mathcal{F}$ as*

$$\mathrm{Proj}_{\mathcal{F}}(Q^{\tilde{\pi}}) := \arg\min_{f \in \mathcal{F}} \left\{ |f_h(s_h, a_h) - Q_h^{\tilde{\pi}}(s_h, a_h)|, \forall h \in [H], (s_h, a_h) \in \mathrm{supp}(d_h^{\mu}) \right\}.$$

By Assumption 2.1, we have

$$|\mathrm{Proj}_{\mathcal{F}}(Q^{\tilde{\pi}})(s_h, a_h) - Q_h^{\tilde{\pi}}(s_h, a_h)| \leq \xi_h, \quad \forall h \in [H], (s_h, a_h) \in \mathrm{supp}(d_h^{\mu}).$$

**Definition 5** (Projection of Bellman operations). *For any $f \in \mathcal{F}$ and $\tilde{\pi} \in \Pi^{soft}(T)$ for some $T$, we define the projection of the Bellman operation $\mathbb{T}^{\tilde{\pi}} f$ onto $\mathcal{F}$ as*

$$\mathrm{Proj}_{\mathcal{F}}(\mathbb{T}^{\tilde{\pi}} f) := \arg\min_{f' \in \mathcal{F}} \left\{ \|f_h' - \mathbb{T}_h^{\tilde{\pi}} f_{h+1}\|_{\infty}, \forall h \in [H] \right\}.$$

By Assumption 2.2, we have

$$\|\mathrm{Proj}_{\mathcal{F}}(\mathbb{T}^{\tilde{\pi}} f) - \mathbb{T}_h^{\tilde{\pi}} f_{h+1}\|_{\infty} \leq \nu_h, \forall h \in [H].$$

## B.3 Induced MDPs

We now introduce the notion of *induced* MDPs which is originally used in [Zanette et al., 2021].

**Definition 6** (Induced MDPs). *For any policy $\pi \in \Pi^{all}$ and any sequence of functions $Q = \{Q_h\}_{h \in [H]} \in \{\mathcal{S} \times \mathcal{A} \to \mathbb{R}\}^H$, the $(Q, \pi)$-induced MDPs, denoted by $M(Q, \pi)$ is the MDP that is identical to the original MDP $M$ except only that the expected reward of $M(Q, \pi)$ is given by $\{r_h^{\pi, Q}\}_{h \in [H]}$, where*

$$r_h^{\pi, Q}(s, a) := r_h(s, a) - \mathcal{E}_h^\pi(f_h, f_{h+1})(s, a).$$

By definition of $M(\pi, Q)$, $Q$ is the fixed point of the Bellman equation $Q_h = \mathbb{T}_{h, M(\pi, Q)}^\pi Q_{h+1}$.

**Lemma B.3.** *For any $\pi \in \Pi^{all}$ and any sequence of functions $Q = \{Q_h\}_{h \in [H]} \in \{\mathcal{S} \times \mathcal{A} \to \mathbb{R}\}^H$, we have*

$$Q_{M(\pi, Q)}^\pi = Q,$$

*where $M(\pi, Q)$ is the induced MDP given in Definition 6.*

## B.4 Error decomposition

The key starting point for the proofs of all of the three main theorems is the following error decomposition that decomposes the sub-optimality into three sources of errors: the Bellman error under the comparator policy $\pi$, the gap values in the initial states, and the online-regret term due to the induced MDPs. In online RL, the sub-optimality of a greedy policy against an optimal policy can be decomposed into the sub-optimality in the Bellman errors and the error in the initial states [Dann et al., 2021], using the standard value-function error decomposition in [Jiang et al., 2017, Lemma 1]. However, in our setting, we compete against an arbitrary policy $\pi$ (not necessarily an optimal policy) and the learned policy $\pi^t$ is not greedy with respect to the current action-value function $Q^t$ – thus [Jiang et al., 2017, Lemma 1] cannot apply here. Instead, we develop an error decomposition – Lemma B.4 which generalizes what was implicit in Zanette et al. [2021].

**Lemma B.4** (Error decomposition). *For any action-value functions $Q \in \{\mathcal{S} \times \mathcal{A} \to \mathbb{R}\}^H$ and any policies $\pi, \tilde{\pi} \in \Pi^{all}$, we have*

$$\mathrm{SubOpt}_\pi^M(\tilde{\pi}) = \sum_{h=1}^H \mathbb{E}_\pi[\mathcal{E}_h^{\tilde{\pi}}(Q_h, Q_{h+1})(s_h, a_h)] + Q_1(s_1, \tilde{\pi}_1) - V_1^{\tilde{\pi}}(s_1) + \mathrm{SubOpt}_\pi^{M(Q, \tilde{\pi})}(\tilde{\pi}).$$

*Proof of Lemma B.4.* We have

$$\mathrm{SubOpt}_\pi^M(\tilde{\pi}) = V_1^\pi(s_1) - V_1^{\tilde{\pi}}(s_1)$$
$$= \left(V_1^\pi(s_1) - V_{1, M(Q, \tilde{\pi})}^\pi(s_1)\right) + \left(V_{1, M(Q, \tilde{\pi})}^{\tilde{\pi}}(s_1) - V_1^{\tilde{\pi}}(s_1)\right) + \left(V_{1, M(Q, \tilde{\pi})}^\pi(s_1) - V_{1, M(Q, \tilde{\pi})}^{\tilde{\pi}}(s_1)\right)$$
$$= \sum_{h=1}^H \mathbb{E}_\pi[\mathcal{E}_h^{\tilde{\pi}}(Q_h, Q_{h+1})(s_h, a_h)] + Q_1(s_1, \tilde{\pi}_1) - V_1^{\tilde{\pi}}(s_1) + \mathrm{SubOpt}_\pi^{M(Q, \tilde{\pi})}(\tilde{\pi}),$$

where in the last equality, for the first term, we use, by Definition 6, that

$$V_1^\pi(s_1) - V_{1, M(Q, \tilde{\pi})}^\pi(s_1) = \sum_{h=1}^H \mathbb{E}_\pi \left[r_h(s_h, a_h) - r_h^{\tilde{\pi}, Q}(s_h, a_h)\right] = \sum_{h=1}^H \mathbb{E}_\pi[\mathcal{E}_h^{\tilde{\pi}}(Q_h, Q_{h+1})(s_h, a_h)],$$

for the second term, we use, by Lemma B.3, that

$$V_{1, M(Q, \tilde{\pi})}^\pi(s_1) = Q_{1, M(Q, \tilde{\pi})}^\pi(s_1, \tilde{\pi}_1) = Q_1(s_1, \tilde{\pi}),$$

and for the last term, we use the definition of value sub-optimality in Equation (1). $\square$

## B.5 Decoupling lemma

One of the central tools for our proofs is the following decoupling lemma. The decoupling lemma essentially decouples the Bellman residuals under the $\pi$-induced state-action distribution into the squared Bellman residuals under the $\mu$-induced state-action distribution and the new data diversity measure in Definition 3 and additive terms of low order.

**Lemma B.5** (Decoupling argument). *Under Assumption 2.2, for any $f \in \mathcal{F}$, any $\tilde{\pi} \in \Pi^{soft}(T)$ for some $T$, any $\pi \in \Pi^{all}$, any $\lambda > 0$, and any $\epsilon \geq 0$, we have*

$$\sum_{h=1}^{H} \mathbb{E}_\pi[\mathcal{E}_h^{\tilde{\pi}}(f_h, f_{h+1})(s_h, a_h)] \leq \frac{1}{2\lambda} \sum_{h=1}^{H} \left( \sum_{k=1}^{K} \mathbb{E}_{\mu^k}\left[\mathcal{E}_h^{\tilde{\pi}}(f_h, f_{h+1})(s_h, a_h)^2\right] + K\nu_h^2 + 4bK\nu_h \right)$$
$$+ \frac{\lambda H \cdot \mathcal{C}(\pi; \epsilon)}{2K} + H\epsilon + \sum_{h=1}^{H} \nu_h,$$

*where $\mathcal{C}(\pi; \epsilon)$ is defined in Definition 3.*

*Proof of Lemma B.5.* We have

$$\sum_{h=1}^{H} \mathbb{E}_\pi[\mathcal{E}_h^{\tilde{\pi}}(f_h, f_{h+1})(s_h, a_h)] = \sum_{h=1}^{H} \mathbb{E}_\pi \left[ (\mathbb{T}_h^{\tilde{\pi}} f_{h+1} - f_h)(s_h, a_h) \right]$$

$$\leq \sum_{h=1}^{H} \mathbb{E}_\pi \left[ (\text{Proj}_{\mathcal{F}_h}(\mathbb{T}_h^{\tilde{\pi}} f_{h+1}) - f_h)(s_h, a_h) \right] + \bar{\nu}$$

$$\leq \sum_{h=1}^{H} \sqrt{\mathcal{C}(\pi, \epsilon)\mathbb{E}_\mu \left[ (\text{Proj}_{\mathcal{F}_h}(\mathbb{T}_h^{\tilde{\pi}} f_{h+1}) - f_h)(s_h, a_h)^2 \right]} + H\epsilon + \bar{\nu}$$

$$\leq \sum_{h=1}^{H} \sqrt{\mathcal{C}(\pi, \epsilon)\left(\mathbb{E}_\mu[\mathcal{E}_h^{\tilde{\pi}}(f_h, f_{h+1})(s_h, a_h)^2] + \nu_h^2 + 4b\nu_h\right)} + H\epsilon + \bar{\nu}$$

$$\leq \sqrt{H\mathcal{C}(\pi, \epsilon) \sum_{h=1}^{H} \left(\mathbb{E}_\mu[\mathcal{E}_h^{\tilde{\pi}}(f_h, f_{h+1})(s_h, a_h)^2] + \nu_h^2 + 4b\nu_h\right)} + H\epsilon + \bar{\nu}$$

$$\leq \frac{K}{2\lambda} \sum_{h=1}^{H} \left(\mathbb{E}_\mu[\mathcal{E}_h^{\tilde{\pi}}(f_h, f_{h+1})(s_h, a_h)^2] + \nu_h^2 + 4b\nu_h\right) + \frac{\lambda H\mathcal{C}(\pi, \epsilon)}{2K} + H\epsilon + \bar{\nu},$$

where the first inequality uses Assumption 2.2, the second inequality uses the definition of $C_\pi(\epsilon)$, the third inequality uses Assumption 2.2 (again), the fourth inequality uses Cauchy-Schwartz inequality, and the last inequality uses the AM-GM inequality $\sqrt{xy} \leq \frac{K}{2\lambda}x + \frac{\lambda}{2K}y$. $\square$

## B.6 Regret of the multiplicative weights algorithm for the actors

Now we establish the regret bound for the online-regret term due to the induced MDPs. The result in the following lemma is quite standard and can be readily generalized from a similar result in Zanette et al. [2021]. We present the proof here for completeness.

**Lemma B.6.** *Consider an arbitrary sequence of value functions $\{Q^t\}_{t \in [T]}$ such that $\max_{h,t} \|Q_h^t\|_\infty \leq b$ and define the following sequence of policies $\{\pi^t\}_{t \in [T+1]}$ where*

$$\pi^1(\cdot|s) = \text{Uniform}(\mathcal{A}), \forall s,$$
$$\pi_h^{t+1}(a|s) \propto \pi_h^t(a|s) \exp\left(\eta Q_h^t(s, a)\right), \forall(s, a, h, t).$$

*Suppose $\eta = \sqrt{\frac{\ln \text{Vol}(\mathcal{A})}{4(e-2)b^2 T}}$ and $T \geq \frac{\ln \text{Vol}(\mathcal{A})}{(e-2)}$, where $\text{Vol}(\mathcal{A})$ denotes the volume of the action set $\mathcal{A}$.*
[8] *For an arbitrary policy $\pi \in \Pi^{all}$, we have*

$$\sum_{t=1}^{T} \left( V_{1,M(\pi^t, Q^t)}^{\pi}(s_1) - V_{1,M(\pi^t, Q^t)}^{\pi^t}(s_1) \right) \leq 4Hb\sqrt{T \ln \text{Vol}(\mathcal{A})}.$$

---

[8]When $|\mathcal{A}| < \infty$, $\text{Vol}(\mathcal{A}) = |\mathcal{A}|$.

*Proof of Lemma B.6.* The proof for this lemma is quite standard as shown in Zanette et al. [2021]. We rewrote the proof with a slight modification for completeness. For simplicity, we write $M_t := M(\pi^t, Q^t)$. We will see that the key property that enables this lemma is that $Q_h^t = Q_{h,M_t}^{\pi^t}$ (Lemma B.3), which allows us to relate the value difference lemma to the log policy ratio. Using the value difference lemma (Lemma G.2), we have

$$V_{1,M_t}^{\pi}(s_1) - V_{1,M_t}^{\pi^t}(s_1) = \sum_{h=1}^{H} \mathbb{E}_\pi A_{h,M_t}^{\pi^t}(s_h, a_h),$$

where $\mathbb{E}_\pi$ is the expectation over the random trajectory $(s_1, a_1, \ldots, s_H, a_H)$ generated by $\pi$ (and the underlying MDP $M_t$[9]). For any $V : \mathcal{S} \to \mathbb{R}$, it follows from the definition of $\{\pi^t\}$ update that we have

$$\log \frac{\pi_h^{t+1}(a|s)}{\pi_h^t(a|s)} = \eta Q_h^t(s,a) - \log\left(\mathbb{E}_{a \sim \pi_h^t(\cdot|s)}\left[\exp\left(\eta Q_h^t(s,a)\right)\right]\right)$$

$$= \eta(Q_h^t(s,a) - V(s)) - \log\left(\mathbb{E}_{a \sim \pi_h^t(\cdot|s)}\left[\exp\left(\eta(Q_h^t(s,a) - V(s))\right)\right]\right).$$

In the equation above, noting that $Q_h^t = Q_{h,M_t}^{\pi^t}$ (Lemma B.3) and replacing $V(s)$ by $V_{h,M_t}^{\pi^t}$, we have

$$\log \frac{\pi_h^{t+1}(a|s)}{\pi_h^t(a|s)} = \eta A_{h,M_t}^{\pi^t}(s,a) - \log\left(\mathbb{E}_{a \sim \pi_h^t(\cdot|s)}\left[\exp\left(\eta A_{h,M_t}^{\pi^t}(s,a)\right)\right]\right), \qquad (6)$$

where we define the advantage function $A_M^\pi = \{A_{h,M}^\pi\}_{h \in [H]}$ as

$$A_{h,M}^\pi(s,a) := Q_{h,M}^\pi(s,a) - V_{h,M}^\pi(s), \forall(s,a,h).$$

Note that $|A_{h,M_t}^{\pi^t}(s,a)| \leq 2b$. By choosing $\eta \in (0, 1/(2b))$, we have

$$\log\left(\mathbb{E}_{a \sim \pi_h^t(\cdot|s)}\left[\exp\left(\eta A_{h,M_t}^{\pi^t}(s,a)\right)\right]\right)$$

$$\leq \log\left(\mathbb{E}_{a \sim \pi_h^t(\cdot|s)}\left[1 + \eta A_{h,M_t}^{\pi^t}(s,a) + (e-2)\eta^2 A_{h,M_t}^{\pi^t}(s,a)^2\right]\right)$$

$$= \log\left(\mathbb{E}_{a \sim \pi_h^t(\cdot|s)}\left[1 + (e-2)\eta^2 A_{h,M_t}^{\pi^t}(s,a)^2\right]\right)$$

$$\leq \log(1 + (e-2)\eta^2 4b^2)$$

$$\leq 4(e-2)b^2\eta^2 \qquad (7)$$

where the first inequality uses that $e^x \leq 1 + x + (e-2)x^2, \forall x \leq 1$ and $|\eta A_{h,M_t}^{\pi^t}(s,a)| \leq 1, \forall(s,a)$, the first equality uses that $\mathbb{E}_{a \sim \pi_h^t(\cdot|s)}\left[A_{h,M_t}^{\pi^t}(s,a)\right] = 0$, the second inequality uses that $|A_{h,M_t}^{\pi^t}(s,a)| \leq 2b$, and the last inequality uses that $\log(1+x) \leq x, \forall x \geq 0$. Combining Equation (7) and Equation (6), we have

$$A_{h,M_t}^{\pi^t}(s,a) \leq \frac{1}{\eta} \log \frac{\pi_h^{t+1}(a|s)}{\pi_h^t(a|s)} + 4(e-2)b^2\eta.$$

Thus, for any $h \in [H]$, we have

$$\sum_{t=1}^{T} \mathbb{E}_\pi A_{h,M_t}^{\pi^t}(s_h, a_h)$$

$$\leq \frac{1}{\eta} \sum_{t=1}^{T} \left(\mathbb{E}_\pi\left[KL[\pi_h(\cdot|s_h)\|\pi_h^t(\cdot|s_h)]\right] - \mathbb{E}_\pi\left[KL[\pi_h(\cdot|s_h)\|\pi_h^{t+1}(\cdot|s_h)]\right]\right) + 4(e-2)Tb^2\eta$$

---

[9] Note that $\Pr((s_1, a_1, \ldots, s_H, a_H)|\pi, M) = \Pr((s_1, a_1, \ldots, s_H, a_H)|\pi, M_t)$ since $M_t$ and $M$ have identical transition kernels.

$$= \frac{1}{\eta} \left( \mathbb{E}_\pi \left[ KL[\pi_h(\cdot|s_h)\|\pi_h^1(\cdot|s_h)] \right] - \mathbb{E}_\pi \left[ KL[\pi_h(\cdot|s_h)\|\pi_h^{T+1}(\cdot|s_h)] \right] \right) + 4(e-2)Tb^2\eta$$

$$\leq \frac{1}{\eta} \mathbb{E}_\pi \left[ KL[\pi_h(\cdot|s_h)\|\pi_h^1(\cdot|s_h)] \right] + 4(e-2)Tb^2\eta$$

$$\leq \frac{1}{\eta} \log(\text{Vol}(\mathcal{A})) + 4(e-2)Tb^2\eta,$$

where the second inequality uses the non-negativity of KL divergence, and the last inequality uses that $KL[\pi_h(\cdot|s_h)\|\pi_h^1(\cdot|s_h)] = -H[\pi_h(\cdot|s_h)] + \log(\text{Vol}(\mathcal{A})) \leq \log(\text{Vol}(\mathcal{A}))$ where $\pi^1$ is uniform over $\mathcal{A}$ and $\text{Vol}(\mathcal{A})$ denotes the volume over the compact set $\mathcal{A}$. Combining all pieces together, we have

$$\sum_{t=1}^{T} \mathbb{E}_\pi A_{h,M_t}^{\pi^t}(s_h, a_h) \leq \frac{1}{\eta} H \log(\text{Vol}(\mathcal{A})) + 4(e-2)THb^2\eta.$$

Minimizing the RHS of the above equation with respect to $\eta$ yields $\eta = \sqrt{\frac{\log \text{Vol}(\mathcal{A})}{4(e-2)b^2T}}$ and

$$\sum_{t=1}^{T} \mathbb{E}_\pi A_{h,M_t}^{\pi^t}(s_h, a_h) \leq 2H\sqrt{4(e-2)b^2T \log \text{Vol}(\mathcal{A})} \leq 4Hb\sqrt{T \log \text{Vol}(\mathcal{A})}.$$

Finally, we need that

$$\eta = \sqrt{\frac{\log \text{Vol}(\mathcal{A})}{4(e-2)b^2T}} \leq \frac{1}{2b},$$

which implies $T \geq \frac{\ln \text{Vol}(\mathcal{A})}{(e-2)}$. $\qquad\qquad\square$

We are now ready to establish the proofs of our three main theorems.

## Appendix C    Proof of Theorem 1

To construct our proof for Theorem 1, we first establish two following lemmas. The first lemma, Lemma C.1 establishes that the in-distribution squared Bellman residuals are bounded by the unbiased proxy of the squared Bellman error $\mathcal{L}_{\tilde{\pi}}(f)$, up to some estimation and approximation errors. The second lemma, Lemma C.2, asserts that the unbiased proxy of the squared Bellman error at the projection of $Q^{\tilde{\pi}}$ is close to zero, up to some estimation and approximation errors.

**Lemma C.1.** *For any $\delta > 0, \epsilon > 0$ and any $T \in \mathbb{N}$, under Assumption 2.2, with probability at least $1 - \delta$, it holds uniformly over all $f \in \mathcal{F}$ and $\tilde{\pi} \in \Pi^{soft}(T)$ that*

$$\sum_{h=1}^{H}\sum_{k=1}^{K} \mathbb{E}_k \left[ \mathcal{E}_h^{\tilde{\pi}}(f_h, f_{h+1})(s_h, a_h)^2 \right] \leq 2\mathcal{L}_{\tilde{\pi}}(f) + 40b(b+2)KH\epsilon + 12bK\sum_{h=1}^{H}\nu_h$$
$$+ 144(e-2)b^2H\left[ d_{\mathcal{F}}(\epsilon) + d_{\Pi}(\epsilon, T) + \ln(1/\delta) \right],$$

*where $\mathcal{L}_{\tilde{\pi}}(f) := \sum_{h=1}^{H} \hat{L}_{\tilde{\pi}}(f_h, f_{h+1}) - \inf_{g \in \mathcal{F}} \sum_{h=1}^{H} \hat{L}_{\tilde{\pi}}(g_h, f_{h+1})$.*

**Lemma C.2.** *Under Assumption 2.1, for any $T \in \mathbb{N}$, with probability at least $1 - \delta$, it holds uniformly for any $\tilde{\pi} \in \Pi^{soft}(T)$ that*

$$\mathcal{L}_{\tilde{\pi}}(\text{Proj}_{\mathcal{F}}(Q^{\tilde{\pi}})) \leq 36(e-2)b^2H\left( 2d_{\mathcal{F}}(\epsilon) + d_{\Pi}(\epsilon, T) + \ln\frac{H}{\delta} \right) + 6b(3b+4)\epsilon KH + 15bK\sum_{h=1}^{H}\xi_h,$$

*where $\text{Proj}_{\mathcal{F}}(Q^{\tilde{\pi}})$ is the projection of $Q^{\tilde{\pi}}$ onto $\mathcal{F}$, formally defined in Definition 4.*

## C.1    Proof of Theorem 1

With the two lemmas above, we are ready to prove Theorem 1. This proof is also laying a foundational step for our proofs of Theorem 2 and Theorem 3 that we shall present shortly. The proofs for the two lemmas above are presented immediately after the proof of Theorem 1.

*Proof of Theorem 1.*  Using Lemma B.4, we have

$$\mathrm{SubOpt}_\pi^M(\pi^t) = \sum_{h=1}^H \mathbb{E}_\pi[\mathcal{E}_h^{\pi^t}(\underline{Q}_h^t, \underline{Q}_{h+1}^t)(s_h, a_h)] + \Delta_1\underline{Q}_1(s_1, \pi_1^t) + \mathrm{SubOpt}_\pi^{M_t}(\pi^t)$$

where we denote

$$M_t := M(\underline{Q}^t, \pi^t),$$

$$\Delta_1\underline{Q}_1(s_1, \pi^t) := \underline{Q}_1(s_1, \pi^t) - V_1^{\pi^t}(s_1).$$

**Bounding $\sum_{t=1}^T \mathrm{SubOpt}_\pi^{M_t}(\pi^t)$.**   Note that $\sum_{t=1}^T \mathrm{SubOpt}_\pi^{M_t}(\pi^t)$ can be controlled by standard tools from online learning (Lemma B.6); thus it remains to control the first $H + 1$ terms.

**Bounding $\Delta_1\underline{Q}_1(s_1, \pi_1^t)$.**   Due to Lemma C.2, the event that $\mathrm{Proj}_\mathcal{F}(Q^{\pi^t}) \in \mathcal{F}(\beta; \pi^t)$ holds occur at probability at least $1 - \delta$. Furthermore, under this event, we have

$$\Delta_1\underline{Q}_1(s_1, \pi_1^t) = \underline{Q}_1(s_1, \pi^t) - V_1^{\pi^t}(s_1)$$
$$\leq \mathrm{Proj}_{\mathcal{F}_1}(Q_1^{\pi^t}) - V_1^{\pi^t}(s_1)$$
$$\leq \xi_1,$$

where the first inequality exploits Line 2 of Algorithm 2, and the last inequality uses Assumption 2.1.

**Bounding $\sum_{h=1}^H \mathbb{E}_\pi[\mathcal{E}_h^{\pi^t}(\underline{Q}_h^t, \underline{Q}_{h+1}^t)(s_h, a_h)]$.**   It follows from Lemma B.5 that

$$\sum_{h=1}^H \mathbb{E}_\pi[\mathcal{E}_h^{\pi^t}(\underline{Q}_h^t, \underline{Q}_{h+1}^t)(s_h, a_h)] \leq \sqrt{H\mathcal{C}(\pi; \epsilon)\sum_{h=1}^H \left(\mathbb{E}_\mu[\mathcal{E}_h^{\tilde{\pi}}(\underline{Q}_h^t, \underline{Q}_{h+1}^t)(s_h, a_h)^2] + \nu_h^2 + 4b\nu_h\right)}$$
$$+ H\epsilon + \bar{\nu}.$$

The term $\sum_{h=1}^H \mathbb{E}_\mu[\mathcal{E}_h^{\tilde{\pi}}(\underline{Q}_h^t, \underline{Q}_{h+1}^t)(s_h, a_h)^2]$ is bounded by Lemma C.1, with notice that $\mathcal{L}_{\pi^t}(\underline{Q}^t) \leq \beta$ (due to the definition of $\mathcal{F}(\beta; \pi^t)$ in Algorithm 2).

Combining the three steps above via the union bound completes our proof.    □

We now prove the two support lemmas.

## C.2    Proof of Lemma C.1

*Proof of Lemma C.1.*  Let us consider any fixed $f \in \mathcal{F}$ and any $\pi \in \Pi^{all}$. By Lemma B.1, we have

$$\mathbb{E}_{k,h}[\Delta L_\pi(f_h, f_{h+1}; z_h^k)] = \mathcal{E}_h^\pi(f_h, f_{h+1})(s_h^k, a_h^k)^2,$$
$$\mathbb{E}_{k,h}[\Delta L_\pi(f_h, f_{h+1}; z_h^k)^2] \leq 36b^2 \mathcal{E}_h^\pi(f_h, f_{h+1})(s_h^k, a_h^k)^2.$$

Combining with Lemma B.2, we have that with probability at least $1 - \delta$, for any $\iota \in [0, \frac{1}{13b^2}]$,

$$\sum_{k=1}^K \mathbb{E}_k[\mathcal{E}_h^\pi(f_h, f_{h+1})(s_h, a_h)^2] - \sum_{k=1}^K \Delta L_\pi(f_h, f_{h+1}; z_h^k)$$

$$\leq 36(e-2)b^2\iota \sum_{k=1}^K \mathbb{E}_k[\mathcal{E}_h^\pi(f_h, f_{h+1})(s_h, a_h)^2] + (1/\iota)\log(1/\delta).$$

By setting $\iota = \frac{1}{72(e-2)b^2}$, the above inequality becomes

$$\sum_{k=1}^{K} \mathbb{E}_k[\mathcal{E}_h^\pi(f_h, f_{h+1})(s_h, a_h)^2] \leq 2\sum_{k=1}^{K} \Delta L_\pi(f_h, f_{h+1}; z_h^k) + 144(e-2)b^2 \ln(1/\delta).$$

For any $\epsilon > 0$, and for any $f \in \mathcal{F}, \pi \in \Pi^{soft}(T)$, by definition of $\epsilon$-covering, there exist $f'$ and $\pi'$ in the $\epsilon$-cover of $\mathcal{F}$ and $\Pi^{soft}(T)$, i.e.,

$$\|f_h - f'_h\|_\infty \leq \epsilon, \|\pi_h - \pi'_h\|_{1,\infty} \leq \epsilon.$$

By simple calculations, we have

$$|\mathcal{E}_h^\pi(f_h, f_{h+1})(s_h, a_h)^2 - \mathcal{E}_h^{\pi'}(f'_h, f'_{h+1})(s_h, a_h)^2| \leq 4b(b+2)\epsilon,$$
$$|\Delta L_\pi(f_h, f_{h+1}; z_h^k) - \Delta L_{\pi'}(f'_h, f'_{h+1}; z_h^k)| \leq 18b(b+2)\epsilon.$$

Thus, by the union bound, we have with probability at least $1 - \delta$, it holds uniformly over all $f \in \mathcal{F}, \pi \in \Pi^{soft}(T)$ that

$$\sum_{h=1}^{H}\sum_{k=1}^{K} \mathbb{E}_k[\mathcal{E}_h^\pi(f_h, f_{h+1})(s_h, a_h)^2] \leq 2\sum_{h=1}^{H}\sum_{k=1}^{K} \Delta L_\pi(f_h, f_{h+1}; z_h^k) + 40b(b+2)KH\epsilon$$

$$+ 144(e-2)b^2 \sum_{h=1}^{H} \ln(N(\epsilon; \mathcal{F}_h, \|\cdot\|_\infty)N(\epsilon; \Pi_h^{soft}(T), \|\cdot\|_{1,\infty})/\delta).$$

Finally, notice that

$$|l_\pi(\mathbb{T}_h^\pi f_{h+1}, f_{h+1}; z_h) - l_\pi(\text{Proj}_{\mathcal{F}_h}(\mathbb{T}_h^\pi f_{h+1}), f_{h+1}; z_h)| \leq 6b\nu_h.$$

Thus, we have

$$\sum_{h=1}^{H}\sum_{k=1}^{K} \Delta L_\pi(f_h, f_{h+1}; z_h^k) \leq \mathcal{L}_\pi(f) + 6bK\sum_{h=1}^{H} \nu_h.$$

We can then conclude our proof. $\qquad\square$

## C.3 Proof of Lemma C.2

In order to prove Lemma C.2, we shall first prove the following lemma, which establishes the confidence radius of the empirical squared Bellman errors that we used to establish the version space in Algorithm 2.

**Lemma C.3.** *Consider any $\delta > 0, \epsilon > 0, T \in \mathbb{N}$, let*

$$\beta_\epsilon := 36(e-2)b^2 \left(2d_\mathcal{F}(\epsilon) + d_\Pi(\epsilon, T) + \ln(H/\delta)\right) + 6b(3b+4)\epsilon K.$$

*With probability at least $1 - \delta$, it holds uniformly over any $\pi \in \Pi^{soft}(T)$, $f \in \mathcal{F}$, and $h \in [H]$ that*

$$\sum_{k=1}^{K} \left(\mathbb{T}_h^\pi f_{h+1}(x_h^k) - r_h^k - f_{h+1}(s_{h+1}^k, \pi_{h+1})\right)^2 \leq \inf_{g_h \in \mathcal{F}_h} \sum_{k=1}^{K} \left(g_h(x_h^k) - r_h^k - f_{h+1}(s_{h+1}^k, \pi_{h+1})\right)^2 + \beta_\epsilon.$$

*Proof of Lemma C.3.* Let us fix any $h \in [H]$. For any $(f, g, \pi) \in \mathcal{F} \times \mathcal{F} \times \Pi^{soft}(T)$ and any $k \in [K]$, define the following random variable

$$Z_{k,h}(f, g, \pi) := \left(g_h(x_h^k) - r_h^k - f_{h+1}(s_{h+1}^k, \pi_{h+1})\right)^2 - \left(\mathbb{T}_h^\pi f_{h+1}(x_h^k) - r_h^k - f_{h+1}(s_{h+1}^k, \pi_{h+1})\right)^2.$$

Denote

$$\mathbb{E}_{k,h}[\cdot] := \mathbb{E}\left[\cdot \,\middle|\, \{z_h^i\}_{h\in[H]}^{i\in[k-1]}, s_1^k, a_1^k, r_1^k, \ldots, s_{h-1}^k, a_{h-1}^k, r_{h-1}^k, s_h^k, a_h^k\right].$$

By Lemma B.1, we have

$$\mathbb{E}_{k,h}\left[Z_{k,h}(f,g,\pi)\right] = \mathcal{E}_h^\pi(g_h, f_{h+1})(s_h^k, a_h^k)^2,$$
$$\mathbb{E}_{k,h}\left[Z_{k,h}^2(f,g,\pi)\right] \le 36b^2 \mathcal{E}_h^\pi(g_h, f_{h+1})(s_h^k, a_h^k)^2.$$

Thus, combing with Lemma B.2, for any $(f, g, \pi) \in \mathcal{F} \times \mathcal{F} \times \Pi^{soft}(T)$, with probability at least $1 - \delta$, for any $\iota \in [0, \frac{1}{13b^2}]$,

$$\sum_{k=1}^{K}\mathbb{E}_{k,h}\left[Z_{k,h}(f,g,\pi)\right] - \sum_{k=1}^{K} Z_{k,h}(f,g,\pi) \le 36(e-2)b^2\iota \sum_{k=1}^{K}\mathcal{E}_h^\pi(g_h, f_{h+1})(s_h^k, a_h^k)^2 + \frac{\ln(1/\delta)}{\iota}.$$

By setting $\iota = 1/(36(e-2)b^2)$, the above inequality becomes

$$-\sum_{k=1}^{K} Z_{k,h}(f,g,\pi) \le 36(e-2)b^2 \ln(1/\delta). \tag{8}$$

For any $\epsilon > 0$, let $\mathcal{F}^\epsilon$ and $\Pi^\epsilon$ be $\epsilon$-covers of $\mathcal{F}$ and $\Pi^{soft}(T)$, respectively, with respect to $\|\cdot\|_\infty$ and $\|\cdot\|_{\infty,1}$, respectively, where $\|u - v\|_\infty := \sup_{(s,a)} |u(s,a) - v(s,a)|$ and $\|\pi - \pi'\|_{\infty,1} := \sup_s \sum_{a \in \mathcal{A}} |\pi(a|s) - \pi'(a|s)|$. Using the union bound, it follows from Equation (8) that with probability at least $1 - \delta$, it holds uniformly over any $h \in [H]$ and any $(f, g, \pi) \in \mathcal{F}^\epsilon \times \mathcal{F}^\epsilon \times \Pi^\epsilon$ that

$$-\sum_{k=1}^{K} Z_{k,h}(f,g,\pi) \le 18(e-2)b^2\left[\ln(H/\delta) + 2d_\mathcal{F}(\epsilon) + d_\Pi(\epsilon, T)\right].$$

For any $(f, g, \pi) \in \mathcal{F} \times \mathcal{F} \times \Pi^{soft}(T)$, there exist $(f_\epsilon, g_\epsilon, \pi_\epsilon) \in \mathcal{F}^\epsilon \times \mathcal{F}^\epsilon \times \Pi^\epsilon$ such that

$$\|f_h - (f_\epsilon)_h\|_\infty \le \epsilon, \|g_h - (g_\epsilon)_h\|_\infty \le \epsilon, \|\pi_h - (\pi_\epsilon)_h\|_{\infty,1} \le \epsilon, \forall h \in [H].$$

It is easy to compute the discretization error that

$$Z_{k,h}(f,g,\pi) - Z_{k,h}(f_\epsilon, g_\epsilon, \pi_\epsilon) \le 18b(b+1)\epsilon.$$

Using the discretization argument and the union bound complete our proof. $\qquad\square$

We are now ready to prove Lemma C.2.

*Proof of Lemma C.2.* Consider the event that the inequality in Lemma C.3 holds. Under this event, for any $\tilde{\pi} \in \Pi^{soft}(T)$, we have

$$\sum_{k=1}^{K} l_{\tilde{\pi}}(\mathrm{Proj}_{\mathcal{F}_h}(Q_h^{\tilde{\pi}}), \mathrm{Proj}_{\mathcal{F}_{h+1}}(Q_{h+1}^{\tilde{\pi}}); z_h^k) \le \sum_{k=1}^{K} l_{\tilde{\pi}}(Q_h^{\pi^t}, Q_{h+1}^{\tilde{\pi}}; z_h^k) + 6bK\xi_h$$

$$= \sum_{k=1}^{K} l_{\tilde{\pi}}(\mathbb{T}_h^{\tilde{\pi}} Q_{h+1}^{\tilde{\pi}}, Q_{h+1}^{\tilde{\pi}}; z_h^k) + 6bK\xi_h$$

$$\le \sum_{k=1}^{K} l_{\tilde{\pi}}(\mathbb{T}_h^{\tilde{\pi}} \mathrm{Proj}_{\mathcal{F}_{h+1}}(Q_{h+1}^{\tilde{\pi}}), \mathrm{Proj}_{\mathcal{F}_{h+1}}(Q_{h+1}^{\tilde{\pi}}); z_h^k) + 12bK\xi_h$$

$$\le \sum_{k=1}^{K} l_{\tilde{\pi}}(g_h, \mathrm{Proj}_{\mathcal{F}_{h+1}}(Q_{h+1}^{\tilde{\pi}}); z_h^k) + \beta_\epsilon + 12bK\xi_h \quad \text{(for any } g_h \in \mathcal{F}_h)$$

$$\le \sum_{k=1}^{K} l_{\tilde{\pi}}(g_h, Q_{h+1}^{\tilde{\pi}}; z_h^k) + \beta_\epsilon + 15bK\xi_h,$$

where we use Assumption 2.1 for the first, second, and last inequalities, the third inequality uses Lemma C.3, and the equality uses $Q_{h+1}^{\tilde{\pi}} = \mathbb{T}_h^{\tilde{\pi}} Q_{h+1}^{\tilde{\pi}}$. Rearranging the last inequality completes our proof. $\qquad\square$

# Appendix D  Proof of Theorem 2

In this appendix, we present our complete argument to establish Theorem 2. In order to prove Theorem 2, the key is to establish a connection from the squared Bellman error under the data distribution $\mu$ to the regularized objective in Algorithm 3. This key idea should become clear in the following proof.

*Proof of Theorem 2.* Similar to the proof of Theorem 1, our starting point is using Lemma B.4:

$$\mathrm{SubOpt}_\pi^M(\pi^t) = \sum_{h=1}^H \mathbb{E}_\pi[\mathcal{E}_h^{\pi^t}(\underline{Q}_h^t, \underline{Q}_{h+1}^t)(s_h, a_h)] + \Delta_1 \underline{Q}_1(s_1, \pi_1^t) + \mathrm{SubOpt}_\pi^{M_t}(\pi^t)$$

and we bound $\sum_{t=1}^T \mathrm{SubOpt}_\pi^{M_t}(\pi^t)$ using Lemma B.6. We now bound the remaining terms. For any $\gamma > 0$, we have

$$\sum_{h=1}^H \mathbb{E}_\pi[\mathcal{E}_h^{\pi^t}(\underline{Q}_h^t, \underline{Q}_{h+1}^t)(s_h, a_h)]$$

$$\leq \frac{K}{2\lambda} \sum_{h=1}^H \left( \mathbb{E}_\mu[\mathcal{E}_h^{\pi^t}(\underline{Q}_h^t, \underline{Q}_{h+1}^t)(s_h, a_h)^2] + \nu_h^2 + 4b\nu_h \right) + \frac{\lambda H \mathcal{C}(\pi, \epsilon)}{2K} + H\epsilon + \bar{\nu}$$

$$\leq \frac{\mathcal{L}_{\pi^t}(\underline{Q}) + 0.5\iota_1 + 0.5K\bar{\nu}^2 + 2bK\bar{\nu}}{\lambda} + \frac{\lambda H \mathcal{C}(\pi, \epsilon)}{2K} + H\epsilon + \bar{\nu},$$

where the first inequality uses Lemma B.5 and the second inequality uses Lemma C.1, and here $\iota_1 := 40b(b+2)KH\epsilon + 12bK \sum_{h=1}^H \nu_h + 144(e-2)b^2 H [d_\mathcal{F}(\epsilon) + d_\Pi(\epsilon, T) + \ln(1/\delta)]$. Thus, we have

$$\sum_{h=1}^H \mathbb{E}_\pi[\mathcal{E}_h^{\pi^t}(\underline{Q}_h^t, \underline{Q}_{h+1}^t)(s_h, a_h)] + \Delta_1 \underline{Q}_1(s_1, \pi_1^t)$$

$$\leq \frac{\mathcal{L}_{\pi^t}(\underline{Q}) + \lambda\Delta_1 \underline{Q}_1(s_1, \pi_1^t) + 0.5\iota_1 + 0.5K \sum_{h=1}^H \nu_h^2 + 2bK \sum_{h=1}^H \nu_h}{\lambda} + \frac{\lambda H \mathcal{C}(\pi, \epsilon)}{2K}$$

$$+ H\epsilon + \sum_{h=1}^H \nu_h$$

$$\leq \frac{\mathcal{L}_{\pi^t}(\mathrm{Proj}_\mathcal{F}(Q^{\pi^t})) + \lambda\Delta_1 \mathrm{Proj}_{\mathcal{F}_1}(Q_1^{\pi^t})(s_1, \pi_1^t) + 0.5\iota_1 + \sum_{h=1}^H \nu_h^2 + 2bK \sum_{h=1}^H \nu_h}{\lambda}$$

$$+ \frac{\lambda H \mathcal{C}(\pi, \epsilon)}{2K} + H\epsilon + \sum_{h=1}^H \nu_h$$

$$\leq \frac{\iota_2 + \lambda\xi_1 + 0.5\iota_1 + \sum_{h=1}^H \nu_h^2 + 2bK \sum_{h=1}^H \nu_h}{\lambda} + \frac{\lambda H \mathcal{C}(\pi, \epsilon)}{2K} + H\epsilon + \sum_{h=1}^H \nu_h,$$

where the second inequality uses the fact that $\underline{Q}_h^t$ is a minimizer over $\mathcal{F} \ni \mathrm{Proj}_\mathcal{F}(Q^{\pi^t})$ of $\mathcal{L}_{\pi^t}(f) + \lambda f_1(s_1, \pi_1^t)$ (which has the same minimizer as $\mathcal{L}_{\pi^t}(f) + \lambda\Delta_1 f_1(s_1, \pi_1^t)$), and the last inequality uses Lemma C.2, and here we define $\iota_2 := 36(e-2)b^2 H \left(2d_\mathcal{F}(\epsilon) + d_\Pi(\epsilon, T) + \ln \frac{H}{\delta}\right) + 6b(3b + 4)\epsilon KH + 15bK \sum_{h=1}^H \xi_h$. $\qquad\square$

# Appendix E  Proof of Theorem 3

In this appendix, we give our complete proof for Theorem 3. In order to develop our argument for proving Theorem 3, we shall start with a generalized form of posterior sampling in Section E.1 and develop our key support result in Proposition 2. We then use Proposition 2 and the similar machinery developed in Section D to complete our argument for proving Theorem 3.

## E.1 Generalized form of posterior and Proposition 2

We start with recalling the posterior distribution defined in Line 1 of Algorithm 4 as

$$\hat{p}(f|\mathcal{D}, \pi) \propto \exp\left(-\lambda f_1(s_1, \pi_1)\right) p_0(f) \prod_{h \in [H]} \frac{\exp\left(-\gamma \hat{L}_\pi(f_h, f_{h+1})\right)}{\mathbb{E}_{f'_h \sim p_{0,h}} \exp\left(-\gamma \hat{L}_\pi(f'_h, f_{h+1})\right)}. \tag{9}$$

Similar to the proof strategy in Dann et al. [2021], we now consider a slightly more general form of the posterior distribution with an extra parameter $\alpha \in [0, 1]$ and in an equivalent but more useful form. In concrete, consider any $\alpha \in [0, 1]$ and define the potential functions:

$$\widehat{\Phi}_h(f, \pi; \mathcal{D}) := -\ln p_0(f_h) + \alpha\gamma \sum_{k=1}^{K} \Delta L_{\tilde{\pi}}(f_h, f_{h+1}; z_h^k)$$

$$+ \alpha \ln \mathbb{E}_{\tilde{f}_h \sim p_0} \exp\left(-\gamma \sum_{k=1}^{K} \Delta L_{\tilde{\pi}}(f_h, f_{h+1}; z_h^k)\right),$$

$$\widehat{\Phi}(f, \pi; \mathcal{D}) := \sum_{h=1}^{H} \widehat{\Phi}_h(f, \pi; \mathcal{D}),$$

$$\Delta_1 f_1(s_1, \pi) := f_1(s_1, \pi) - V_1^\pi(s_1).$$

where recall that $\Delta L_{\tilde{\pi}}(f_h, f_{h+1}; z_h^k)$ is defined in Table 3. Define the generalized posterior distribution

$$\hat{p}(f|\mathcal{D}, \pi) \propto \exp\left(-\widehat{\Phi}(f, \pi; \mathcal{D}) - \lambda \Delta f_1(s_1, \pi)\right), \tag{10}$$

where it is equivalent to the posterior defined in Equation (9) when $\alpha = 1$. We shall use Equation (10) for the posterior for the rest of this section. We shall also define the complexity measure of this generalized posterior – a counterpart to that of the canonical posterior form in Definition 2.

**Definition 7.** *Define*

$$\kappa_h(\alpha, \epsilon, \tilde{\pi}) := (1 - \alpha) \ln \mathbb{E}_{f_{h+1} \sim p_0}\left[p_{0,h}\left(\mathcal{F}_h^{\tilde{\pi}}(\epsilon; f_{h+1})\right)^{-\alpha/(1-\alpha)}\right],$$

*where recall that* $\mathcal{F}_h^{\tilde{\pi}}(\epsilon; f_{h+1}) = \{f' \in \mathcal{F}_h : \sup_{s,a} |\mathcal{E}_h^{\tilde{\pi}}(f', f_{h+1})(s, a)| \le \epsilon\}$ *which is defined in Definition 2. Define the complexity measure*

$$d_0(\epsilon, \alpha) := \sup_{T \in \mathbb{N}, \tilde{\pi} \in \Pi^{soft}(T)} \sum_{h=1}^{H} \kappa_h(\alpha, \epsilon, \tilde{\pi}). \tag{11}$$

Note that we have

$$\lim_{\alpha \to 1^-} d_0(\epsilon, \alpha) = d_0(\epsilon).$$

We now state our key milestone result – Proposition 2 to support the argument for proving Theorem 3. The proof of Proposition 2 is deferred to Section E.3.

**Notation** $\mathbb{E}_{\tilde{\pi} \sim P_t(\cdot|\mathcal{D})}$**.** Note that in Algorithm 4, each policy $\pi^t$ for $t \in [T]$ is a random variable that depends on both the offline data $\mathcal{D}$ and the randomization of sampling from the posteriors. That is, when conditioned on the offline data $\mathcal{D}$, each $\pi_t$ is still a random variable. We denote $P_t(\cdot|\mathcal{D})$ as the posterior distribution of $\pi^t$ conditioned on $\mathcal{D}$. Note that for any $\tilde{\pi} \sim P_t(\cdot|\mathcal{D})$ and any $t \in [T]$, we have $\tilde{\pi} \in \Pi^{soft}(T)$.

**Proposition 2.** *For any* $\gamma \in [0, \frac{1}{144(e-2)b^2}]$, $\epsilon > 0$, $\delta > 0$, $\alpha \in (0, 1]$, $T \in \mathbb{N}$, *and any* $t \in [T]$ *and* $\lambda > 0$, *we have,*

$$\mathbb{E}_{\mathcal{D}} \mathbb{E}_{\tilde{\pi} \sim P_t(\cdot|\mathcal{D})} \mathbb{E}_{f \sim \hat{p}(\cdot|\mathcal{D}, \tilde{\pi})}\left[0.125\alpha\gamma K \sum_{h=1}^{H} \mathbb{E}_\mu[\mathcal{E}_h^{\tilde{\pi}}(f_h, f_{h+1})(s_h, a_h)^2] + \lambda \Delta f_1(s_1, \tilde{\pi})\right]$$

$$\lesssim \lambda\epsilon + \alpha\gamma H b^2 \cdot \max\{d_{\mathcal{F}}(\epsilon), d_{\Pi}(\epsilon, T), \ln\frac{\ln Kb^2}{\delta}\} + \alpha\gamma b^2 KH \cdot \max\{\epsilon, \delta\} + \gamma HK\frac{\epsilon^2}{\alpha}$$

$$+ \sum_{h=1}^{H} \sup_{\tilde{\pi}_h \in \Pi_h^{soft}(T)} \kappa_h(\alpha, \epsilon, \tilde{\pi}_h) + \sup_{\tilde{\pi} \in \Pi^{soft}(T)} \sum_{h=1}^{H} \ln\frac{1}{p_0(\mathcal{F}_h(\epsilon; Q_h^{\tilde{\pi}_h}))}.$$

We now have all main components needed to construct our argument for proving Theorem 3.

## E.2 Proof of Theorem 3

*Proof of Theorem 3.* We start with the error decomposition argument.

**Step 1: Error decomposition.** Similar to the first step of the proof of Theorem 2, using Lemma B.4, we have

$$\text{SubOpt}_\pi^M(\pi^t) = \sum_{h=1}^{H} \mathbb{E}_\pi[\mathcal{E}_h^{\pi^t}(\underline{Q}_h^t, \underline{Q}_{h+1}^t)(s_h, a_h)] + \Delta_1\underline{Q}_1(s_1, \pi_1^t) + \text{SubOpt}_\pi^{M_t}(\pi^t)$$

where we denote $M_t := M(\underline{Q}^t, \pi^t)$ and $\Delta_1\underline{Q}_1(s_1, \pi^t) := \underline{Q}_1(s_1, \pi^t) - V_1^{\pi^t}(s_1)$. Since term $\sum_{t=1}^{T} \text{SubOpt}_\pi^{M_t}(\pi^t)$ can be controlled Lemma B.6, it remains to control

$$J := \mathbb{E}_{\mathcal{D}}\left[\sum_{h=1}^{H} \mathbb{E}_\pi[\mathcal{E}_h^{\pi^t}(\underline{Q}_h^t, \underline{Q}_{h+1}^t)(s_h, a_h)] + \Delta_1\underline{Q}_1(s_1, \pi_1^t)\right]$$

$$= \mathbb{E}_{\mathcal{D}}\mathbb{E}_{\tilde{\pi}\sim P_t(\cdot|\mathcal{D})}\mathbb{E}_{f\sim\hat{p}(\cdot|\tilde{\pi},\mathcal{D})}\left[\sum_{h=1}^{H} \mathbb{E}_\pi[\mathcal{E}_h^{\tilde{\pi}}(f_h, f_{h+1})(s_h, a_h)] + \Delta_1 f_1(s_1, \tilde{\pi}_1)\right].$$

**Step 2: Decoupling argument.** Using Lemma B.5, we have

$$\sum_{h=1}^{H} \mathbb{E}_\pi[\mathcal{E}_h^{\tilde{\pi}}(f_h, f_{h+1})(s_h, a_h)] + \Delta_1 f_1(s_1, \tilde{\pi}_1)$$

$$\leq \frac{0.125K\gamma}{\lambda} \sum_{h=1}^{H} \left(\mathbb{E}_\mu[\mathcal{E}_h^{\tilde{\pi}}(f_h, f_{h+1})(s_h, a_h)^2] + \nu_h^2 + 4b\nu_h\right) + \frac{0.5\lambda HC(\pi, \epsilon_c)}{K\gamma} + \Delta_1 f_1(s_1, \tilde{\pi}_1)$$

$$+ H\epsilon_c + \sum_{h=1}^{H}\nu_h$$

$$= \frac{0.125K\gamma\sum_{h=1}^{H}\mathbb{E}_\mu[\mathcal{E}_h^{\tilde{\pi}}(f_h, f_{h+1})(s_h, a_h)^2] + \lambda\Delta_1 f_1(s_1, \tilde{\pi}_1) + \iota_1}{\lambda} + \frac{0.5\lambda HC(\pi, \epsilon_c)}{K\gamma}$$

$$+ H\epsilon_c + \sum_{h=1}^{H}\nu_h$$

where $\iota_1 := 0.125K\gamma\left(\sum_{h=1}^{H}\nu_h^2 + 4b\sum_{h=1}^{H}\nu_h\right)$.

Applying Proposition 2, taking the limit $\alpha \to 1^-$, and re-organizing the terms complete our proof.

$\square$

It remains to prove Proposition 2, which is the focus of the remaining appendix.

## E.3 Proof of Proposition 2

Our proof strategy for Proposition 2 builds upon Dann et al. [2021] where the central idea in the proof is to upper and lower bound the log-partition function – which in our case is as follows:

$$Z_t := \mathbb{E}_{\mathcal{D}}\mathbb{E}_{\tilde{\pi}\sim P_t(\cdot|\mathcal{D})}\mathbb{E}_{f\sim\hat{p}(\cdot|\mathcal{D},\tilde{\pi})}\left[\widehat{\Phi}(f, \tilde{\pi}; \mathcal{D}) + \lambda\Delta_1 f_1(s_1, \tilde{\pi}) + \ln\hat{p}(f|\mathcal{D}, \tilde{\pi})\right], \tag{12}$$

for any $t \in [T]$ and any $T \in \mathbb{N}$. The key technical distinction is that we need to handle the statistical dependence induced by $\mathbb{E}_{\tilde{\pi} \sim P_t(\cdot|\mathcal{D})}$ – which is absent in Dann et al. [2021]. In concrete, when $\tilde{\pi}$ depends on $\mathcal{D}$, then

$$\mathbb{E}\Delta L_h^{\tilde{\pi}}(f_h, f_{h+1})(s_h^k, a_h^k) \neq \mathcal{E}_h^{\tilde{\pi}}(f_h, f_{h+1})(s_h^k, a_h^k)^2,$$

since $\tilde{\pi}$ depends on $(s_h^k, a_h^k)$. We develop an machinery to handle such issue in posterior sampling by carefully controlling the variance of the variable of interest (thus we can leverage the variance-dependent concentration inequality in Lemma B.2) and integrating it into posterior sampling using a uniform convergence argument. Roughly speaking, several milestone results during the process of developing our proof argument, we need to bound the form of

$$\mathbb{E}_{\mathcal{D}}\mathbb{E}_{\tilde{\pi} \sim P_t(\cdot|\mathcal{D})}\mathbb{E}_{f \sim \hat{p}(\cdot|\mathcal{D}, \tilde{\pi})}[S(f, \tilde{\pi}, \mathcal{D})]$$

where $S(f, \tilde{\pi}, \mathcal{D})$ is a function of $f, \tilde{\pi}, \mathcal{D}$. It is useful to view $S(f, \tilde{\pi}, \mathcal{D})$ as a stochastic process indexed by $(f, \tilde{\pi})$. In our machinery, we shall first construct an upper bound on the variance of the random process, namely

$$V(f, \tilde{\pi}) \geq \mathbb{E}_{\mathcal{D}}[S(f, \tilde{\pi}, \mathcal{D})^2].$$

Using a discretization argument, the union bound and Lemma B.2, we have with probability at least $1 - \delta$, for any $f \in \mathcal{F}, \tilde{\pi} \in \Pi^{soft}(T)$, for any $t \in [0, \frac{1}{\sup S(f, \tilde{\pi}, \mathcal{D})}]$, we have

$$S(f, \tilde{\pi}, \mathcal{D}) \leq O_K(1) + \mathbb{E}_{\mathcal{D}}[S(f, \tilde{\pi}, \mathcal{D})] + (e - 2)t\mathbb{E}_{\mathcal{D}}[S(f, \tilde{\pi}, \mathcal{D})^2] + \frac{\ln(N/\delta)}{t}$$

where $O_K(1)$ is a discretization error that can be controlled, and $N$ is a covering number of $\mathcal{F} \times \Pi^{soft}(T)$. Note that $S(f, \tilde{\pi}, \mathcal{D})$ often involves the squared loss which satisfies the Bernstein condition (see Lemma G.1) – thus we can roughly bound $\mathbb{E}_{\mathcal{D}}[S(f, \tilde{\pi}, \mathcal{D})^2] \leq \alpha |\mathbb{E}_{\mathcal{D}}[S(f, \tilde{\pi}, \mathcal{D})]|$ for some constant $\alpha$. To integrate the high-probability bound into in-expected bound, we use the argument:

$$\mathbb{E}_{\mathcal{D}}\mathbb{E}_{\tilde{\pi} \sim P_t(\cdot|\mathcal{D})}\mathbb{E}_{f \sim \hat{p}(\cdot|\mathcal{D}, \tilde{\pi})}[S(f, \tilde{\pi}, \mathcal{D})] \leq O_K(1) + \mathbb{E}_{\mathcal{D}}\mathbb{E}_{\tilde{\pi} \sim P_t(\cdot|\mathcal{D})}\mathbb{E}_{\mathcal{D}}[S(f, \tilde{\pi}, \mathcal{D})]$$
$$+ (e - 2)t\mathbb{E}_{\mathcal{D}}\mathbb{E}_{\tilde{\pi} \sim P_t(\cdot|\mathcal{D})}\mathbb{E}_{\mathcal{D}}[S(f, \tilde{\pi}, \mathcal{D})^2] + \frac{\ln(N/\delta)}{t} + \delta \sup S(f, \tilde{\pi}, \mathcal{D}).$$

### E.3.1 Lower-bounding log-partition function.

In this appendix, we give a lower bound of the log-partition function defined in Equation (12). The final lower bound is presented in Proposition 3. In order to establish such a lower bound, we first present a series of support lemmas that will culminate into Proposition 3.

The following lemma decomposes the log-partition function $Z$ into different terms that we shall control separately.

**Lemma E.1.** *For any $t \in [T]$ and any $T \in \mathbb{N}$, we have*

$$Z_t \geq \underbrace{\mathbb{E}_{\mathcal{D}}\mathbb{E}_{\tilde{\pi} \sim P_t(\cdot|\mathcal{D})}\mathbb{E}_{f \sim \hat{p}(\cdot|\mathcal{D}, \tilde{\pi})}\left[\lambda\Delta f_1(s_1, \tilde{\pi}) + (1 - 0.5\alpha)\ln\frac{\hat{p}(f_1|\mathcal{D}, \tilde{\pi})}{p_0(f_1)}\right]}_{A_t}$$

$$+ 0.5\alpha\sum_{h=1}^{H}\underbrace{\mathbb{E}_{\mathcal{D}}\mathbb{E}_{\tilde{\pi} \sim P_t(\cdot|\mathcal{D})}\mathbb{E}_{f \sim \hat{p}(\cdot|\mathcal{D}, \tilde{\pi})}\left[2\gamma\sum_{k=1}^{K}\Delta L_{\tilde{\pi}}(f_h, f_{h+1}; z_h^k) + \ln\frac{\hat{p}(f_h, f_{h+1}|\mathcal{D}, \tilde{\pi})}{p_0(f_h, f_{h+1})}\right]}_{B_{h,t}}$$

$$+ \sum_{h=1}^{H}\underbrace{\mathbb{E}_{\mathcal{D}}\mathbb{E}_{\tilde{\pi} \sim P_t(\cdot|\mathcal{D})}\mathbb{E}_{f \sim \hat{p}(\cdot|\mathcal{D}, \tilde{\pi})}\left[\alpha\ln\mathbb{E}_{f_h' \sim p_0}\exp\left(-\gamma\sum_{k=1}^{K}\Delta L_{\tilde{\pi}}(f_h', f_{h+1}; z_h^k)\right) + (1 - \alpha)\ln\frac{\hat{p}(f_{h+1}|\mathcal{D}, \tilde{\pi})}{p_0(f_{h+1})}\right]}_{C_{h,t}}.$$

*Proof of Lemma E.1.* This is a simple adaptation of the decomposition in [Dann et al., 2021, Lemma 6]. □

We now control each term of the above decomposition of $Z$ separately – where a majority of these steps are where our technical arguments depart from those in Dann et al. [2021]. In particular, Lemma E.4, Lemma E.5, and Lemma E.7 are our *new* technical results.

**Bounding $A_t$.**

**Lemma E.2.** *We have*

$$A_t \geq \lambda \mathbb{E}_{\mathcal{D}} \mathbb{E}_{\tilde{\pi} \sim P_t(\cdot|\mathcal{D})} \mathbb{E}_{f \sim \hat{p}(\cdot|\mathcal{D}, \tilde{\pi})} \Delta f_1(s_1, \tilde{\pi}).$$

*Proof of Lemma E.2.* It simply follows from that:

$$\mathbb{E}_{\mathcal{D}} \mathbb{E}_{\tilde{\pi} \sim P_t(\cdot|\mathcal{D})} \mathbb{E}_{f \sim \hat{p}(\cdot|\mathcal{D}, \tilde{\pi})} \left[ (1 - 0.5\alpha) \ln \frac{\hat{p}(f_1|\mathcal{D}, \tilde{\pi})}{p_0(f_1)} \right] = (1 - 0.5\alpha) D_{\mathrm{KL}}[\hat{p}(\cdot|\mathcal{D}, \tilde{\pi}) \| p_0] \geq 0.$$

$\square$

**Bounding $B_{h,t}$.**

**Lemma E.3.** *For any $f, \tilde{\pi}$, $0 \leq \gamma \leq \frac{1}{72(e-2)b^2}$, and $h \in [H]$, we have*

$$\ln \mathbb{E}_{(s_{h+1}, r_h) \sim P_h(\cdot|s_h, a_h)} \exp\left(-2\gamma \Delta L_{\tilde{\pi}}(f_h, f_{h+1}; z_h)\right) \leq -2\gamma(1 - 72(e-2)\gamma b^2) \mathcal{E}_h^{\tilde{\pi}}(f_h, f_{h+1})(s_h, a_h)^2.$$

*Proof of Lemma E.3.* For simplicity, we write $\mathbb{E} = \mathbb{E}_{(s_{h+1}, r_h) \sim P_h(\cdot|s_h, a_h)}$. We have

$$\begin{aligned}
\ln \mathbb{E} \exp\left(-2\gamma \Delta L_{\tilde{\pi}}(f_h, f_{h+1}; z_h)\right) &\leq \mathbb{E} \exp\left(-2\gamma \Delta L_{\tilde{\pi}}(f_h, f_{h+1}; z_h)\right) - 1 \\
&\leq -2\gamma \mathbb{E} \Delta L_{\tilde{\pi}}(f_h, f_{h+1}; z_h) + (e-2)4\gamma^2 \mathbb{E} \Delta L_{\tilde{\pi}}(f_h, f_{h+1}; z_h)^2 \\
&\leq -2\gamma(1 - (e-2)2\gamma 36 b^2) \mathcal{E}_h^2(f_h, f_{h+1}, \tilde{\pi})(s_h, a_h)
\end{aligned}$$

where the first inequality uses $\ln x \leq x - 1, \forall x \geq 0$, the second inequality uses $e^x \leq 1 + x + (e - 2)x^2, \forall |x| \leq 1$ and $|2\gamma \mathbb{E} \Delta L_{\tilde{\pi}}(f_h, f_{h+1}; z_h)| \leq 18\gamma b^2 \leq 1$, the third inequality uses Lemma B.1 and $\gamma \leq \frac{1}{72(e-2)b^2}$. $\square$

**Lemma E.4.** *Define the random variable*

$$\xi_h^{\tilde{\pi}}(f_h, f_{h+1}; z_h) := -2\gamma \Delta L_{\tilde{\pi}}(f_h, f_{h+1}; z_h) - \ln \mathbb{E}_{(s_{h+1}, r_h) \sim P_h(\cdot|s_h, a_h)} \exp\left(-2\gamma \Delta L_{\tilde{\pi}}(f_h, f_{h+1}; z_h)\right).$$

*For any $\gamma \in [0, \frac{1}{144(e-2)b^2}]$, $t \in [0, \frac{1}{26\gamma b^2}]$, $\epsilon > 0$, $\delta > 0$, $T \in \mathbb{N}$ with probability at least $1 - \delta$, it holds uniformly over all $\tilde{\pi} \in \Pi^{soft}(T)$, $f_h \in \mathcal{F}_h$, $f_{h+1} \in \mathcal{F}_{h+1}$ that*

$$\sum_{k=1}^{K} \xi_h^{\tilde{\pi}}(f_h, f_{h+1}; z_h^k) \leq D + c \sum_{k=1}^{K} e_k^2,$$

*where*

$$\begin{cases}
D &:= 120\gamma b(b+2)K\epsilon + \frac{2d_{\mathcal{F}}(\epsilon) + d_{\Pi}(\epsilon, T) + \ln(1/\delta)}{t}, \\
c &:= 320 b^2 \gamma^2 (e-2)t, \\
e_k &:= \mathcal{E}_h^{\tilde{\pi}}(f_h, f_{h+1})(s_h^k, a_h^k).
\end{cases} \tag{13}$$

*Proof of Lemma E.4.* For simplicity, denote

$$\begin{cases}
u_k &:= \ln \mathbb{E}_{(s_{h+1}, r_h) \sim P_h(\cdot|s_h, a_h)} \exp\left(-2\gamma \Delta L_{\tilde{\pi}}(f_h, f_{h+1}; z_h^k)\right), \\
v_k &:= -2\gamma \Delta L_{\tilde{\pi}}(f_h, f_{h+1}; z_h^k), \\
w_k &:= v_k - u_k, \\
e_k &:= \mathcal{E}_h^{\tilde{\pi}}(f_h, f_{h+1})(s_h^k, a_h^k).
\end{cases} \tag{14}$$

We have

$$u_k \geq \mathbb{E}_{(s_{h+1}, r_h) \sim P_h(\cdot|s_h, a_h)} \ln \exp\left(-2\gamma \Delta L_{\tilde{\pi}}(f_h, f_{h+1}; z_h^k)\right) = -2\gamma e_k^2,$$

where the first inequality uses Jensen's inequality for concave function $\ln(\cdot)$ and the equality uses Lemma B.1. Now using Lemma E.3 with $\gamma \leq \frac{1}{144(e-2)b^2}$, we have

$$u_k \leq -\gamma e_k^2. \tag{15}$$

We also have $\mathbb{E}v_k = -2\gamma e_k^2$ by Lemma B.1. Thus, we have

$$|u_k| \leq 2\gamma e_k^2, \text{ and } \mathbb{E}[w_k] = -2\gamma e_k^2 - u_k \leq 0.$$

Hence, we have

$$\begin{aligned}
\mathbb{E}w_k^2 &= \mathbb{E}(v_k - u_k)^2 \\
&\leq 2\mathbb{E}(v_k^2 + u_k^2) \\
&\leq 288b^2\gamma^2 e_k^2 + 8\gamma^2 e_k^4 \\
&\leq 320b^2\gamma^2 e_k^2
\end{aligned}$$

where the first inequality uses Cauchy-Schwartz inequality, the second inequality uses Lemma B.1 and that $|u_k| \leq 2\gamma e_k^2$, and the last inequality uses that $|e_k| \leq 2b$. Also note that $|w_k| \leq |v_k| + |u_k| \leq 2\gamma(9b^2) + 2\gamma(4b^2) = 26\gamma b^2$. Thus, by Lemma B.2, for any $\delta > 0$, for any $t \in [0, \frac{1}{26\gamma b^2}]$, with probability at least $1 - \delta$, we have

$$\begin{aligned}
\sum_{k=1}^{K} w_k &\leq \sum_{k=1}^{K} \mathbb{E}w_k + (e-2)t \cdot \mathbb{E}\sum_{k=1}^{K} w_k^2 + \frac{\ln(1/\delta)}{t} \\
&\leq 320b^2\gamma^2(e-2)t\sum_{k=1}^{K} e_k^2 + \frac{\ln(1/\delta)}{t}.
\end{aligned}$$

We apply the discretization argument and the union bound to obtain that: For any $\delta > 0, \epsilon > 0$, $T \in \mathbb{N}$ it holds uniformly over all $\tilde{\pi} \in \Pi_h^{soft}(T), f_h \in \mathcal{F}_h, f_{h+1} \in \mathcal{F}_{h+1}$ that

$$\sum_{k=1}^{K} w_k \leq 120\gamma b(b+2)K\epsilon + 320b^2\gamma^2(e-2)t\sum_{k=1}^{K} e_k^2 + \frac{2d_{\mathcal{F}}(\epsilon) + d_{\Pi}(\epsilon, T) + \ln(1/\delta)}{t}.$$

$\square$

**Lemma E.5.** *For any $\gamma \in [0, \frac{1}{144(e-2)b^2}]$, $\epsilon > 0$, $\delta > 0$, we have*

$$\begin{aligned}
B_{h,t} &\geq 0.5\gamma\mathbb{E}_{\mathcal{D}}\mathbb{E}_{\tilde{\pi}\sim P_t(\cdot|\mathcal{D})}\mathbb{E}_{f\sim\hat{p}(\cdot|\mathcal{D},\tilde{\pi})}\left[\sum_{k=1}^{K} \mathcal{E}_h^{\tilde{\pi}}(f_h, f_{h+1})(s_h^k, a_h^k)^2\right] \\
&\geq -120\gamma b(b+2)K\epsilon - 640(e-2)b^2\gamma(2d_{\mathcal{F}}(\epsilon) + d_{\Pi}(\epsilon, T) + \ln(1/\delta)) - 26\gamma b^2 K\delta.
\end{aligned}$$

*Proof of Lemma E.5.* Define the random variables $u_k, v_k, w_k, e_k$ as Equation (14). Recall $D, c$ are defined in Equation (13) for any $t \in [0, \frac{1}{26\gamma b^2}]$. Define the event $E$ such that the inequality

$$\sum_{k=1}^{K} \xi_h^{\tilde{\pi}}(f_h, f_{h+1}; z_h^k) \leq \underbrace{320b^2\gamma^2(e-2)t}_{c}\sum_{k=1}^{K} e_k^2 + D, \tag{16}$$

holds uniformly over all $\tilde{\pi} \in \Pi^{soft}(T), f_h \in \mathcal{F}_h, f_{h+1} \in \mathcal{F}_{h+1}$. By Lemma E.4, we have

$$\Pr(E) \geq 1 - \delta, \text{ thus } \Pr(E^c) \leq \delta.$$

We have

$$\mathbb{E}_{\mathcal{D}}\mathbb{E}_{\tilde{\pi}\sim P_t(\cdot|\mathcal{D})}\mathbb{E}_{f\sim\hat{p}(\cdot|\mathcal{D},\tilde{\pi})}\left[\sum_{k=1}^{K}(-w_k + ce_k^2) + \ln\frac{\hat{p}(f_h, f_{h+1}|\mathcal{D},\tilde{\pi})}{p_0(f_h, f_{h+1})}\right]$$

$$\geq \mathbb{E}_{\mathcal{D}} \mathbb{E}_{\tilde{\pi} \sim P_t(\cdot|\mathcal{D})} \inf_p \mathbb{E}_{f \sim p} \left[ \sum_{k=1}^K (-w_k + ce_k^2) + \ln \frac{p(f_h, f_{h+1})}{p_0(f_h, f_{h+1})} \right]$$

$$= -\mathbb{E}_{\mathcal{D}} \mathbb{E}_{\tilde{\pi} \sim P_t(\cdot|\mathcal{D})} \ln \mathbb{E}_{f_h, f_{h+1} \sim p_0} \exp \left( \sum_{k=1}^K (w_k - ce_k^2) \right)$$

$$= -\mathbb{E}_{\mathcal{D}} 1\{E\} \mathbb{E}_{\tilde{\pi} \sim P_t(\cdot|\mathcal{D})} \ln \mathbb{E}_{f_h, f_{h+1} \sim p_0} \exp \left( \sum_{k=1}^K (w_k - ce_k^2) \right)$$

$$- \mathbb{E}_{\mathcal{D}} 1\{E^c\} \mathbb{E}_{\tilde{\pi} \sim P_t(\cdot|\mathcal{D})} \ln \mathbb{E}_{f_h, f_{h+1} \sim p_0} \exp \left( \sum_{k=1}^K (w_k - ce_k^2) \right)$$

$$\geq -\mathbb{E}_{\mathcal{D}} 1\{E\} \mathbb{E}_{\tilde{\pi} \sim P_t(\cdot|\mathcal{D})} \ln \mathbb{E}_{f_h, f_{h+1} \sim p_0} \exp \left( \sum_{k=1}^K (w_k - ce_k^2) \right) - 26\gamma b^2 K \delta$$

$$\geq -D - 26\gamma b^2 K \delta, \tag{17}$$

where the first equality uses Lemma G.3, the second inequality uses that $\Pr(E^c) \leq \delta$ and $\sum_{k=1}^K (w_k - ce_k^2) \leq \sum_{k=1}^K w_k \leq 26\gamma b^2 K$ and the last inequality uses Equation (16). Thus, using the same notations as Lemma Lemma E.4, we have

$$B_{h,t} = \mathbb{E}_{\mathcal{D}} \mathbb{E}_{\tilde{\pi} \sim P_t(\cdot|\mathcal{D})} \mathbb{E}_{f \sim \hat{p}(\cdot|\mathcal{D}, \tilde{\pi})} \left[ -\sum_{k=1}^K v_k + \ln \frac{\hat{p}(f_h, f_{h+1}|\mathcal{D}, \tilde{\pi})}{p_0(f_h, f_{h+1})} \right]$$

$$= \mathbb{E}_{\mathcal{D}} \mathbb{E}_{\tilde{\pi} \sim P_t(\cdot|\mathcal{D})} \mathbb{E}_{f \sim \hat{p}(\cdot|\mathcal{D}, \tilde{\pi})} \left[ \sum_{k=1}^K (-w_k + ce_k^2) + \ln \frac{\hat{p}(f_h, f_{h+1}|\mathcal{D}, \tilde{\pi})}{p_0(f_h, f_{h+1})} \right]$$

$$+ \mathbb{E}_{\mathcal{D}} \mathbb{E}_{\tilde{\pi} \sim P_t(\cdot|\mathcal{D})} \mathbb{E}_{f \sim \hat{p}(\cdot|\mathcal{D}, \tilde{\pi})} \left[ \sum_{k=1}^K (-u_k - ce_k^2) \right]$$

$$\geq -D - 26\gamma b^2 K \delta + (\gamma - c) \mathbb{E}_{\mathcal{D}} \mathbb{E}_{\tilde{\pi} \sim P_t(\cdot|\mathcal{D})} \mathbb{E}_{f \sim \hat{p}(\cdot|\mathcal{D}, \tilde{\pi})} \left[ \sum_{k=1}^K e_k^2 \right]$$

where the inequality uses Equation (17) and Equation (15). Finally, setting

$$t = \frac{1}{640 b^2 (e-2)\gamma} < \frac{1}{13 b^2 \gamma}$$

completes our proof.

$\square$

**From squared Bellman errors to *in-expectation* squared Bellman errors and fixing a non-rigorous argument of Dann et al. [2021].** Lemma E.5 only bounds $B_h$ with the squared Bellman errors $\sum_{k=1}^K \mathcal{E}_h^{\tilde{\pi}}(f_h, f_{h+1})(s_h^k, a_h^k)^2$ while the *in-expectation* squared Bellman errors $\sum_{k=1}^K \mathbb{E}_{\mu^k}[\mathcal{E}_h^{\tilde{\pi}}(f_h, f_{h+1})(s_h, a_h)^2]$ are what we need for showing Proposition 2. There is no an immediate path to go from the squared Bellman error to the in-expectation squared Bellman errors as the order of $\mathbb{E}_{\mathcal{D}}$ and $\mathbb{E}_{\tilde{\pi} \sim P_t(\cdot|\mathcal{D})} \mathbb{E}_{f \sim \hat{p}(\cdot|\mathcal{D}, \tilde{\pi})}$ are **not** exchangeable, i.e.,

$$\mathbb{E}_{\mathcal{D}} \mathbb{E}_{\tilde{\pi} \sim P_t(\cdot|\mathcal{D})} \mathbb{E}_{f \sim \hat{p}(\cdot|\mathcal{D}, \tilde{\pi})} \left[ \sum_{k=1}^K \mathcal{E}_h^{\tilde{\pi}}(f_h, f_{h+1})(s_h^k, a_h^k)^2 \right]$$

$$\neq \mathbb{E}_{\mathcal{D}} \mathbb{E}_{\tilde{\pi} \sim P_t(\cdot|\mathcal{D})} \mathbb{E}_{f \sim \hat{p}(\cdot|\mathcal{D}, \tilde{\pi})} \left[ \sum_{k=1}^K \mathbb{E}_{\mu^k}[\mathcal{E}_h^{\tilde{\pi}}(f_h, f_{h+1})(s_h, a_h)^2] \right]. \tag{18}$$

A similar caveat arises in the online setting in Dann et al. [2021] as well. In particular, a non-rigorous argument of [Dann et al., 2021, Lemma 8] is that they conclude (an online analogue of) the LHS of Equation (18) is equal to (an online analogue of) its RHS.

To fix this issue without ultimately incurring a sub-optimality rate that is slower than $1/\sqrt{K}$, we need to change the squared Bellman error into the in-expectation squared Bellman error, up to some estimation error that scales faster than $K^\alpha$ for any $\alpha > 0$. Note that, a standard Azuma–Hoeffding inquality (and the union bound) give an estimation error that scales with $K^{1/2}$. To achieve the logarithmic dependence on $K$, the following lemma exploits the *non-negativity* of the squared Bellman error and uses the localization argument of Bartlett et al. [2005] to obtain an estimation error rate that scales polylogarithmic with $K$.

**Lemma E.6** (Improved online-to-batch argument for non-negative R.V.s [Nguyen-Tang et al., 2023]).
*Let $\{X_k\}$ be any real-valued stochastic process adapted to the filtration $\{\mathcal{F}_k\}$, i.e. $X_k$ is $\mathcal{F}_k$-measurable. Suppose that for any $k$, $X_k \in [0, H]$ almost surely for some $H > 0$. For any $K > 0$, with probability at least $1 - \delta$, we have:*

$$\sum_{k=1}^{K} \mathbb{E}\left[X_k | \mathcal{F}_{k-1}\right] \leq 2 \sum_{k=1}^{K} X_k + \frac{16}{3} H \log(\log_2(KH)/\delta) + 2.$$

With Lemma E.6, we now actually make a connection from the squared Bellman error to the *in-expectation* squared Bellman error in the following lemma, which incorporates the uniform convergence argument into the posterior sampling in the same spirit with our earlier argument in Section E.3.

**Lemma E.7.** *For any $\delta, \epsilon > 0$ and $T \in \mathbb{N}$, any $t \in [T]$, we have*

$$\mathbb{E}_{\mathcal{D}}\mathbb{E}_{\tilde{\pi} \sim P_t(\cdot|\mathcal{D})}\mathbb{E}_{f \sim \hat{p}(\cdot|\mathcal{D},\tilde{\pi})} \sum_{k=1}^{K} \mathcal{E}_h^{\tilde{\pi}}(f_h, f_{h+1})(s_h^k, a_h^k)^2$$

$$\geq 0.5 \mathbb{E}_{\mathcal{D}}\mathbb{E}_{\tilde{\pi} \sim P_t(\cdot|\mathcal{D})}\mathbb{E}_{f \sim \hat{p}(\cdot|\mathcal{D},\tilde{\pi})} \sum_{k=1}^{K} \mathbb{E}_{\mu^k}\left[\mathcal{E}_h^{\tilde{\pi}}(f_h, f_{h+1})(s_h, a_h)^2\right]$$

$$- b(b+2)K\epsilon - \frac{32}{3}b^2\left(2d_{\mathcal{F}}(\epsilon) + d_\Pi(\epsilon, T) + \ln\frac{\ln 4Kb^2}{\delta}\right) - 1 - 2Kb^2\delta.$$

*Proof of Lemma E.7.* For simplicity, we denote $X(f, \mathcal{D}) := \sum_{k=1}^{K} \mathcal{E}_h^{\tilde{\pi}}(f_h, f_{h+1})(s_h^k, a_h^k)^2$ and $X(f) := \mathbb{E}_{\mathcal{D}}[X(f, \mathcal{D})]$, and $\Delta := 8bK\epsilon + \frac{64}{3}b^2\left(2d_{\mathcal{F}}(\epsilon) + d_\Pi(\epsilon, T) + \frac{\ln\ln 4Kb^2}{\delta}\right) + 2$. We define the event:

$$E = \left\{\mathcal{D} : X(f) \leq 2X(f, \mathcal{D}) + \Delta, \forall f_h \in \mathcal{F}_h, f_{h+1} \in \mathcal{F}_{h+1}, \tilde{\pi} \in \Pi_h^{soft}(T)\right\}.$$

Due to the non-negativity of $\mathcal{E}_h^{\tilde{\pi}}(f_h, f_{h+1})(s, a)^2$, Lemma E.6 and the union bound, we have

$$\Pr(E) \geq 1 - \delta \text{ and } \Pr(E^c) \leq \delta.$$

We have

$$2\mathbb{E}_{\mathcal{D}}\mathbb{E}_{\tilde{\pi} \sim P_t(\cdot|\mathcal{D})}\mathbb{E}_{f \sim \hat{p}(\cdot|\mathcal{D},\tilde{\pi})} X(f, \mathcal{D})$$

$$= 2\mathbb{E}_{\mathcal{D}}\mathbb{E}_{\tilde{\pi} \sim P_t(\cdot|\mathcal{D})}\mathbb{E}_{f \sim \hat{p}(\cdot|\mathcal{D},\tilde{\pi})} X(f, \mathcal{D})1\{E\} + 2\mathbb{E}_{\mathcal{D}}\mathbb{E}_{\tilde{\pi} \sim P_t(\cdot|\mathcal{D})}\mathbb{E}_{f \sim \hat{p}(\cdot|\mathcal{D},\tilde{\pi})} X(f, \mathcal{D})1\{E^c\}$$

$$\geq 2\mathbb{E}_{\mathcal{D}}\mathbb{E}_{\tilde{\pi} \sim P_t(\cdot|\mathcal{D})}\mathbb{E}_{f \sim \hat{p}(\cdot|\mathcal{D},\tilde{\pi})} X(f, \mathcal{D})1\{E\}$$

$$\geq \mathbb{E}_{\mathcal{D}}1\{E\}\mathbb{E}_{\tilde{\pi} \sim P_t(\cdot|\mathcal{D})}\mathbb{E}_{f \sim \hat{p}(\cdot|\mathcal{D},\tilde{\pi})}(X(f) - \Delta)$$

$$= \mathbb{E}_{\mathcal{D}}1\{E\}\mathbb{E}_{\tilde{\pi} \sim P_t(\cdot|\mathcal{D})}\mathbb{E}_{f \sim \hat{p}(\cdot|\mathcal{D},\tilde{\pi})} X(f) - \Delta \Pr(E)$$

$$\geq \mathbb{E}_{\mathcal{D}}1\{E\}\mathbb{E}_{\tilde{\pi} \sim P_t(\cdot|\mathcal{D})}\mathbb{E}_{f \sim \hat{p}(\cdot|\mathcal{D},\tilde{\pi})} X(f) - \Delta$$

$$= \mathbb{E}_{\mathcal{D}}\mathbb{E}_{\tilde{\pi} \sim P_t(\cdot|\mathcal{D})}\mathbb{E}_{f \sim \hat{p}(\cdot|\mathcal{D},\tilde{\pi})} X(f) - \Delta - \mathbb{E}_{\mathcal{D}}1\{E^c\}\mathbb{E}_{\tilde{\pi} \sim P_t(\cdot|\mathcal{D})}\mathbb{E}_{f \sim \hat{p}(\cdot|\mathcal{D},\tilde{\pi})} X(f)$$

$$\geq \mathbb{E}_{\mathcal{D}}\mathbb{E}_{\tilde{\pi} \sim P_t(\cdot|\mathcal{D})}\mathbb{E}_{f \sim \hat{p}(\cdot|\mathcal{D},\tilde{\pi})} X(f) - \Delta - \mathbb{E}_{\mathcal{D}}1\{E^c\}\mathbb{E}_{f \sim \hat{p}(\cdot|\mathcal{D},\tilde{\pi})} 4Kb^2$$

$$= \mathbb{E}_{\mathcal{D}}\mathbb{E}_{f \sim \hat{p}(\cdot|\mathcal{D},\tilde{\pi})} X(f) - \Delta - 4Kb^2 \Pr(E^c)$$

$$\geq \mathbb{E}_{\mathcal{D}}\mathbb{E}_{f \sim \hat{p}(\cdot|\mathcal{D},\tilde{\pi})} X(f) - \Delta - 4Kb^2\delta$$

where the fourth inequality uses $|X(f)| \geq 4Kb^2$ and the last inequality uses $\Pr(E^c) \leq \delta$. $\qquad\square$

**Bounding $C_{h,t}$.**

**Lemma E.8.** *For any $\epsilon > 0$ and $T \in \mathbb{N}$, we have*

$$C_{h,t} \geq - \max_{\tilde{\pi} \in \Pi_h^{soft}(T)} \kappa_h(\alpha, \epsilon, \tilde{\pi}) - \gamma\alpha 6bK\epsilon.$$

*where $\kappa_h(\alpha, \epsilon, \tilde{\pi})$ is defined in Equation (11).*

*Proof of Lemma E.8.* We have

$$C_{h,t} = \mathbb{E}_{\mathcal{D}}\mathbb{E}_{\tilde{\pi} \sim P_t(\cdot|\mathcal{D})}\mathbb{E}_{f \sim \hat{p}(\cdot|\mathcal{D},\tilde{\pi})}\left[\alpha \ln \mathbb{E}_{f_h' \sim p_0} \exp\left(-\gamma \sum_{k=1}^{K} \Delta L_{\tilde{\pi}}(f_h', f_{h+1}; z_h^k)\right) + (1-\alpha)\ln \frac{\hat{p}(f_{h+1}|\mathcal{D}, \tilde{\pi})}{p_0(f_{h+1})}\right]$$

$$= (1-\alpha)\mathbb{E}_{\mathcal{D}}\mathbb{E}_{\tilde{\pi} \sim P_t(\cdot|\mathcal{D})}\mathbb{E}_{f \sim \hat{p}(\cdot|\mathcal{D},\tilde{\pi})}\left[\frac{\alpha}{1-\alpha} \ln \mathbb{E}_{f_h' \sim p_0} \exp\left(-\gamma \sum_{k=1}^{K} \Delta L_{\tilde{\pi}}(f_h', f_{h+1}; z_h^k)\right) + \ln \frac{\hat{p}(f_{h+1}|\mathcal{D}, \tilde{\pi})}{p_0(f_{h+1})}\right]$$

$$\geq -(1-\alpha)\mathbb{E}_{\mathcal{D}}\mathbb{E}_{\tilde{\pi} \sim P_t(\cdot|\mathcal{D})} \ln \mathbb{E}_{f_{h+1} \sim p_0}\left(\mathbb{E}_{f_h' \sim p_0} \exp\left(-\gamma \sum_{k=1}^{K} \Delta L_{\tilde{\pi}}(f_h', f_{h+1}; z_h^k)\right)\right)^{\frac{-\alpha}{1-\alpha}}$$

$$\geq - \max_{\tilde{\pi} \in \Pi^{soft}(T)} \kappa_h(\alpha, \epsilon, \tilde{\pi}) - \gamma\alpha 6bK\epsilon.$$

where the first inequality uses Lemma G.3 and the last inequality uses the following inequalities: For any $f_h \in \mathcal{F}_h(\epsilon, f_{h+1}, \tilde{\pi})$, we have

$$|\Delta L_{\tilde{\pi}}(f_h, f_{h+1}; z_h)| \leq 6b|\mathcal{E}_h^{\tilde{\pi}}(f_h, f_{h+1})| \leq 6b\epsilon; \text{ thus}$$

$$\mathbb{E}_{f_h' \sim p_0} \exp\left(-\gamma \sum_{k=1}^{K} \Delta L_{\tilde{\pi}}(f_h', f_{h+1}; z_h)\right) \geq p_{0,h}(\mathcal{F}_h^{\tilde{\pi}}(\epsilon, f_{h+1})) \cdot \exp(-\gamma 6bK\epsilon).$$

$\square$

We are now ready to state the complete form of the lower bound of $Z$.

**Proposition 3.** *For any $\gamma \in [0, \frac{1}{144(e-2)b^2}]$, $\epsilon > 0$, $\delta > 0$, and any $t \in [T]$, we have,*

$$Z \geq \lambda\mathbb{E}_{\mathcal{D}}\mathbb{E}_{\tilde{\pi} \sim P_t(\cdot|\mathcal{D})}\mathbb{E}_{f \sim \hat{p}(\cdot|\mathcal{D},\tilde{\pi})}\Delta f_1(s_1, \tilde{\pi})$$

$$+ 0.125\alpha\gamma \sum_{h=1}^{H} \mathbb{E}_{\mathcal{D}}\mathbb{E}_{\tilde{\pi} \sim P_t(\cdot|\mathcal{D})}\mathbb{E}_{f \sim \hat{p}(\cdot|\mathcal{D},\tilde{\pi})} \sum_{k=1}^{K} \mathbb{E}_{\mu^k}\left[\mathcal{E}_h^{\tilde{\pi}}(f_h, f_{h+1})(s_h, a_h)^2\right]$$

$$- 0.5\alpha H\left(120\gamma b(b+2)K\epsilon + 640(e-2)\gamma b^2\left(2d_{\mathcal{F}}(\epsilon) + d_{\Pi}(\epsilon, T) + \ln(1/\delta)\right)\right) - 13\alpha\gamma b^2 KH\delta$$

$$- 0.25\alpha\gamma H\left(b(b+2)K\epsilon + \frac{32}{3}b^2\left(2d_{\mathcal{F}}(\epsilon) + d_{\Pi}(\epsilon, T) + \ln\frac{4Kb^2}{\delta}\right) + 1 + 2Kb^2\delta\right)$$

$$- \sum_{h=1}^{H} \max_{\tilde{\pi} \in \Pi^{soft}(T)} \kappa_h(\alpha, \epsilon, \tilde{\pi}) - \gamma\alpha 6bKH\epsilon.$$

*Proof of Proposition 3.* Using Lemma E.1, it suffices to bound terms $A$, $B_h$, and $C_h$ defined in Lemma E.1. For this purpose, we use

- Lemma E.2: To bound $A_t$,
- Lemma E.5 and Lemma E.7: To bound $B_{h,t}$,
- Lemma E.8: To bound $C_{h,t}$.

The result is then simply a direct combination of the above lemmas. $\square$

### E.3.2 Upper-bounding log-partition function

In this appendix, we upper bound the log-partition function $Z$. While we follow the proof flow in Dann et al. [2021], due to the statistical dependence in the actor-critic framework of our algorithm, we require different technical arguments to establish this result. In particular, Lemma E.9 and Lemma E.10 are our *new* technical lemmas.

**Proposition 4.** *For any $\epsilon, \delta > 0$, $\gamma > 0$, and $t \in [T]$, we have*

$$Z_t \leq \lambda\epsilon - \inf_{\tilde{\pi} \in \Pi^{soft}(T)} \sum_{h=1}^{H} \ln p_0(\mathcal{F}_h(\epsilon; \tilde{\pi})) + 4\gamma \left( \alpha + \frac{3(e-2)}{\alpha} \right) HK\epsilon^2$$

$$+ 60\alpha\gamma b(b+2)KH\epsilon + \alpha\gamma b^2 H \left( 13 + 36(e-2) \right) \left( 2d_{\mathcal{F}}(\epsilon) + d_{\Pi}(\epsilon, T) + \ln(1/\delta) \right) + 18\alpha\gamma KHb^2\delta.$$

*where recall that $Z_t$ is defined in Equation (12).*

*Proof of Proposition 4.* We have

$$Z_t = \mathbb{E}_{\mathcal{D}} \mathbb{E}_{\tilde{\pi} \sim P_t(\cdot|\mathcal{D})} \mathbb{E}_{f \sim \hat{p}(\cdot|\mathcal{D}, \tilde{\pi})} \left[ \widehat{\Phi}(f, \tilde{\pi}; \mathcal{D}) + \lambda\Delta f_1(s_1, \tilde{\pi}) + \ln \hat{p}(f|\mathcal{D}, \tilde{\pi}) \right]$$

$$= \mathbb{E}_{\mathcal{D}} \mathbb{E}_{\tilde{\pi} \sim P_t(\cdot|\mathcal{D})} \inf_p \mathbb{E}_{f \sim p} \left[ \widehat{\Phi}(f, \tilde{\pi}; \mathcal{D}) + \lambda\Delta f_1(s_1, \tilde{\pi}) + \ln p(f) \right]$$

$$\leq \mathbb{E}_{\mathcal{D}} \mathbb{E}_{\tilde{\pi} \sim P_t(\cdot|\mathcal{D})} \inf_p \mathbb{E}_{f \sim p} \left[ \ln \frac{p(f)}{p_0(f)} + \alpha\gamma \sum_{h=1}^{H} \sum_{k=1}^{K} \Delta L_{\tilde{\pi}}(f_h, f_{h+1}; z_h^k) + \lambda\Delta f_1(s_1, \tilde{\pi}) \right]$$

$$+ \mathbb{E}_{\mathcal{D}} \mathbb{E}_{\tilde{\pi} \sim P_t(\cdot|\mathcal{D})} \inf_p \mathbb{E}_{f \sim p} \left[ \alpha \sum_{h=1}^{H} \ln \mathbb{E}_{\tilde{f}_h \sim p_0} \exp \left( -\gamma \sum_{k=1}^{K} \Delta L_{\tilde{\pi}}(\tilde{f}_h, f_{h+1}; z_h^k) \right) \right]$$

where the second equality uses the fact that $D_{\mathrm{KL}}[p\|\hat{p}] \geq 0$ with the minimum occurring at $p = \hat{p}$ and the inequality uses the triangle inequality. The first term is bounded by Lemma E.9 and Lemma E.11, and the second term is bounded by Lemma E.10 $\qquad\square$

It remains to state and prove Lemma E.9, Lemma E.11 and Lemma E.10.

The following lemma bounds the in-expectation of the loss $\Delta L_{\tilde{\pi}}$ by the in-expectation of the squared Bellman error.

**Lemma E.9.** *For any distribution $p$ over $\mathcal{F}$, for any $\epsilon, \delta > 0$, $\gamma > 0$, any $t \in [T]$, we have*

$$\mathbb{E}_{\mathcal{D}} \mathbb{E}_{\tilde{\pi} \sim P_t(\cdot|\mathcal{D})} \mathbb{E}_{f \sim p} \left[ \alpha\gamma \sum_{k=1}^{K} \Delta L_{\tilde{\pi}}(f_h, f_{h+1}; z_h^k) \right]$$

$$\leq \gamma \left( \alpha + \frac{3(e-2)}{\alpha} \right) \mathbb{E}_{\mathcal{D}} \mathbb{E}_{\tilde{\pi} \sim P_t(\cdot|\mathcal{D})} \mathbb{E}_{f \sim p} \left[ \sum_{k=1}^{K} \mathcal{E}_h^{\tilde{\pi}}(f_h, f_{h+1})(s_h^k, a_h^k)^2 \right]$$

$$+ 30\alpha\gamma b(b+2)K\epsilon + 13\alpha\gamma b^2 \left( 2d_{\mathcal{F}}(\epsilon) + d_{\Pi}(\epsilon, T) + \ln(1/\delta) \right) + 9\alpha\gamma Kb^2\delta.$$

*Proof of Lemma E.9.* For simplicity, define

$$x_k := \alpha\gamma\Delta L_{\tilde{\pi}}(f_h, f_{h+1}; z_h^k),$$
$$e_k := \mathcal{E}_h^{\tilde{\pi}}(f_h, f_{h+1})(s_h^k, a_h^k).$$

By Lemma B.1, we have

$$\mathbb{E}[x_k] = \alpha\gamma e_k^2,$$
$$\mathbb{E}[x_k^2] \leq 36b^2\alpha^2\gamma^2 e_k^2.$$

Thus, by Lemma B.2, for any $\delta > 0$, with probability at least $1 - \delta$, for any $t \in [0, \frac{1}{13\alpha\gamma b^2}]$ we have

$$\sum_{k=1}^{K} x_k \leq \sum_{k=1}^{K} \mathbb{E}[x_k] + t(e-2) \sum_{k=1}^{K} \mathbb{E}[x_k^2] + \frac{\ln(1/\delta)}{t}$$

$$\leq \left(\alpha\gamma + t(e-2)36b^2\gamma^2\right)\sum_{k=1}^{K} e_k^2 + \frac{\ln(1/\delta)}{t}.$$

Using the discretization argument and the union bound, we have that: For any $\epsilon > 0, \delta > 0$, we have

$$\Pr(E) \geq 1 - \delta, \text{ thus } \Pr(E^c) \leq \delta,$$

where $E$ denotes that event that for any $t \in [0, \frac{1}{13\alpha\gamma b^2}]$,

$$\sum_{k=1}^{K} x_k \leq 30\alpha\gamma b(b+2)K\epsilon + \left(\alpha\gamma + t(e-2)36b^2\alpha^2\gamma^2\right)\sum_{k=1}^{K} e_k^2 + \frac{2d_{\mathcal{F}}(\epsilon) + d_\Pi(\epsilon, T) + \ln(1/\delta)}{t},$$

any $f_h \in \mathcal{F}_h, f_{h+1} \in \mathcal{F}_{h+1}, \tilde{\pi} \in \Pi_h^{soft}(T)$. Thus, we have

$$\mathbb{E}_{\mathcal{D}}\mathbb{E}_{\tilde{\pi}\sim P_t(\cdot|\mathcal{D})}\mathbb{E}_{f\sim p}\left[\sum_{k=1}^{K} x_k\right] = \mathbb{E}_{\mathcal{D}}1\{E\}\mathbb{E}_{\tilde{\pi}\sim P_t(\cdot|\mathcal{D})}\mathbb{E}_{f\sim p}\left[\sum_{k=1}^{K} x_k\right] + \mathbb{E}_{\mathcal{D}}1\{E^c\}\mathbb{E}_{\tilde{\pi}\sim P_t(\cdot|\mathcal{D})}\mathbb{E}_{f\sim p}\left[\sum_{k=1}^{K} x_k\right]$$

$$\leq \mathbb{E}_{\mathcal{D}}1\{E\}\mathbb{E}_{\tilde{\pi}\sim P_t(\cdot|\mathcal{D})}\mathbb{E}_{f\sim p}\Bigg[30\alpha\gamma b(b+2)K\epsilon$$

$$+ \left(\alpha\gamma + t(e-2)36\alpha^2 b^2\gamma^2\right)\sum_{k=1}^{K} e_k^2 + \frac{2d_{\mathcal{F}}(\epsilon) + d_\Pi(\epsilon, T) + \ln(1/\delta)}{t}\Bigg] + 9\alpha\gamma K b^2\delta$$

$$\leq \mathbb{E}_{\mathcal{D}}\mathbb{E}_{\tilde{\pi}\sim P_t(\cdot|\mathcal{D})}\mathbb{E}_{f\sim p}\Bigg[30b(b+2)K\epsilon$$

$$+ \left(\alpha\gamma + t(e-2)36\alpha^2 b^2\gamma^2\right)\sum_{k=1}^{K} e_k^2 + \frac{2d_{\mathcal{F}}(\epsilon) + d_\Pi(\epsilon, T) + \ln(1/\delta)}{t}\Bigg] + 9\alpha\gamma K b^2\delta.$$

Picking $t = \frac{1}{13\alpha\gamma b^2}$ completes the proof. $\qquad\square$

The following lemma bounds the in-expectation negation of the loss proxy $\Delta L_{\tilde{\pi}}$.

**Lemma E.10.** *For any $\delta > 0, \epsilon > 0, \gamma > 0$, any $\tilde{f}_h \in \mathcal{F}_h$, any $t \in [T]$, and any distribution $p$ over $\mathcal{F}$, we have*

$$\mathbb{E}_{\mathcal{D}}\mathbb{E}_{\tilde{\pi}\sim P_t(\cdot|\mathcal{D})}\mathbb{E}_{f\sim p}\left[-\gamma\sum_{k=1}^{K}\Delta L_{\tilde{\pi}}(\tilde{f}_h, f_{h+1}; z_h^k)\right] \leq 36(e-2)\gamma b^2\left(d_{\mathcal{F}}(\epsilon) + d_\Pi(\epsilon, T) + \ln(1/\delta)\right)$$

$$+ 9\gamma K b^2\delta + 30\gamma b(b+2)K\epsilon.$$

*Proof of Lemma E.10.* For simplicity, define

$$y_k := -\gamma\Delta L_{\tilde{\pi}}(f_h, f_{h+1}; z_h^k),$$
$$e_k := \mathcal{E}_h^{\tilde{\pi}}(f_h, f_{h+1})(s_h^k, a_h^k).$$

By Lemma B.1, we have

$$\mathbb{E}[y_k] = -\gamma e_k^2,$$
$$\mathbb{E}[y_k^2] \leq 36b^2\gamma^2 e_k^2.$$

Thus, by Lemma B.2, for any $\delta > 0$, with probability at least $1 - \delta$, for any $t \in [0, \frac{1}{13\gamma b^2}]$ we have

$$\sum_{k=1}^{K} y_k \leq \sum_{k=1}^{K}\mathbb{E}[y_k] + t(e-2)\sum_{k=1}^{K}\mathbb{E}[y_k^2] + \frac{\ln(1/\delta)}{t}$$

$$\leq -\gamma\left(1 - 36t(e-2)b^2\gamma\right)\sum_{k=1}^{K} e_k^2 + \frac{\ln(1/\delta)}{t}.$$

Setting $t = \frac{1}{36(e-2)b^2\gamma} < \frac{1}{13\gamma b^2}$ in the above inequality, we obtain

$$\sum_{k=1}^{k} y_k \le 36(e-2)b^2\gamma \ln(1/\delta).$$

Using the discretization argument and the union bound, we have that: For any $\epsilon > 0, \delta > 0$, we have

$$\Pr(E) \ge 1 - \delta, \text{ thus } \Pr(E^c) \le \delta,$$

where $E$ denotes that event ,

$$\sum_{k=1}^{K} y_k \le 30\gamma b(b+2)K\epsilon + 36(e-2)\gamma b^2 \left( d_{\mathcal{F}}(\epsilon) + d_\Pi(\epsilon, T) + \ln(1/\delta) \right),$$

any $f_{h+1} \in \mathcal{F}_{h+1}, \tilde{\pi} \in \Pi_h^{soft}(T)$. Thus, we have

$$\mathbb{E}_{\mathcal{D}}\mathbb{E}_{\tilde{\pi}\sim P_t(\cdot|\mathcal{D})}\mathbb{E}_{f\sim p}\left[\sum_{k=1}^{K} y_k\right] = \mathbb{E}_{\mathcal{D}}1\{E\}\mathbb{E}_{\tilde{\pi}\sim P_t(\cdot|\mathcal{D})}\mathbb{E}_{f\sim p}\left[\sum_{k=1}^{K} y_k\right] + \mathbb{E}_{\mathcal{D}}1\{E^c\}\mathbb{E}_{\tilde{\pi}\sim P_t(\cdot|\mathcal{D})}\mathbb{E}_{f\sim p}\left[\sum_{k=1}^{K} y_k\right]$$

$$\le 30\gamma b(b+2)K\epsilon + 36(e-2)\gamma b^2 \left( d_{\mathcal{F}}(\epsilon) + d_\Pi(\epsilon, T) + \ln(1/\delta) \right) + 9\gamma K b^2\delta.$$

$\square$

The following lemma bounds the in-expectation squared Bellman errors with the regularization term and the data distribution term, under the infimum realization of the data distribution $p$.

**Lemma E.11.** *For any $\epsilon > 0, \beta \ge 0$, any $t \in [T]$, we have*

$$\mathbb{E}_{\mathcal{D}}\mathbb{E}_{\tilde{\pi}\sim P_t(\cdot|\mathcal{D})}\inf_p \mathbb{E}_{f\sim p}\left[\lambda\Delta f_1(s_1, \tilde{\pi}) + \ln\frac{p(f)}{p_0(f)} + \beta\sum_{h=1}^{H}\sum_{k=1}^{K}\mathcal{E}_h^{\tilde{\pi}}(f_h, f_{h+1})(s_h^k, a_h^k)^2\right]$$

$$\le \lambda\epsilon - \inf_{\tilde{\pi}\in\Pi^{soft}(T)}\ln p_0(\mathcal{F}_h(\epsilon; Q_h^{\tilde{\pi}})) + 4\beta HK\epsilon^2.$$

*where recall that $\mathcal{F}_h(\epsilon; f_{h+1})$ is defined in [Definition 2](#).*

*Proof of [Lemma E.11](#).* For any $f \in \mathcal{F}(\epsilon; Q^{\tilde{\pi}})$, we have

$$\|f_h - Q_h^{\tilde{\pi}}\|_\infty \le \epsilon, \forall h.$$

Thus, we have

$$|\mathcal{E}_h^{\tilde{\pi}}(f_h, f_{h+1})(s, a)| \le \|\mathbb{T}_h^{\tilde{\pi}}f_h - f_{h+1}\|_\infty = \|\mathbb{T}_h^{\tilde{\pi}}f_h - \mathbb{T}_h^{\tilde{\pi}}Q_h^{\tilde{\pi}} - f_{h+1} + Q_{h+1}^{\tilde{\pi}}\|_\infty$$

$$\le \|\mathbb{T}_h^{\tilde{\pi}}f_h - \mathbb{T}_h^{\tilde{\pi}}Q_h^{\tilde{\pi}}\|_\infty + \|f_{h+1} - Q_{h+1}^{\tilde{\pi}}\|_\infty$$

$$\le 2\epsilon.$$

Thus, by choosing

$$p(f) = \frac{p_0(f)1\{f \in \mathcal{F}(\epsilon; \tilde{\pi})\}}{p_0(\mathcal{F}(\epsilon; \tilde{\pi}))},$$

we have

$$\inf_p \mathbb{E}_{f\sim p}\left[\lambda\Delta f_1(s_1, \tilde{\pi}) + \ln\frac{p(f)}{p_0(f)} + \beta\sum_{h=1}^{H}\sum_{k=1}^{K}\mathcal{E}_h^{\tilde{\pi}}(f_h, f_{h+1})(s_h^k, a_h^k)^2\right]$$

$$\le \lambda\epsilon - \ln p_0(\mathcal{F}_h(\epsilon; \tilde{\pi})) + 4\beta HK\epsilon^2.$$

$\square$

We by now have everything needed to prove [Proposition 2](#).

### E.3.3 Proof of Proposition 2

*Proof of Proposition 2.* By Proposition 3, we have

$$Z_t \geq \lambda \mathbb{E}_{\mathcal{D}} \mathbb{E}_{\tilde{\pi} \sim P_t(\cdot | \mathcal{D})} \mathbb{E}_{f \sim \hat{p}(\cdot | \mathcal{D}, \tilde{\pi})} \Delta f_1(s_1, \tilde{\pi})$$

$$+ 0.125 \alpha \gamma \sum_{h=1}^{H} \mathbb{E}_{\mathcal{D}} \mathbb{E}_{\tilde{\pi} \sim P_t(\cdot | \mathcal{D})} \mathbb{E}_{f \sim \hat{p}(\cdot | \mathcal{D}, \tilde{\pi})} \sum_{k=1}^{K} \mathbb{E}_{\mu^k} \left[ \mathcal{E}_h^{\tilde{\pi}}(f_h, f_{h+1})(s_h, a_h)^2 \right]$$

$$- 0.5 \alpha H \left( 120 \gamma b(b+2) K \epsilon + 640(e-2) \gamma b^2 \left( 2 d_{\mathcal{F}}(\epsilon) + d_{\Pi}(\epsilon, T) + \ln(1/\delta) \right) \right) - 13 \alpha \gamma b^2 K H \delta$$

$$- 0.25 \alpha \gamma H \left( b(b+2) K \epsilon + \frac{32}{3} b^2 \left( 2 d_{\mathcal{F}}(\epsilon) + d_{\Pi}(\epsilon, T) + \frac{\ln \ln 4 K b^2}{\delta} \right) + 1 + 2 K b^2 \delta \right)$$

$$- \sum_{h=1}^{H} \max_{\tilde{\pi} \in \Pi^{soft}(T)} \kappa_h(\alpha, \epsilon, \tilde{\pi}) - \gamma \alpha 6 b K H \epsilon.$$

By Proposition 4, we have

$$Z_t \leq \lambda \epsilon - \inf_{\tilde{\pi} \in \Pi^{soft}(T)} \sum_{h=1}^{H} \ln p_0(\mathcal{F}_h(\epsilon; \tilde{\pi})) + 4 \gamma \left( \alpha + \frac{3(e-2)}{\alpha} \right) H K \epsilon^2$$

$$+ 60 \alpha \gamma b(b+2) K H \epsilon + \alpha b^2 \gamma H \left( 13 + 36(e-2) \right) \left( 2 d_{\mathcal{F}}(\epsilon) + d_{\Pi}(\epsilon, T) + \ln(1/\delta) \right) + 18 \alpha \gamma K H b^2 \delta.$$

Thus, we have

$$\mathbb{E}_{\mathcal{D}} \mathbb{E}_{\tilde{\pi} \sim P_t(\cdot | \mathcal{D})} \mathbb{E}_{f \sim \hat{p}(\cdot | \mathcal{D}, \tilde{\pi})} \left[ 0.125 \alpha \gamma K \sum_{h=1}^{H} \mathbb{E}_{\mu} [\mathcal{E}_h^{\tilde{\pi}}(f_h, f_{h+1})(s_h, a_h)^2] + \lambda \Delta f_1(s_1, \tilde{\pi}) \right]$$

$$\lesssim \lambda \epsilon + \alpha \gamma H b^2 \cdot \max\{d_{\mathcal{F}}(\epsilon), d_{\Pi}(\epsilon, T), \ln \frac{\ln K b^2}{\delta}\} + \alpha \gamma b^2 K H \cdot \max\{\epsilon, \delta\} + \gamma H K \frac{\epsilon^2}{\alpha}$$

$$+ \sum_{h=1}^{H} \max_{\tilde{\pi}_h \in \Pi_h^{soft}} \kappa_h(\alpha, \epsilon, \tilde{\pi}_h) + \sup_{\tilde{\pi} \in \Pi^{soft}(T)} \sum_{h=1}^{H} \ln \frac{1}{p_0(\mathcal{F}_h(\epsilon; Q_h^{\tilde{\pi}_h}))}.$$

$\square$

## Appendix F  Proof of Proposition 1

In this appendix, we prove Proposition 1, which is a simple reduction from Theorem 1, Theorem 2, and Theorem 3.

*Proof of Proposition 1.* We recall that Proposition 1 consists of two parts of statements: Part (i) – the simplified bounds of all three algorithms into one unified form under no misspecification, and Part (ii) – the specialization of the unified bound into the special cases of finite function classes and linear function classes.

### Part (i): The unified sub-optimality bounds for VS, RO, and PS

We recall that the first part of Proposition 1 is that:

$$\forall \hat{\pi} \in \{\hat{\pi}^{vs}, \hat{\pi}^{ro}, \hat{\pi}^{ps}\}, \mathbb{E}_{\mathcal{D}} \text{SubOpt}_{\pi}(\hat{\pi}) = \tilde{\mathcal{O}} \left( \frac{H b}{\sqrt{K}} \sqrt{\tilde{d}(1/K) \cdot \mathcal{C}(\pi; 1/\sqrt{K})} + \frac{H b \sqrt{\ln \text{Vol}(\mathcal{A})}}{T} \right),$$
(19)

where

$$\tilde{d}(1/K) = \begin{cases} \tilde{d}_{opt}(1/K, T) & \text{if } \hat{\pi} \in \{\hat{\pi}^{vs}, \hat{\pi}^{ro}\}, \\ \tilde{d}_{ps}(1/K, T) & \text{if } \hat{\pi} = \hat{\pi}^{ps}, \end{cases}$$

where we recall in Section 4.2 that

$$\tilde{d}_{opt}(\epsilon, T) := \max\{d_{\mathcal{F}}(\epsilon), d_{\Pi}(\epsilon, T)\},$$

$$\tilde{d}_{ps}(\epsilon, T) := \max\{d_{\mathcal{F}}(\epsilon), d_{\Pi}(\epsilon, T), \frac{d_0(\epsilon)}{\gamma H b^2}, \frac{d'_0(\epsilon)}{\gamma H b^2}\},$$

and $d_{\mathcal{F}}(\epsilon)$, $d_{\Pi}(\epsilon, T)$, $d_0(\epsilon)$, and $d'_0(\epsilon)$ are defined in Section 2.4. Also recall that for Proposition 1, we assume that there is no misspecification, i.e., $\xi_h = \nu_h = 0, \forall h \in [H]$.

**For $\hat{\pi}^{vs}$.** It follows from Theorem 1, where we choose $\epsilon_c = 1/\sqrt{K}$, and $\epsilon = 1/K$ that with probability at least $1 - 2\delta$, we have

$$\text{SubOpt}_{\pi}(\hat{\pi}^{vs}) \lesssim \sqrt{K^{-1} \cdot H \cdot \mathcal{C}(\pi; 1/\sqrt{K})(Hb^2 \max\{\tilde{d}_{opt}(1/K, T), \ln(H/\delta)\} + b^2 H)} + H/\sqrt{K} + \zeta_{opt}$$

$$\lesssim \sqrt{K^{-1} \cdot H^2 b^2 \cdot \mathcal{C}(\pi; 1/\sqrt{K}) \max\{\tilde{d}_{opt}(1/K, T), \ln(H/\delta)\}} + \frac{Hb\sqrt{\ln \text{Vol}(\mathcal{A})}}{T}$$

Thus we have

$$\text{SubOpt}_{\pi}(\hat{\pi}^{vs}) = \mathcal{O}\left( \frac{Hb}{\sqrt{K}} \sqrt{\mathcal{C}(\pi; 1/\sqrt{K}) \cdot \max\{\tilde{d}_{opt}(1/K, T), \ln(H/\delta)\}} + \frac{Hb\sqrt{\ln \text{Vol}(\mathcal{A})}}{T} \right). \tag{20}$$

**For $\hat{\pi}^{ro}$.** The sub-optimality bound for $\hat{\pi}^{ro}$ is obtained from Theorem 2 with the same parameter setting as that for $\hat{\pi}^{vs}$, where we set $\epsilon = 1/K$, $\epsilon_c = 1/\sqrt{K}$, and $T \geq K \ln \text{Vol}(\mathcal{A})$. Additionally, we shall need to set the regularization parameter $\lambda$. Since the bound in Theorem 2 holds for any $\lambda > 0$, we shall minimize this bound with respect to $\lambda > 0$, which results in the optimal $\lambda$ as

$$\lambda_* = \sqrt{\frac{2KHb^2 \cdot \max\{\tilde{d}_{opt}(1/K, T), \ln(H/\delta)\}}{H \cdot \mathcal{C}(\pi, 1/\sqrt{K})}}.$$

and the sub-optimality bound as

$$\text{SubOpt}_{\pi}(\hat{\pi}^{ro}) = \mathcal{O}\left( \frac{Hb}{\sqrt{K}} \sqrt{\mathcal{C}(\pi; 1/\sqrt{K}) \cdot \max\{\tilde{d}_{opt}(1/K, T), \ln(H/\delta)\}} + \frac{Hb\sqrt{\ln \text{Vol}(\mathcal{A})}}{T} \right). \tag{21}$$

**For $\hat{\pi}^{ps}$.** We specialize the sub-optimality of $\hat{\pi}^{ps}$ from Theorem 3. Similar to the case of $\hat{\pi}^{vs}$ and $\hat{\pi}^{ro}$, we set: $\epsilon = 1/K$, $\epsilon_c = 1/\sqrt{K}$, and $T \geq K \ln \text{Vol}(\mathcal{A})$. Additionally, we need to set the failure probability $\delta \in [0, 1]$, the learning rate $\gamma \in [0, \frac{1}{144(e-2)b^2}]$ and the regularization parameter $\lambda > 0$. For $\delta$, we set $\delta = 1/K$. For $\lambda$, we minimize the bound in Theorem 3 with respect to $\lambda$, which results into $\lambda = \lambda_*$ which is give as

$$\lambda_* = \gamma \sqrt{\frac{KHb^2 \cdot \max\{\tilde{d}_{ps}(1/K, T), \ln(K \ln(Kb^2))\}}{H \cdot \mathcal{C}(\pi, 1/\sqrt{K})}},$$

turns the sub-optimality bound into

$$\mathbb{E}_{\mathcal{D}} \text{SubOpt}_{\pi}(\hat{\pi}^{ps}) = \mathcal{O}\left( \frac{Hb}{\sqrt{K}} \sqrt{\mathcal{C}(\pi; 1/\sqrt{K}) \cdot \max\{\tilde{d}_{ps}(1/K, T), \ln(K \ln(Kb^2))\}} + \frac{Hb\sqrt{\ln \text{Vol}(\mathcal{A})}}{T} \right). \tag{22}$$

Finally, we choose $\gamma \in [0, \frac{1}{144(e-2)b^2}]$ to minimize $\tilde{d}_{ps}(\epsilon, T) = \max\{d_{\mathcal{F}}(\epsilon), d_{\Pi}(\epsilon, T), \frac{d_0(\epsilon)}{\gamma H b^2}, \frac{d'_0(\epsilon)}{\gamma H b^2}\}$, which occurs at $\gamma = \frac{1}{144(e-2)b^2}$, and thus

$$\tilde{d}_{ps}(\epsilon, T) = \max\left\{ d_{\mathcal{F}}(\epsilon), d_{\Pi}(\epsilon, T), \frac{144(e-2)d_0(\epsilon)}{H}, \frac{144(e-2)d'_0(\epsilon)}{H} \right\}. \tag{23}$$

Overall, we have that Equation (20), Equation (21), and Equation (22) can be unified into Equation (19).

**Part (ii): Specializing to the finite function classes and linear function classes**

We consider two common cases.

**Case 1. Finite function class.** We consider the case that $\mathcal{F}_h$ and $\Pi_h^{soft}(T)$ have finite elements for all $h \in [H]$. Then we have $\tilde{d}(\epsilon) = \mathcal{O}(\max_{h \in [H]} \max\{\ln |\mathcal{F}_h|, \ln |\Pi_h^{soft}(T)|\}), \forall \epsilon$, due to that $d_0'(\epsilon) \leq d_0(\epsilon) \leq H \max_{h \in [H]} \ln |\mathcal{F}_h|$ and Equation (23).

**Case 2. Linear function class.** We consider the case that the function class $\mathcal{F}_h$ is linear in some (known) feature map $\phi_h : \mathcal{S} \times \mathcal{A} \to \mathbb{R}^d$. Concretely, the corresponding function class and the policy class defined in Section 2.3 are simplified into:

$$\mathcal{F}_h = \{(s,a) \mapsto \langle \phi_h(s,a), w \rangle : \|w\|_2 \leq b\},$$

$$\Pi_h^{soft}(T) := \left\{ (s,a) \mapsto \frac{\exp(\langle \phi_h(s,a), \theta \rangle)}{\sum_{a' \in \mathcal{A}} \exp(\langle \phi_h(s,a'), \theta \rangle)} : \|\theta\|_2 \leq \eta T \right\}.$$

We have

$$d_{\mathcal{F}}(\epsilon) \leq d \ln(1 + \frac{2b}{\epsilon}),$$

$$d_{\Pi}(\epsilon, T) \leq d \ln(1 + \frac{16\eta T}{\epsilon}),$$

$$d_0'(\epsilon) \leq d_0(\epsilon) \leq c_1 dH \ln(c_2/\epsilon),$$

where the first two inequalities use [Zanette et al., 2021, Lemma 6] and the last inequality follows the discussion in Section 2.4. Note that $d_{\Pi}(\epsilon, T)$ depends only logarithmically in $T$.

$\square$

# Appendix G   Support Lemmas

In this section, for convenience, we present some simple yet useful lemmas that our proofs above often refer to.

The following lemma establishes the variance condition for the squared loss, which is typically used along with Bernstein's inequality.

**Lemma G.1.** *Consider any real-valued function class $\mathcal{F}$. Consider the squared loss $L(f(x), y) = (f(x) - y)^2$. Assume bounded loss $L(f(x), y) \leq M^2$ for any $f \in \mathcal{F}$, for some $M > 0$. Let $f_*(x) = \mathbb{E}[y|x]$ and assume that $L(f_*(x), y) \leq B^2$ for some $B > 0$ (we do not require that $f_* \in \mathcal{F}$). Let $z = (x, y)$ and define*

$$\mathcal{G} = \{\phi(\cdot) : \phi(z) = L(f(x), y) - L(f_*(x), y), f \in \mathcal{F}\}.$$

*Then, for all $\phi \in \mathcal{G}$, we have*

$$\mathbb{E}_y[\phi(z)^2] \leq 2(M^2 + B^2)\mathbb{E}_y[\phi(z)], \forall x.$$

*where $\mathbb{E}_y$ is the expectation taken over $y$ given $x$.*

*Proof of Lemma G.1.* Consider any $\phi \in \mathcal{G}$ (with the corresponding $f \in \mathcal{F}$). For any $x$, we have

$$\mathbb{E}_y[\phi(z)] = (f(x) - f_*(x))^2.$$

Thus, we have

$$\begin{aligned}
\phi(z)^2 &= (f(x) - f_*(x))^2(f(x) + f_*(x) - 2y)^2 \\
&\leq (f(x) - f_*(x))^2 2[(f(x) - y)^2 + (f_*(x) - y)^2] \\
&\leq 2(M^2 + B^2)(f(x) - f_*(x))^2.
\end{aligned}$$

The first inequality uses Cauchy-Schwartz. The second inequality uses that $f_* \in \mathcal{F}$ and $L(f(x), y) \leq M^2, \forall f \in \mathcal{F}$. Thus, for any $x$, we have

$$\mathbb{E}_y[\phi(z)^2] \leq 2(M^2 + B^2)(f(x) - f_*(x))^2 = 2(M^2 + B^2)\mathbb{E}_y[\phi(z)].$$

The equation uses that $\mathbb{E}_y[\phi(z)] = (f(x) - f_*(x))^2$. $\square$

The following lemma is a simple decomposition of the value gap in the initial state, typically known as the performance difference lemma in the RL literature.

**Lemma G.2** (Performance difference lemma). *For any policy $\pi, \widetilde{\pi}$, we have*

$$V_1^{\pi}(s_1) - V_1^{\widetilde{\pi}}(s_1) = \sum_{h=1}^{H} \mathbb{E}_{\pi} \left[ Q_h^{\widetilde{\pi}}(s_h, a_h) - V_h^{\widetilde{\pi}}(s_h) \right],$$

*where $\mathbb{E}_{\pi}$ denotes the expectation over the random trajectory $(s_1, a_1, \ldots, s_h, a_h)$ generated by $\pi$ (and the underlying MDP).*

*Proof of Lemma G.2.* We simply expand $V_1^{\pi}(s_1) = \mathbb{E}_{a_1, s_2 | s_1, \pi}[r_1(s_1, a_1) + V_2^{\pi}(s_2)]$ and use recursion to obtain the lemma. $\qquad\qquad\square$

The following lemma presents a simple connection from a form of a log partition function to the expectation under the infimum realization of the sampling distribution.

**Lemma G.3.** *For any density functions $p$ and $p_0$ and any function $f$, we have*

$$\inf_{p} \mathbb{E}_{x \sim p(x)} \left[ f(x) + \ln \frac{p(x)}{p_0(x)} \right] \geq - \ln \mathbb{E}_{x \sim p_0} \exp(-f(x)).$$

*Proof of Lemma G.3.* Define the density function

$$q(x) = \frac{p_0(x) \exp(-f(x))}{Z(f)} \text{ where } Z(f) := \mathbb{E}_{x \sim p_0(x)} \exp(-f(x)).$$

Then, we have

$$\begin{aligned} \mathbb{E}_{x \sim p(x)} \left[ f(x) + \ln \frac{p(x)}{p_0(x)} \right] &= \mathbb{E}_{x \sim p(x)} \ln \frac{p(x)}{q(x)} - \ln Z(f) \\ &= KL[p \| q] - \ln Z(f) \\ &\geq - \ln Z(f). \end{aligned}$$

$\square$

