# OpenReview forum: "On Sample-Efficient Offline Reinforcement Learning: Data Diversity, Posterior Sampling and Beyond"
_NeurIPS.cc/2023/Conference — NeurIPS 2023 poster_

### Official Review · Reviewer_FhSe · 2023-07-04

**Soundness:** 4 excellent
**Presentation:** 4 excellent
**Contribution:** 4 excellent
**Rating:** 7
**Confidence:** 2

**Summary:**

This paper proposes a new notion of data diversity that expands all the prior data diversity conditions in offline reinforcement learning literature. Based on this notion, the paper develops concrete value sampling-based, regularized optimization-based and posterior sampling-based algorithms with corresponding sample complexity results. The paper shows that these three algorithms can achieve comparable sample efficiency.

**Strengths:**

1. The proposed data diversity condition and its corresponding suboptimality guarantee seem to be general and tighter than previous diversity conditions characterizing the state-of-the-art bounds for the main streams of offline reinforcement learning frameworks. This data diversity condition is a potential inspiration for future work.

2. The writing is clear and presents a fair and clean picture compared to previous results.

**Weaknesses:**

There are some typos and inconsistencies in the notation, such as $p_{0,h}$ in $p_{0,h}(\mathcal{F}_h^{\tilde{\pi}}(\epsilon; f\_{h+1}))$ in line 176 and $f_h$ and $g_h$ in Algorithm 2, which should be $f$ and $g$, respectively. Similar inconsistencies appear in Algorithm 3.

**Questions:**

Could you please explain how the new diversity notion is able to cover value sampling-based, regularized optimization-based, and posterior sampling-based algorithms, especially posterior sampling-based algorithms? Is it because the previous diversity conditions are not always able to cover all three types of algorithms? If the previous conditions are able to cover all three types, is the main advantage of the new notion its tightness? If the previous conditions cannot cover any of the algorithm types, could you describe the reason behind their failure and why the new notion is able to succeed?

**Limitations:**

See weakness.

---

> ### Author Rebuttal · Authors · 2023-08-09
>
> We thank you for your positive feedback. We elaborate further on the data diversity notion below.
>
> ---
>
> **Question**: **About (the advantage of) data diversity**
>
> **Our response**: Our notion of data diversity provides a tighter characterization of distribution shift than the previous notions of data coverage and facilitates a unified framework for the analysis of the three algorithms we consider. That said, our tight(er) results (especially) for the regularized optimization-based method and the (new) posterior sampling-based method come from our refined analysis, not solely from the data diversity notion alone. The idea of the data diversity notion, which we discovered naturally and turns out to be inspired by the transfer learning literature, is to account for the error when we decouple the Bellman error under a target policy into the squared Bellman error under the behavior policy. With the data diversity notion, we showed that the scenarios for the offline data in which offline RL is learnable are enlarged compared to the picture depicted by the prior data coverage notions. With our refined analysis, we showed that the regularized optimization-based algorithm and the new (pessimistic) posterior sampling algorithm are not only provably efficient but also have comparable guarantees just as the (intractable) version space-based algorithm (under standard assumptions).

---

> > ### Comment · Reviewer_FhSe · 2023-08-21
> >
> > Thanks for the clarification. I will keep my belief as before.

---

> > > ### Author Response · Authors · 2023-08-21
> > >
> > > Thank you for taking the time to read our rebuttal.

---

### Official Review · Reviewer_ufo5 · 2023-07-05

**Soundness:** 3 good
**Presentation:** 3 good
**Contribution:** 3 good
**Rating:** 5
**Confidence:** 3

**Summary:**

This paper investigates the problem of sample-efficient learning from historical datasets in the context of offline reinforcement learning and explores the role of data diversity and posterior sampling in improving the efficiency of RL algorithms. The authors propose a new notion of data diversity and study three classes of offline RL algorithms based on version spaces, regularized optimization, and posterior sampling. They find that these algorithms achieve comparable sample efficiency, contrary to prior work. The paper also introduces a novel model-free posterior sampling algorithm for offline RL.



**Strengths:**

This work studies offline RL problems within the context of general function approximation. The introduction of the novel notion of data diversity is intriguing, as it offers a fresh perspective on the problem. Additionally, the inclusion of the posterior sampling algorithm is interesting, as it appears to be a new contribution to the literature on offline RL theory.

**Weaknesses:**

It is important to ensure the accuracy of the comparison with LCB-based algorithms, as the algorithm mentioned in [JYW21] may not be considered the SOTA algorithm when compared to improved LCB-based algorithms proposed in works such as [1] and [2], which have demonstrated better dependencies on d.

Additionally, it is worth noting that there has been an update in [XCJ+21], where a new arXiv version employs $\Pi^{soft}$ instead of $\Pi^{all}$. While this minor issue does not impact my evaluation, it would be beneficial for the authors to acknowledge this update and adjust the comparisons accordingly.

[1] Near-optimal offline reinforcement learning with linear representation: Leveraging variance information with pessimism

[2] Nearly Minimax Optimal Offline Reinforcement Learning with Linear Function Approximation: Single-Agent MDP and Markov Game


**Questions:**

n/a

---

> ### Author Rebuttal · Authors · 2023-08-09
>
> We thank you for your positive feedback. We address all of your concerns below.
>
> ---
>
> #### **Concern 1**:  **Compare with [1][2]**
> **Relevant refs**:
> > [1] Ming Yin, Mengdi Wang, Yu-Xiang Wang. “Near-optimal offline reinforcement learning with linear representation: Leveraging variance information with pessimism”. ICLR 2022.
>
> > [2] Wei Xiong, Han Zhon, Chengshuai Shi, Liwei Wang. “Nearly Minimax Optimal Offline Reinforcement Learning with Linear Function Approximation: Single-Agent MDP and Markov Game”. ICLR 2023.
>
> **Our response**: We will update the comparison of the instantiation of our results to the linear function class to the results of [1][2] in our revision. In this case, the bounds in [1][2] have the same dependence on $d$ as our bound and have a tighter dependence on $H$. We remark that this improved dependence on $H$ of [1][2] is due to the variance-weighted value iteration algorithm that they employ to capture different (heteroscedastic) variance of the transition kernels $P_h$ at different time steps $h$. This improved dependence on $H$ comes at the cost that the algorithm is more complicated and they need the offline data to be explorative over all dimensions of the feature map so that the estimation error of the variance can be controlled.
>
> We remark that leveraging the variance information (using the variance-weighted value iteration) as in [1][2] is **complementary/orthogonal** to the basic algorithms we study in our paper, i.e., this idea can be added to our algorithms to improve the dependence on $H$, as we discussed as future work in Section 5.
>
> ---
>
> **Concern 2**: **Compare with updated version of [XCJ+21]**
>
> **Our response**: For the version space-based result, we compared with Theorem 3.1 of (the latest version of) [XCJ+21] where their bound scales with the complexity of the comparator policy class (not the soft policy class). We believe that the updated version was to correct their Theorem 3.2. For comparison in the linear case, we included the bound in their Theorem 3.2 of their updated version in Table 2 of our supplementary. For the regularized optimization-based algorithm, [XCJ+21] employed $\Pi^{soft}$ instead of $\Pi^{all}$, as we acknowledged in our Table 1.  Regardless, we fully acknowledged the invention of the version space-based algorithm and the regularized optimization-based algorithm (for offline RL) to [XCJ+21] (e.g. see Line 189-193).

---

> > ### Comment · Reviewer_ufo5 · 2023-08-14
> >
> > Thank you for your response. However, I would like to point out that without the explorative assumption and the variance-weighted algorithm design, LCB-based algorithms can enhance their dependency on $d$ by employing the reference-advantage decomposition [2] or data splitting scheme [JYW21]. So I would like to maintain my belief that asserting an improvement over LCB-based algorithms remains improper.

---

> > > ### Author Response · Authors · 2023-08-15
> > >
> > > We thank the reviewer for the further comments on the comparison with LCB-based algorithms. We agree that for the enhanced dependence on $d$ alone, the LCB-based algorithm in [2] does not need the variance-weighted algorithm design. We, however, remark that, as far as we can tell, this enhanced dependence on $d$ in [2] requires the explorative assumption (their Assumption 1) (e.g., please see the “High-order Error from Correlated Advantage Function” paragraph in Section 5 of [2] and the detailed proof of Theorem 1 in Section D of [2]). To the best of our knowledge, we are unaware of any reference-advantage decomposition argument that has an enhanced dependence on $d$ without some explorative assumption such as Assumption 1 of [2]. If we’ve missed any reference otherwise, we would be happy to take any suggestion to make our comparison more accurate.
> > >
> > > We also agree that the data splitting scheme in [JYW21] also has an enhanced dependence on $d$. However, the data splitting scheme in [JYW21] only uses $K/H$ episodes (instead of using all $K$ episodes) for estimating the value functions. As a result, the improved $d$ dependence of the data splitting in [JYW21] comes at the cost of much looser dependence on $H$. As a concrete example, in the finite spectrum condition and under the well-explored dataset, the sub-optimality presented in Proposition 4.11 of [JYW21] incurs an additional factor of $\sqrt{H}$ (as also discussed by [JYW21] in the paragraph after Proposition 4.11).
> > >
> > > We appreciate your suggestion and will continue to update our revision to make accurate comparisons with LCB-based algorithms ([1][2]) based on the discussion here.

---

### Official Review · Reviewer_YpYn · 2023-07-06

**Soundness:** 3 good
**Presentation:** 4 excellent
**Contribution:** 3 good
**Rating:** 6
**Confidence:** 3

**Summary:**

This work aims to point out what enables the sample efficiency of offline reinforcement learning.
The authors first define a new notion of data diversity based on the inducing the Bellman error under one policy with the Bellman error under a different policy.
The authors then propose a unified view, Generic Offline Policy Optimisation framework, to study the three offline algorithms: version spaces, regularised optimisation, and posterior sampling.
Under standard assumption, the authors showed that all these three algorithms can achieve comparable upper bounds of the policies' sub-optimality.

**Strengths:**

1. The proposed view is novel, and can unify the three different types of offline reinforcement learning algorithms.

2. The whole work is very well presented, and the structure is super clear, which raises a high comprehensibility.

3. All definitions and assumptions are explicitly and clearly given in Section 2. I personally love this practice, as it helps a lot on figuring the scope of this work.

4. The results are also well-organised and clearly listed in Section 4.

5. The connection between this work and [TJJ20] is clear, thus it's interesting to see that offline RL can be connected to transfer learning, though there are certain technical differences.

**Weaknesses:**

### Major

1. **Practical issues of posterior sampling algorithms**:
My biggest concern about this work is the performance of the posterior sampling based algorithms in practice.
I appreciate that the authors remark some feasible solutions to implement an offline RL algorithm based on posterior sampling from line 241 to line 245.
However, given the tractable approximations from line 243 to line 245, I kind of worry the *performance* as well as the *time complexity* of the PS implementations.
I'd be happy to raise my score if the authors can discuss further details about this concern in the updated version.

2. **Discussion about the limitations**:
The authors mainly focus on the missing variance information in their discussion about the limitation of the work.
Given the assumptions listed in the Section 2.3 as well as the practical issues I mentioned above, the limitations can be expanded to cover all of them.

### Minor

1. $H$ in line 107 and line 110:
The authors can move the notation $H$ from line 110 where it's defined to line 107 where it first appears.

2. $t$ in line 129 is not defined:
Though the meaning of $t$ can be inferred given the context, it's better to be defined before its appearance in line 129.

3. Introduction section is too lengthy
I appreciate the gentle and comprehensive introduction to this work in Section 1.
But, it might be a bit lengthy, and as a result Section 5 is too short to further discussion about the limitation and future directions of this work.

**Questions:**

1. Can the authors comment on the *performance* as well as the *time complexity* of possible PS implementations?
(See the major issue #1 in the weakness section)

2. What are the slight refinements in line 190?
The authors claimed that they have made slight refinements to the RO-based and VS-based algorithms, but they haven't mentioned these refinements anywhere else.

**Limitations:**

See the major issue #2 in the weakness section.

---

> ### Author Rebuttal · Authors · 2023-08-09
>
> We thank you for your positive feedback. We address all of your concerns and questions below.
>
> ---
> #### **Question 1**: **Practical issues of posterior sampling**
>
> **Our response**: We emphasize that **the focus of our paper is on the statistical aspects of offline RL** and **not** on the computational aspects (nor on any practical approximation or implementations). Our remarks on lines 243-245 serve merely as remarks regarding implementation/approximation for the interested readers/practitioners, rather than a formal treatment. We do not claim any practical insights into posterior sampling as a contribution.
>
> Regarding the performance and computational considerations regarding posterior sampling in practice, we would like to make the following comments. First, our posterior sampling is oracle-efficient, if we assume access to the expectation-computing oracle and the sampling oracle. As we noted in our paper (Lines 241-245), the expectation-computing oracle can potentially be replaced by another sampling oracle, inspired by the recent idea of [AZ22]. More specifically, $f’$ in the denominator of the likelihood in line 1 of Algorithm 4 can be replaced by a random sample from an inner-loop posterior distribution.  In turn, the sampling oracle in practice can be approximated by first-order sampling methods such as Langevin Monte Carlo. A rigorous treatment of this idea in our case is another avenue for future work.
>
> Second, the performance (both statistically, computationally, and empirically) of such first-order sampling approximation to posterior sampling (even for online RL and especially for offline RL) is another active research area. As a concrete example of the success of first-order sampling approximation to posterior sampling, a recent work of [2] shows that in (online) high-dimensional linear contextual bandits, one can use Langevin Monte Carlo (LMC) to approximate the “feel-good” Thompson sampling [3] (a posterior sampling algorithm that lays a foundation for our pessimistic posterior sampling algorithm) where  LMC obtains an optimal sample complexity of $\mathcal{O}(d^2/\epsilon^2)$ with the computational complexity of $\mathcal{O}(d^{9}/ \epsilon^8)$ ([2] also showed the superior performance of LMC compared to standard posterior sampling in their experiments).
>
> Third, as a note about using sampling in place of optimization,  in certain non-convex settings, sampling even converges provably much faster than optimization (e.g., $\mathcal{O}(d/\epsilon)$ or $\mathcal{O}(d^2 \log(1/\epsilon))$ vs $\Omega((1/\epsilon)^d)$ in the non-convex setting considered in [1]). Thus, it might be possible that the computational complexity of the LMC algorithm (to approximate our pessimistic posterior sampling) is smaller than that of the regularized optimization-based algorithm in certain scenarios. However, this speculation needs a more formal characterization which is also an avenue for future work.
>
> We will add this discussion to our revised version.
>
> **Relevant refs**:
>
> [1] Yi-An Ma, Yuansi Chen, Chi Jin, Nicolas Flammarion, and Michael I. Jordan. “Sampling can be faster than optimization”. PNAS 2019.
>
> [2] Tom Huix, Matthew Zhang, and Alain Durmus. “Tight Regret and Complexity Bounds for Thompson Sampling via Langevin Monte Carlo”. AISTATS 2023.
>
> [3] Tong Zhang. “Feel-Good Thompson Sampling for Contextual Bandits and Reinforcement Learning”. SIAM 2022. (cited as [Zha22] in our submission)
>
> ---
> #### **Question 2**: **Clarification of ”Slight refinements”**
>
> **Our response**:  We consider episodic time-inhomogeneous MDP instead of discounted MDP, thus the algorithms are refined accordingly. For the version space algorithm, compared to [XCJ+21], we employed the actor-critic framework instead of directly solving the min-max optimization over the function class and the policy class.

---

> > ### Comment · Reviewer_YpYn · 2023-08-19
> >
> > Thanks for the responses from the authors. Since my major concern #1 as been well alleviated, I decided to raise my score.

---

> > > ### Author Response · Authors · 2023-08-19
> > >
> > > Thank you for taking your time to read our rebuttal and for giving your feedback. We will integrate your suggestions (including the “minor” ones) in our next revision.

---

### Official Review · Reviewer_niYV · 2023-07-28

**Soundness:** 3 good
**Presentation:** 3 good
**Contribution:** 3 good
**Rating:** 4
**Confidence:** 4

**Summary:**

This paper introduces a novel data coverage measure called "data diversity," which is more stringent than existing data coverage measures like single-policy concentrability. The authors claim that actor-critic algorithms based on VS, RO, and posterior sampling can achieve state-of-the-art sample complexity. The posterior sampling based algorithm is the first of its kind for offline RL.


**Strengths:**

The new data coverage measure is innovative and tighter than existing measures in the literature. The authors' demonstration of good sample complexity under this measure is an important contribution. Additionally, the framework and the posterior sampling approach for offline RL are novel.


**Weaknesses:**

However, I have some doubts about the claim that these algorithms achieve the state-of-the-art convergence rate. The policy $\hat{\pi}$ is obtained by uniformly drawing a $t$ from $[T]$ and letting $\hat{\pi}$ be $\pi^t$. This policy is not a Markov policy. According to the definition of $V^{\pi}$, $V^{\hat{\pi}}=\frac{1}{T}\sum_{i=1}^TV^{{\pi}^i}$. Is this correct? The $1/T$ term comes from the additional expectation over the uniform distribution. For the chosen $t$, $\pi^t$ can have constant variance, and as a result, the high probability bound may not hold for this chosen $\pi^t$.

On the other hand, recent literature such as "Optimal conservative offline RL with general function approximation via augmented Lagrangian", "Offline Primal-Dual Reinforcement Learning for Linear MDP", and "Revisiting the linear-programming framework for offline RL with general function approximation" have policies that are Markov policies and achieve the state-of-the-art sample complexity. Therefore, the authors should compare their results with these papers.

To ensure a fair comparison, the output policy should be a Markov policy. If not, the authors need to clarify this in the introduction and after the theorems. Additionally, the authors should clearly explain the source of randomness for $\hat{\pi}$ in the theorems.

If the authors can address these issues clearly, I would increase the rating.

**Questions:**

None

---

> ### Author Rebuttal · Authors · 2023-08-09
>
> We thank the reviewer for the positive feedback. We address all of your concerns below.
>
> ---
> ####  **Concern 1**: **"This policy ($\hat{\pi}$) is not a Markov policy ... "**
>
> **Our response**:
> We understand our notation in line 7 of Algorithm 1 might have caused some confusion. To clarify, $\hat{\pi}$ is simply chosen uniformly from the policy set $ \{ \pi^1, …, \pi^T \}$. We emphasize that $\hat{\pi}$ (as well as each $\pi^t$) **is Markovian** (i.e., at each step $h$, each $\pi^t_h$ depends solely on the current state). The uniform mixture of policies (resulting from the multiplicative weights algorithm) and its Markovian property are quite standard in the literature, e.g., see [ZWB21], [XCJ+21], [CXJA22].
>
> ---
>
> ####  **Concern 2**: **Comparison with recent literature [1][2][3]**
>
> > **Relevant refs**:
>
> > [1] Paria Rashidinejad, Kunhe Yang, Stuart Russell, Jiantao Jiao. "Optimal conservative offline RL with general function approximation via augmented Lagrangian". ArXiv 2022 (cited as [RZY+22] in our submission).
>
> > [2] Germano Gabbianelli, Gergely Neu, Matteo Papini. "Offline Primal-Dual Reinforcement Learning for Linear MDP". ArXiv 2023.
>
> > [3] Asuman Ozdaglar, Jiawei Zhang, Kaiqing Zhang. “Revisiting the linear-programming framework for offline RL with general function approximation". ArXiv 2022.
>
> **Our response**:
> We believe that the primal-dual methods, e.g. [1][2][3] and [ZHH+22, CJ22] provide an important alternative to addressing offline RL. However, as we also acknowledged in footnote 5 of our submission, the guarantees of primal-dual methods use **a different set of assumptions** than the value-based methods we considered (the former assumes realizability for the ratio between the state-action occupancy density of the target policy and the state-action occupancy density of the behavior policy, except for [2] where this assumption is implicitly realized under a stronger assumption of linear MDP). This makes the results presented in our paper and the results in [1][3] **not directly comparable**, though our results and [1][3] both achieve the optimal sample complexity of $\mathcal{O}(\epsilon^{-2})$.
>
> Since [2] (which appeared online on May 22 after the main submission deadline) works in linear MDP, [2] is more comparable to the instantiation of our results to the linear function class. [2] consider primal-dual methods for offline RL in both infinite-horizon discounted MDP and average-reward MDP. Our analysis framework for the regularized optimization method in the episodic MDP should translate to the infinite-horizon discounted MDP as well, where the regularized optimization achieves the optimal sample complexity of $\mathcal{O}(\epsilon^{-2})$ while the sample complexity in [2] in this setting is $\mathcal{O}(\epsilon^{-4})$.  However, [2] offers a better computational complexity ($\mathcal{O}(K)$ vs $\mathcal{O}(K^{7/5})$ ) (where $K$ is the number of offline episodes) and also works in the average-reward MDP setting which is beyond the episodic MDP setting considered in our work; though our bounds hold for general function approximation that is beyond the strong assumption of linear MDPs in [2].  We will add this discussion and comparison in our revision.

---

> > ### Comment · Reviewer_niYV · 2023-08-16
> >
> > Thank the authors for answering my questions. However, I do not think my concerns have been fully adreesed.
> >
> > The authors give high probability bound but have not clearly tell the readers what the randomness comes from. Does the randomness contains the randomness for the uniform distribution for selecting a policy from $\{\pi^1,\cdots,\pi^T\}$?
> > I am also not convinced that the mixed policy is a Markov policy. The index is selected in the begining by uniform sampling  and does not change. At any horizon, the action depends not only on the current state but also depends on the random index determined in the begining. So I do not think it is a Markov policy.

---

> > > ### Author Response · Authors · 2023-08-17
> > >
> > > We thank the reviewer for further clarification of the questions when our initial response was not clear enough.
> > >
> > > **Randomness in the high-probability bound**. The randomness in the high probability bounds is solely from the randomness of the offline dataset, not from the policy mixture. To describe it in a context, the value sub-optimality (defined in Eq. (1)) of $\hat{\pi}$ in our theorems can be written explicitly as follows:
> > >
> > >  $SubOpt_{\pi}(\hat{\pi}) =  V_1^{\pi}(s_1) -  V_1^{\hat{\pi}}(s_1)  = V_1^{\pi}(s_1) -  \frac{1}{T} \sum_{t=1}^T V_1^{\pi^t}(s_1)$.
> > >
> > > We thank the reviewer again for the good question. Even though the mixture policy is standard in the literature [ZWB21,XCJ+21,CXJA22], we will make sure to clarify the source of the randomness of our high-probability bound statements in our revised version.
> > >
> > > **Markovian of the mixture policy**. We remark that even though $\hat{\pi}$ is a mixture of several policies, i.e., $\hat{\pi}\_h(a_h|s_h) = \frac{1}{T} \sum_{t=1}^T \pi^t_h(a_h|s_h)$ for all $h \in [H]$, $\hat{\pi}_h$ is Markovian by definition since each $\pi^t_h$ is Markovian. We explain further in the following.
> > >
> > > By definition, a Markovian policy is a mapping solely from the current state to a distribution over the action space, completely independent of any previous states and actions.
> > >
> > > First, each $\pi^t_h$ is Markovian because it is fully characterized by another Markovian policy $\pi^{t-1}_h$ (by induction from the uniform policy $\pi^1_h$ which is Markovian), and the state-action value function estimate (that depends solely on the current state and action, not on any previous states and actions). For the full formulae of $\pi^t_h$, please see Line 5 of Algorithm 1 in our submission.
> > >
> > > Second, note that
> > > $\hat{\pi}\_h(a_h|s_h) =   \frac{1}{T} \sum_{t=1}^T \pi^t_h(a_h|s_h)$               (please notice the same time index $h$ on both sides of the equation), the actions from $\hat{\pi}\_h$ solely depends on the current state $s_h$, not on any previous states and actions $s_{h’}, a_{h’}$ for $h’ < h$. Thus, by definition, each $\hat{\pi}_h$ is also Markovian. In other words, mixture — a simple weighting across several Markovian policies at the same time index $h$, does not break the Markovian property.
> > >
> > > As a concrete example, let’s consider the simple case that $T = 2$ and $\mathcal{A} = $ `{`$a_1, a_2$`}`, $\pi^1_h(a_1|s) = \pi^1_h(a_2|s) = 0.5$, and $\pi^2_h(a_1|s) = 0.2$, $\pi^2_h(a_2|s) = 0.8$. Then, by the definition of the mixture policy, we simply have $\hat{\pi}_h(a_1|s) = 0.5(\pi^1_h(a_1|s) + \pi^2_h(a_1|s)) = 0.5 * (0.5 + 0.2) = 0.35$ and $\hat{\pi}_h(a_2|s) = 0.5(\pi^1_h(a_2|s) + \pi^2_h(a_2|s)) = 0.5 * (0.5 + 0.8) = 0.65$. The mixture policy $\hat{\pi}_h$ is obviously Markovian (the probability action at each time step $h$ depends solely on the current state at time $h$, not on any previous states and actions).
> > >
> > > We’re happy to answer new questions if our clarification is still not clear enough.

---

> > > > ### Comment · Reviewer_niYV · 2023-08-18
> > > >
> > > > I'm grateful to the authors for replying to my questions. Nonetheless, I believe the authors  have  not addressed the concerns about the Markovian property and the high probability bound. Consequently, I'm modifying my score to 4.
> > > >
> > > > I understand that $\pi^t_h$ by itself adheres to the Markov policy. However, the mixed policy $\hat\pi$ isn't Markovian, as actions depend not only on the previous  state but also on the random seed  for selecting $t,h$  initially.
> > > > It's important for the authors to recognize that $\hat\pi$ isn't equivalent to $\pi^t_h$.
> > > >
> > > > The policy is Markovian only if, at any time horizon, the agent uniformly samples $t,h$ and employs $\pi^t_h$.
> > > > Regarding the high probability bound, the randomness doesn't encompass the randomness of the uniform distribution. Hence, the high probability bound doesn't imply the near optimality of the Markov policy $\pi^t_h$ and solely demonstrates accuracy for the mixed policy, which isn't Markovian.
> > > >
> > > > I'm open to increasing my score if the authors can demonstrate that the near optimality holds for $\pi^t_h$ considering both the randomness of the data distribution and the uniform distribution.
> > > > Otherwise, I'm not convinced that a theoretical guarantee for the mixed policy holds enough significance.
> > > >
> > > >
> > > > I can also give an example showing that the mixed policy may not be Markovian:
> > > > There are two states $(s^1,s^2)$ and two actions $(a^1,a^2)$. Consider $3$-horizon case with time index t=0,1,2.
> > > >
> > > > Suppose we have $P(s^1|s,a^1)=P(s^2|s,a^1)=1/2$ and $P(s^2|s,a^2)=1$ for any $s\in\{s^1,s^2\}$. Also, suppose there are two policies that form a mixture policy uniformly (with probability 1/2 and 1/2), where each policy $\pi^t$ (where we are using the notation in the paper) always takes action $a^t$ for $t\in \{1,2\}$. Suppose the initial state $s_0=s^1$. We use the subscript to denote the timestep. Then, we have $P(s_2=s^2)=6/8$ and $P(a_2=a^2)=1/2$, and further, by Bayes rule, we have $P(s_2=s^2|a_2=a^2)=1$ because the action $a^2$ only leads to $s^2$. Then, we have $P(a_2=a^2|s_2=s^2)=4/6$. However, we have $P(a_2=a^2|s_2=s^2,s_1=s^1)=0$ since $s_1=s^1$ means we take $a=a^1$ in the beginning. This means the policy is not Markovian.

---

> > > > > ### Author Response · Authors · 2023-08-19
> > > > >
> > > > > We thank the reviewer for the further comments. We are grateful for the reviewer's feedback which has helped improve the clarification of the presentation of our paper.
> > > > >
> > > > > We believe the root of the confusion might be Line 7 of Algorithm 1. We believe we would be better served if we wrote the output as $Uniform(\pi^1, \ldots, \pi^t)$ (there is no sampling over $h$). In other words, Algorithm 1 does not return a mixture policy, even though we may have referred to it as that in the rebuttal. Note that nowhere in the paper do we even use that word. What the Algorithm returns is one of the $T$ policy iterates uniformly sampled. Since each policy iterate is Markovian, what the algorithm returns is Markovian. We apologize for using that terminology in our earlier response. It might have caused unnecessary confusion.
> > > > >
> > > > > Regarding the bound itself, we give a high probability bound on the sub-optimality of the output of Algorithm 1. As we noted above, the output of Algorithm 1 is a Markovian policy. We should also add that the sub-optimality is defined w.r.t. any comparator policy (i.e., results hold for all $\pi \in \Pi^{all}$). We can always choose the comparator policy $\pi$ to be $\pi^*$, so we do bound the near optimality of the policy returned by the learner. The source of randomness as we said before is the offline data and any internal randomization of the algorithm — this is typical in guarantees of this form. We are not deviating from the norm here in specifying these bounds (e.g., see the bounds of similar algorithms in [ZWB21,XCJ+21,CXJA22] for the norm of presentation). For better presentation, we can take the expectation with respect to the randomness of the algorithm when writing out the bound. So, we will replace $SubOpt_{\pi} (\hat{\pi})$ in Theorems 1 and 2 by $ \mathbb{E} [SubOpt_{\pi}(\hat{\pi}) | \mathcal{D} ] = V_1^{\pi}(s_1) - \frac{1}{T} \sum_{t=1}^T V_1^{\pi^t}(s_1)$, and clarify that the expectation is taken w.r.t. uniform sampling of the iterates. The only source of randomness that would remain then is the randomness inherent in the offline dataset.
> > > > >
> > > > > Would that change be acceptable to the reviewer? We do think that will make things very clear.
> > > > >
> > > > > ---
> > > > >
> > > > > For an alternative presentation, we also propose a different approach to addressing the confusion. This approach is essentially equivalent to the first approach above but totally avoids using the term “uniform distribution” over $\pi^1, …, \pi^T$ as the output of Algorithm 1. We propose to revise Line 197 (and thus Line 7 of Algorithm 1 accordingly) to read: “After $T$ update steps, Algorithm 1 *deterministically* outputs a single policy $\hat{\pi} = (\hat{\pi}_1, \ldots, \hat{\pi}_H)$ where we define $\hat{\pi}_h(a|s) = \frac{1}{T} \sum\_{t=1}^T \pi^t_h(a|s)$ for any $h \in [H]$" (please note that $\hat{\pi}_h$ is computed *deterministically* from $\pi^1_h, …, \pi^T_h$ without defining “uniform distribution” as the output of Algorithm 1). With that, since each policy iterate $\pi^t$ is Markovian, what the algorithm returns $\hat{\pi}$ is also Markovian.
> > > > >
> > > > > Now, by linearity of the value function $V^{\pi}$ in terms of $\pi$, we have $V_1^{\hat{\pi}} (s_1)= \frac{1}{T} \sum_{t=1}^T V_1^{\pi^t} (s_1) $. This is a *deterministic* relation, without defining “uniform distribution” as the output of Algorithm 1. Consequently, we have the following *deterministic* relation:
> > > > >
> > > > > $ SubOpt\_{\pi}(\hat{\pi}) = V_1^{\pi}(s_1) - V_1^{\hat{\pi}}(s_1) = V_1^{\pi}(s_1) - \frac{1}{T} \sum_{t=1}^T V_1^{\pi^t}(s_1) = \frac{1}{T} \sum_{t=1}^T ( V_1^{\pi}(s_1)  - V_1^{\pi^t}(s_1))$.
> > > > >
> > > > > Here, there is no internal randomization in the algorithm (as we do not define “uniform distribution” as the output of Algorithm 1) and the only source of randomness in our algorithm and in our high-probability bounds is the offline data.
> > > > >
> > > > > Would that change also be acceptable to the reviewer? We think either of the approaches would help address the unnecessary confusion in our submission version.
> > > > >
> > > > > Again, we apologize for the unnecessary confusion.
> > > > >
> > > > > ---
> > > > >
> > > > > We would also add that in our proof of bounding the value sub-optimality of $\hat{\pi}$, we do bound the value sub-optimality $V_1^{\pi}(s_1)  - V_1^{\pi^t}(s_1)$ of each individual $\pi^t$. Specifically, using Lemma B.4 in our supplementary: $V_1^{\pi}(s_1)  - V_1^{\pi^t}(s_1)= \sum_{h=1}^H \mathbb{E} [\mathcal{E}\_h^{\pi^t}(Q_h, Q_{h+1})] + Q_1(s_1, \pi^t_1) - V_1^{\pi^t}(s_1) + SubOpt_{\pi}^{M(Q, \pi^t)}(\pi^t)$ for any $t \in [T]$. From Lemma B.6: $\sum_{t=1}^T SubOpt_{\pi}^{M(Q, \pi^t)}(\pi^t) ≤ Hb \sqrt{T \ln |\mathcal{A}|}$ (it holds deterministically). The high-probability bounds come from bounding $\sum_{h=1}^H \mathbb{E} [\mathcal{E}\_h^{\pi^t}(Q\_h, Q_{h+1})] + Q_1(s_1, \pi^t_1) - V_1^{\pi^t}(s_1)$ for any $t$. In fact, we bound $\sum_{h=1}^H \mathbb{E} [\mathcal{E}\_h^{\tilde{\pi}}(Q_h, Q_{h+1})] + Q_1(s_1, \tilde{\pi}_1) - V_1^{\tilde{\pi}}(s_1)$ uniformly over all $\tilde{\pi} \in \Pi^{soft}$ as $\pi^t \in \Pi^{soft}$.

---

> > > > > > ### Author Response · Authors · 2023-08-21
> > > > > >
> > > > > > As the current discussion phase is closing, for the sake of any other discussions after this phase, we thought we would add our comment to the reviewer’s example, in case it was not clear from our previous response. We believe the example was constructed out of the confusion and does not support any conclusion (let alone some miscalculations in the example). In short, it's merely by the very definition that $\hat{\pi} = (\hat{\pi}\_1, \ldots, \hat{\pi}_H)$ where $\hat{\pi}_h(a_h|s_h) = \frac{1}{T} \sum\_{t=1}^T \pi^t_h(a_h|s_h)$ which is Markovian by the very definition. In the example, given $\pi^t_h(a^t | s) = 1$ for all $t = 1,2$ and $h = 1,2,3$ in the example, it is merely that $\hat{\pi}_h(a_h = a^t | s_h = s) = 0.5$ for all $t = 1,2$ and $h = 1,2,3$.
> > > > > >
> > > > > > Finally, we believe that our proposed steps for improving the presentation of our paper in our earlier responses should clearly address this unnecessary confusion. We apologize again for the unnecessary confusion and we appreciate the reviewer for taking the time to read our rebuttal.

---

### Decision · Program_Chairs · 2023-09-21

**Decision:**

Accept (poster)

**Comment:**

The paper considers a new notion of diversity in offline RL, and shows the efficiency of regularization optimization and posterior sampling approaches. It contains interesting ideas, and in general, the reviews are positive. However, there was some concern regarding the comparison to another series of works that can "directly" output Markov policies with "high" probability, while the present paper outputs a "mixture" policy. Directly outputting a Markov policy from the current results might lead to weaker guarantees. I concur with the reviewer, and agree that it would be good to comment on the difference, and remark that the results are not completely comparable/equally strong. I would recommend acceptance conditioned on that these remarks are added in the final version of the paper.